# On Expert Estimation in Hierarchical Mixture of Experts: Beyond Softmax Gating Functions

## Abstract

With the growing prominence of the Mixture of Experts (MoE) architecture in developing large-scale foundation models, we investigate the Hierarchical Mixture of Experts (HMoE), a specialized variant of MoE that excels in handling complex inputs and improving performance on targeted tasks. Our investigation highlights the advantages of using varied gating functions, moving beyond softmax gating within HMoE frameworks. We theoretically demonstrate that applying tailored gating functions to each expert group allows HMoE to achieve robust results, even when optimal gating functions are applied only at select hierarchical levels. Empirical validation across diverse scenarios supports these theoretical claims. This includes large-scale multimodal tasks, image classification, and latent domain discovery and prediction tasks, where our modified HMoE models show great performance improvements.

## 1 Introduction

In recent years, the integration of mixture-of-experts (MoE) within large-scale foundation models has markedly advanced the machine learning field (Jiang et al., 2024; Fedus et al., 2022; Riquelme et al., 2021; Zhou et al., 2022; Mustafa et al., 2022). MoE architectures, known for their ability to efficiently handle diverse and complex datasets, have facilitated significant improvements in model performance without a proportional increase in computational demand. They address bottlenecks associated with traditional deep learning architectures by dynamically allocating resources to parts of the model for which they are most relevant (Yuksel et al., 2012; Shazeer et al., 2017). The Hierarchical Mixture of Experts (HMoE) model (Fritsch et al., 1996) is a special type of MoE architecture that is characterized by a layered structure of decision modules and expert networks that operate in tandem to refine decision-making at each level, optimizing the allocation of computational resources and enhancing specialization for complex tasks. Unlike the standard MoE, which typically involves a single gating network directing inputs to various expert networks, HMoE introduces multiple layers of gating mechanisms and experts. This hierarchical design divides the problem space recursively, allowing different experts to specialize in subspaces of the input space, leading to enhanced flexibility and model generalization (Jiang & Tanner, 1999; Azran & Meir, 2004).

Figure 1 compares HMoE and standard MoE in processing multimodal input data. The hierarchical structure of HMoE makes it particularly effective at handling complex inputs, such as data that can be divided into semantically meaningful subgroups. This recursive partitioning enables HMoE to select features and specialize in various segments of the input space more effectively, especially in high-dimensional data scenarios (Peralta & Soto, 2014). The unique capability of HMoE to handle complex datasets makes it particularly valuable across a range of applications. Historically, HMoE has been applied on image classification (Irsoy & Alpaydın, 2021), speech recognition (Peng et al., 1996; Zhao et al., 1994), and complex decision-making tasks (Jeremiah et al., 2013; Moges et al., 2016). However, there is a notable lack of recent studies on HMoE in the literature, partly due to its more complex structure as compared to standard MoE. For instance, while standard MoE requires the selection of a single gating function, HMoE necessitates the choice of multiple gating functions, introducing additional hyperparameters and therefore greater complication in model specification. Given the increasing complexity of input data in the modern era, such as multiple modalities or subgroups defined by ambiguous latent domains, there is a growing demand for models that can deliver accurate and individualized predictions for each subgroup. Therefore, it is worthwhile to study

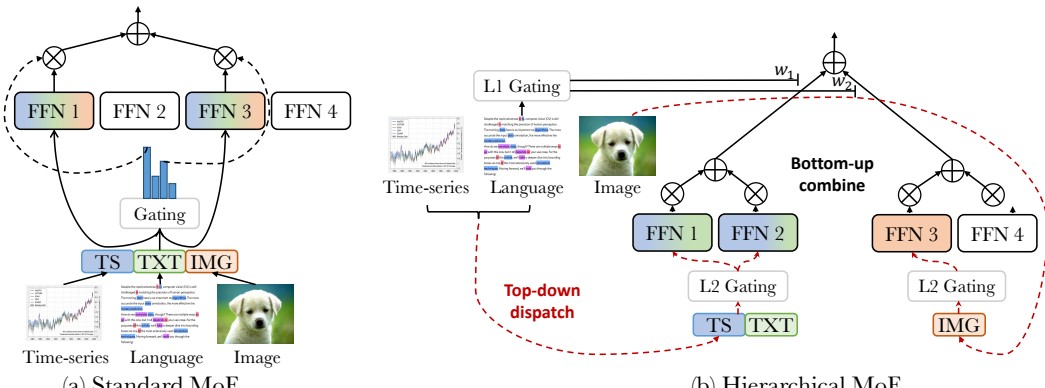

(a) Standard MoE          (b) Hierarchical MoE

Figure 1: Comparison of HMoE and standard MoE in managing multimodal input: MoE excels at processing homogeneous inputs. However, it faces challenges with more intricate structures, such as inputs that can be split into subgroups or those with inherently hierarchical configurations. By contrast, HMoE improves upon this by decomposing tasks into subproblems and directing subsets of data to specialized groups of experts. This approach allows for more granular specialization and enhances the model's capability to handle complex inputs.

HMoE, which can leverage the intrinsic information within complex input structures and achieve superior performance on corresponding tasks.

In this paper, we investigate distinct selections of gating functions within HMoE and their impact on overall performance. This is a critical issue and will lay the groundwork for future research in this relatively unstudied domain. It is important to note that expert specialization, as discussed in Dai et al. (2024), is a critical problem that involves understanding how quickly an expert becomes specialized in specific tasks or aspects of the data. To address this, we conduct a comprehensive analysis of the convergence behavior of experts within two-level HMoE models, using three different combinations of the conventional softmax gating (Jordan & Jacobs, 1994) and the Laplace gating as suggested in Han et al. (2024). Our theoretical analysis reveals that employing Laplace gating at both levels of the HMoE framework accelerates expert convergence and significantly improves performance relative to baseline. We further validate this through extensive empirical evaluations across diverse scenarios, demonstrating HMoE's effectiveness on complex datasets, such as those with inherent hierarchies or clustered data that can be partitioned into subgroups. By incorporating the three aforementioned combinations of gating functions, our experiments confirm that using Laplace gating at both levels consistently improves performance across multiple downstream tasks compared to the standard softmax gating baseline. Additionally, we observe that different combinations of Laplace and softmax gating can also noticeably enhance results, leading to better and more robust performance by offering a broader selection of gating function combinations. These findings highlight the practical benefits of selecting appropriate gating functions to enhance HMoE's capabilities.

**Notations.** We let $[n]$ stand for the set $\{1, 2, \ldots, n\}$ for any $n \in \mathbb{N}$. Next, for any set $S$, we denote $|S|$ as its cardinality. For any vector $v \in \mathbb{R}^d$ and $\alpha := (\alpha_1, \alpha_2, \ldots, \alpha_d) \in \mathbb{N}^d$, we let $v^\alpha = v_1^{\alpha_1} v_2^{\alpha_2} \ldots v_d^{\alpha_d}$, $|v| := v_1 + v_2 + \ldots + v_d$ and $\alpha! := \alpha_1! \alpha_2! \ldots \alpha_d!$, while $\|v\|$ stands for its $L^2$-norm value. For any two positive sequences $(a_n)_{n \geq 1}$ and $(b_n)_{n \geq 1}$, we write $a_n = \mathcal{O}(b_n)$ or $a_n \lesssim b_n$ if there exists $C > 0$ such that $a_n \leq C b_n$ for all $n \in \mathbb{N}$. Meanwhile, the notation $a_n = \mathcal{O}_P(b_n)$ indicates that $a_n/b_n$ is stochastically bounded. Lastly, for any two probability density functions $p, q$ dominated by the Lebesgue measure $\mu$, we denote $h^2(p, q) = \frac{1}{2} \int (\sqrt{p} - \sqrt{q})^2 d\mu$ as their squared Hellinger distance and $V(p, q) = \frac{1}{2} \int |p - q| d\mu$ as their Total Variation distance.

## 2 THEORETICAL CONTRIBUTIONS AND METHODS

We conduct a convergence analysis of expert estimation in the two-level Gaussian HMoE under three settings of alternatively using the Softmax gating and Laplace gating in the two levels of the model. Our goal is to find which gating combination would induce the fastest expert estimation rate.

### 2.1 SOFTMAX-SOFTMAX HMoE

We begin by considering the scenario when the two-level Gaussian HMoE is equipped with the Softmax gating in both levels. More specifically, let us assume that an i.i.d. sample of size $n$:

$(\boldsymbol{X}_1, Y_1), (\boldsymbol{X}_2, Y_2), \ldots, (\boldsymbol{X}_n, Y_n)$ in $\mathbb{R}^d \times \mathbb{R}$, where $\boldsymbol{X}_i$ is an input and $Y_i$ is a response variable, is generated from that model whose conditional density function is given by

$$p_{G_*}^{SS}(y|\boldsymbol{x}) := \sum_{i_1=1}^{k_1^*} \sigma((\boldsymbol{a}_{i_1}^*)^\top \boldsymbol{x} + b_{i_1}^*) \sum_{i_2=1}^{k_2^*} \sigma((\boldsymbol{\omega}_{i_2|i_1}^*)^\top \boldsymbol{x} + \beta_{i_2|i_1}^*) \pi(y|(\boldsymbol{\eta}_{i_1 i_2}^*)^\top \boldsymbol{x} + \tau_{i_1 i_2}^*, \nu_{i_1 i_2}^*).$$
(1)

Above, the abbreviation $SS$ stands for "Softmax-Softmax", indicating that the softmax gating is used in both levels of the Gaussian HMoE. Next, we define

$$G_* := \sum_{i_1=1}^{k_1^*} \exp(b_{i_1}^*) \sum_{i_2=1}^{k_2^*} \exp(\beta_{i_2|i_1}^*) \delta_{(\boldsymbol{a}_{i_1}^*, \boldsymbol{\omega}_{i_2|i_1}^*, \tau_{i_1 i_2}^*, \boldsymbol{\eta}_{i_1 i_2}^*, \nu_{i_1 i_2}^*)}$$

as a corresponding *mixing measure*, i.e., a weighted sum of Dirac measures $\delta$, where $(b_{i_1}^*, \boldsymbol{a}_{i_1}^*, \beta_{i_2|i_1}^*, \boldsymbol{\omega}_{i_2|i_1}^*, \tau_{i_1 i_2}^*, \boldsymbol{\eta}_{i_1 i_2}^*, \nu_{i_1 i_2}^*)$ are true yet unknown parameters in the parameter space $\Theta \subseteq \mathbb{R} \times \mathbb{R}^d \times \mathbb{R} \times \mathbb{R}^d \times \mathbb{R}^q \times \mathbb{R}_+$. Additionally, $k_1^*$ denotes the number of mixtures in the two-level Gaussian HMoE, whereas $k_2^*$ is the number of experts in each mixture. For any integer $k \in \mathbb{N}$ and real-valued vector $(v_i)_{i=1}^k$, we denote by $\sigma(v_i) := \exp(v_i)/\sum_{j=1}^k \exp(v_j)$ the softmax function. Meanwhile, $\pi(\cdot|\mu, \nu)$ is an univariate Gaussian density function with mean $\mu$ and variance $\nu$.

Recall that expert specialization is an essential problem in the MoE literature where we explore how fast an expert specializes in some tasks or some aspects of the data (Dai et al., 2024; Krishnamurthy et al., 2023), which can be captured through the convergence analysis of expert estimation.

**Maximum likelihood estimation (MLE).** To estimate the unknown parameters, or equivalently the unknown mixing measure $G_*$, we utilize the maximum likelihood method (van de Geer, 2000). For simplicity, we assume that the value of $k_1^*$ is known as the analysis would become unnecessarily complicated otherwise. At the same time, the value of $k_2^*$ remains unknown. Then, we over-specify the true model (1) by considering an MLE within a class of mixing measures with at most $k_1^* k_2$ components, where $k_2 > k_2^*$, as follows:

$$\widehat{G}_n^{SS} := \underset{G \in \mathcal{G}_{k_1^*, k_2}(\Theta)}{\arg\max} \frac{1}{n} \sum_{i=1}^n \log(p_G^{SS}(Y_i|\boldsymbol{X}_i)),$$
(2)

in which $\mathcal{G}_{k_1^*, k_2}(\Theta) := \Big\{ G = \sum_{i_1=1}^{k_1^*} \exp(b_{i_1}) \sum_{i_2=1}^{k_2'} \exp(\beta_{i_2|i_1}) \delta_{(\boldsymbol{a}_{i_1}, \boldsymbol{\omega}_{i_2|i_1}, \boldsymbol{\eta}_{i_1 i_2}, \tau_{i_1 i_2}, \nu_{i_1 i_2})} : k_2' \in [k_2], (b_{i_1}, \boldsymbol{a}_{i_1}, \beta_{i_2|i_1}, \boldsymbol{\omega}_{i_1 i_2}, \tau_{i_1 i_2}, \boldsymbol{\eta}_{i_1 i_2}, \nu_{i_1 i_2}) \in \Theta \Big\}$.

**Assumptions.** For the sake of theory, we make some following standard assumptions on the data as well as the model parameters throughout this paper:

*(A.1) We assume that the parameter space $\Theta$ is compact and the input space $\mathcal{X}$ is bounded to guarantee the MLE convergence.*

*(A.2) In order that the Gaussian HMoE is identifiable, that is, $p_G^{SS}(y|\boldsymbol{x}) = p_{G_*}^{SS}(y|\boldsymbol{x})$ for almost every $(\boldsymbol{x}, y)$ implies $G \equiv G_*$, the softmax gating value must not be invariant to parameter translation. Therefore, we let $\boldsymbol{a}_{k_1^*}^* = \boldsymbol{0}_d, b_{k_1^*}^* = 0$ and $\boldsymbol{\omega}_{k_2^*|i_1}^* = \boldsymbol{0}_d, \beta_{k_2^*|i_1}^* = 0$ for any $i_1 \in [k_1^*]$.*

*(A.3) For any $i_1 \in [k_1^*]$, let $(\boldsymbol{\eta}_{i_1 1}^*, \tau_{i_1 1}^*, \nu_{i_1 1}^*), \ldots, (\boldsymbol{\eta}_{i_1 k_2^*}^*, \tau_{i_1 k_2^*}^*, \nu_{i_1 k_2^*}^*)$ be distinct parameters so that the Gaussian distributions associated with the same parent node are different from each other.*

*(A.4) To ensure that the gating depend on the input, we assume at least one among gating parameters in the first level $\boldsymbol{a}_1^*, \ldots, \boldsymbol{a}_{k_1^*}^*$ (resp. those in the second level $\boldsymbol{\omega}_1^*, \ldots, \boldsymbol{\omega}_{k_1^*}^*$) is different from zero.*

Now, we investigate the convergence behavior of the density estimation $p_{\widehat{G}_n}^{SS}$ to the true density $p_{G_*}^{SS}$ in Theorem 1 whose proof can be found in Appendix F.

**Theorem 1.** *Given an MLE $\widehat{G}_n^{SS}$ defined in equation (2), the corresponding density estimation $p_{\widehat{G}_n^{SS}}^{SS}$ converges to the true density $p_{G_*}^{SS}$ under the Hellinger distance $h$ at following rate:*

$$\mathbb{P}(\mathbb{E}_{\boldsymbol{X}}[h(p_{\widehat{G}_n^{SS}}^{SS}(\cdot|\boldsymbol{X}), p_{G_*}^{SS}(\cdot|\boldsymbol{X}))] > C_1 \sqrt{\log(n)/n}) \lesssim \exp(-c_1 \log n),$$

*where $C_1$ and $c_1$ are universal constants.*

Theorem 1 indicates that the rate for estimating the true conditional density of the Gaussian HMoE is of parametric order $\widetilde{\mathcal{O}}_P(n^{-1/2})$. Consequently, if we are able to construct a loss function among parameters denoted by, for example, $\mathcal{L}(\widehat{G}_n, G_*)$, and establish the bound $\mathcal{L}(\widehat{G}_n, G_*) \lesssim \mathbb{E}_{\boldsymbol{X}}[h(p_{\widehat{G}_n^{SS}}^{SS}(\cdot|\boldsymbol{X}), p_{G_*}^{SS}(\cdot|\boldsymbol{X}))]$, then we will obtain the parameter estimation rates $\mathcal{L}(\widehat{G}_n, G_*) = \widetilde{\mathcal{O}}_P(n^{-1/2})$, which leads to our desired rates for estimating experts. However, while such Hellinger bound has been well studied under the setting of one-level Gaussian MoE Ho et al. (2022); Nguyen et al. (2023), it has remained elusive for the hierarchical setting. In the following paragraph, we will point out fundamental obstacles for deriving that bound.

**Challenges.** Our main technique for deriving the parameter estimation rates is to decompose the density estimation and the true density, i.e. $p_{\widehat{G}_n^{SS}}^{SS}(y|\boldsymbol{x}) - p_{G_*}^{SS}(y|\boldsymbol{x})$, into a combination of linearly independent terms by applying the Taylor expansion to the function $u(\boldsymbol{x}; \boldsymbol{a}, \boldsymbol{\omega}, \boldsymbol{\eta}, \tau, \nu) := \exp(\boldsymbol{a}^\top \boldsymbol{x}) \exp(\boldsymbol{\omega}^\top \boldsymbol{x}) \pi(y|\boldsymbol{\eta}^\top \boldsymbol{x} + \tau, \nu)$ with respect to its parameters. In previous works (Ho et al., 2022; Nguyen et al., 2023), it is well-known that there is an interaction between the mean parameter $\tau$ and variance $\nu$ of the Gaussian density via the partial differential equation (PDE) $\frac{\partial u}{\partial \nu} = \frac{1}{2} \cdot \frac{\partial^2 u}{\partial \tau^2}$. Such PDE induces several linearly dependent terms in the aforementioned decomposition, thereby leading to significantly slow rates for estimating those parameters. In this paper, we discover that the first-level gating parameter $a$ also interacts with the second-level parameters $\boldsymbol{\omega}, \nu, \tau$, that is,

$$\frac{\partial u}{\partial \boldsymbol{\eta}} = \frac{\partial^2 u}{\partial \boldsymbol{a} \partial \tau}, \quad \frac{\partial u}{\partial \boldsymbol{a}} = \frac{\partial u}{\partial \boldsymbol{\omega}}. \tag{3}$$

To the best of our knowledge, these intrinsic interactions have not been noted before in the literature. Therefore, we have to take the solvability of the unforeseen system of poylnomial equations (4) into account to capture that interaction.

**System of polynomial equations.** For each $m \geq 2$, we define $r^{SS}(m)$ as the smallest natural number $r$ such that the following system does not have any non-trivial solutions for the unknown variables $(p_{i_2}, \boldsymbol{q}_{1i_2}, \boldsymbol{q}_2, \boldsymbol{q}_{3i_2}, q_{4i_2}, q_{5i_2})_{i_2=1}^m$

$$\sum_{i_2=1}^m \sum_{(\boldsymbol{\alpha}_1, \boldsymbol{\alpha}_2, \boldsymbol{\alpha}_3, \alpha_4, \alpha_5) \in \mathcal{I}_{\boldsymbol{\rho}_1, \rho_2}^{SS}} \frac{1}{\boldsymbol{\alpha}!} \cdot p_{i_2}^2 \boldsymbol{q}_{1i_2}^{\boldsymbol{\alpha}_1} \boldsymbol{q}_2^{\boldsymbol{\alpha}_2} \boldsymbol{q}_{3i_2}^{\boldsymbol{\alpha}_3} q_{4i_2}^{\alpha_4} q_{5i_2}^{\alpha_5} = 0, \quad 1 \leq |\boldsymbol{\rho}_1| + \rho_2 \leq r, \tag{4}$$

where $\mathcal{I}_{\boldsymbol{\rho}_1, \rho_2}^{SS} := \{(\boldsymbol{\alpha}_1, \boldsymbol{\alpha}_2, \boldsymbol{\alpha}_3, \alpha_4, \alpha_5) \in \mathbb{R}^d \times \mathbb{R}^d \times \mathbb{R}^d \times \mathbb{R} \times \mathbb{R}_+ : \boldsymbol{\alpha}_1 + \boldsymbol{\alpha}_2 + \boldsymbol{\alpha}_3 = \boldsymbol{\rho}_1, |\boldsymbol{\alpha}_3| + \alpha_4 + 2\alpha_5 = \rho_2\}$. Here, a solution is categorized as non-trivial if all the values of $p_{i_2}$ are different from zero and at least one among $q_{4i_2}$ is non-zero. Note that $r^{SS}(m)$ is a monotonically increasing function. However, finding the exact value of $r^{SS}(m)$ is a demanding problem in the field of algebraic geometry (Sturmfels, 2002). Thus, we provide in Lemma 1 (whose proof is in Appendix G) some specific values of $r^{SS}(m)$ when $m$ is small, while those for larger $m$ are left for future development.

**Lemma 1.** *For any $d \geq 1$, we have that $r^{SS}(2) = 4$ and $r^{SS}(3) = 6$, while we conjecture that $r^{SS}(m) \geq 7$ for $m \geq 4$.*

**Voronoi loss.** To precisely characterize the convergence rate of parameter estimation, it is necessary to capture the number of fitted parameters approaching each individual true parameter in both levels of Gaussian HMoE. For that purpose, let us introduce the concept of Voronoi cells (Manole & Ho, 2022). In particular, given an arbitrary mixing measure $G \in \mathcal{G}_{k_1^* k_2}(\Theta)$, we distribute its atoms across the Voronoi cells $\{\mathcal{V}_{j_1}(G), j_1 \in [k_1^*]\}$ and $\{\mathcal{V}_{j_2|j_1}(G), j_1 \in [k_1^*], j_2 \in [k_2^*]\}$ generated by the atoms of $G_*$, where

$$\mathcal{V}_{j_1} \equiv \mathcal{V}_{j_1}(G) := \{i_1 \in [k_1^*] : \|\boldsymbol{a}_{i_1} - \boldsymbol{a}_{j_1}^*\| \leq \|\boldsymbol{a}_{i_1} - \boldsymbol{a}_{\ell_1}^*\|, \forall \ell_1 \neq j_1\},$$

$$\mathcal{V}_{j_2|j_1} \equiv \mathcal{V}_{j_2|j_1}(G) := \{i_2 \in [k_2] : \|\boldsymbol{\zeta}_{i_2|j_1} - \boldsymbol{\zeta}_{j_2|j_1}^*\| \leq \|\boldsymbol{\zeta}_{i_2|j_1} - \boldsymbol{\zeta}_{\ell_2|j_1}^*\|, \forall \ell_2 \neq j_2\},$$

with $\boldsymbol{\zeta}_{i_2|j_1} := (\boldsymbol{\omega}_{i_2|j_1}, \boldsymbol{\eta}_{j_1 i_2}, \tau_{j_1 i_2}, \nu_{j_1 i_2})$ and $\boldsymbol{\zeta}_{j_2|j_1}^* := (\boldsymbol{\omega}_{j_2|j_1}^*, \boldsymbol{\eta}_{j_2|j_1}^*, \tau_{j_1 j_2}^*, \nu_{j_1 j_2}^*)$. Note that when the MLE $\widehat{G}_n$ is sufficiently close to its true counterpart $G_*$, since the value of $k_1^*$ is known, we have $|\mathcal{V}_{j_1}(\widehat{G}_n)| = 1$ for any $j_1 \in [k_1^*]$, meaning that each parameter $a_{j_1}^*$ is fitted by exactly one parameter. On the other hand, as $k_2^*$ is unknown and we over-specify it by a larger value $k_2$, a Voronoi cell $\mathcal{V}_{j_2|j_1}$ could have more than one element. Furthermore, the cardinality of $\mathcal{V}_{j_2|j_1}$ is exactly the number of

fitted parameters converging to $\zeta^*_{j_2|j_1}$. For instance, $|\mathcal{V}_{j_2|j_1}| = 2$ indicates that $\zeta^*_{j_2|j_1}$ is fitted by two parameters. Now, we define a Voronoi loss function based on the Voronoi cells as follows:

$$\mathcal{L}_{(r_1,r_2,r_3)}(G, G_*) := \sum_{j_1=1}^{k_1^*} \Big| \sum_{i_1 \in \mathcal{V}_{j_1}} \exp(b_{i_1}) - \exp(b^*_{j_1}) \Big| + \sum_{j_1=1}^{k_1^*} \sum_{i_1 \in \mathcal{V}_{j_1}} \exp(b_{i_1}) \|\Delta \boldsymbol{a}_{i_1 j_1}\|$$

$$+ \sum_{j_1=1}^{k_1^*} \sum_{i_1 \in \mathcal{V}_{j_1}} \exp(b_{i_1}) \Bigg[ \sum_{j_2:|\mathcal{V}_{j_2|j_1}|=1} \sum_{i_2 \in \mathcal{V}_{j_2|j_1}} \exp(\beta_{i_2|j_1}) \Big( \|\Delta \boldsymbol{\omega}_{i_2 j_2|j_1}\| + \|\Delta \boldsymbol{\eta}_{j_1 i_2 j_2}\| + |\Delta \tau_{j_1 i_2 j_2}| + |\Delta \nu_{j_1 i_2 j_2}| \Big)$$

$$+ \sum_{j_2:|\mathcal{V}_{j_2|j_1}|>1} \sum_{i_2 \in \mathcal{V}_{j_2|j_1}} \exp(\beta_{i_2|j_1}) \Big( \|\Delta \boldsymbol{\omega}_{i_2 j_2|j_1}\|^2 + \|\Delta \boldsymbol{\eta}_{j_1 i_2 j_2}\|^{r_1(|\mathcal{V}_{j_2|j_1}|)} + |\Delta \tau_{j_1 i_2 j_2}|^{r_2(|\mathcal{V}_{j_2|j_1}|)}$$

$$+ |\Delta \nu_{j_1 i_2 j_2}|^{r_3(|\mathcal{V}_{j_2|j_1}|)} \Big) \Bigg] + \sum_{j_1=1}^{k_1^*} \sum_{i_1 \in \mathcal{V}_{j_1}} \exp(b_{i_1}) \sum_{j_2=1}^{k_2^*} \Big| \sum_{i_2 \in \mathcal{V}_{j_2|j_1}} \exp(\beta_{i_2|j_1}) - \exp(\beta^*_{j_2|j_1}) \Big|, \tag{5}$$

where $r_1, r_2, r_3 : \mathbb{N} \to \mathbb{N}$ are some integer-valued functions and we denote $\Delta \boldsymbol{a}_{i_1 j_1} := \boldsymbol{a}_{i_1} - \boldsymbol{a}^*_{j_1}$, $\Delta \boldsymbol{\omega}_{i_2 j_2|j_1} := \boldsymbol{\omega}_{i_2|j_1} - \boldsymbol{\omega}_{j_2|j_1}$, $\Delta \boldsymbol{\eta}_{j_1 i_2 j_2} := \boldsymbol{\eta}_{j_1 i_2} - \boldsymbol{\eta}^*_{j_1 j_2}$, $\Delta \tau_{j_1 i_2 j_2} := \tau_{j_1 i_2} - \tau^*_{j_1 j_2}$ and $\Delta \nu_{j_1 i_2 j_2} := \nu_{j_1 i_2} - \nu^*_{j_1 j_2}$. Given the above loss function, we are ready to characterize the convergence behavior of expert estimation in the following theorem.

**Theorem 2.** *The following Hellinger lower bounds hold true for any $G \in \mathcal{G}_{k_1^*, k_2}(\Theta)$:*

$$\mathbb{E}_{\boldsymbol{X}}[h(p_G^{SS}(\cdot|\boldsymbol{X}), p_{G_*}^{SS}(\cdot|\boldsymbol{X}))] \gtrsim \mathcal{L}_{(\frac{1}{2}r^{SS}, r^{SS}, \frac{1}{2}r^{SS})}(G, G_*).$$

*As a result, we obtain that $\mathcal{L}_{(\frac{1}{2}r^{SS}, r^{SS}, \frac{1}{2}r^{SS})}(\widehat{G}_n^{SS}, G_*) = \widetilde{\mathcal{O}}_P(n^{-1/2})$.*

Proof of Theorem 2 is in Appendix E. The above results together with the formulation of the Voronoi loss $\mathcal{L}_{(\frac{1}{2}r^{SS}, r^{SS}, \frac{1}{2}r^{SS})}$ in equation (5) implies that

**(i) Exact-specified parameters:** The rates for estimating exact-specified parameters $\boldsymbol{a}^*_{j_1}, \boldsymbol{\omega}^*_{j_2|j_1}, \boldsymbol{\eta}^*_{j_1 j_2}, \tau^*_{j_1 j_2}, \nu^*_{j_1 j_2}$ which are approached by exactly one fitted parameter, i.e. their Voronoi cells have only one element $|\mathcal{V}_{j_1}| = |\mathcal{V}_{j_2|j_1}| = 1$, are parametric on the sample size $n$, standing at the order $\widetilde{\mathcal{O}}_P(n^{-1/2})$. Additionally, the gating bias parameters $\exp(b^*_{j_1})$ and $\exp(\beta^*_{j_2|j_1})$ also share the same parametric estimation rates.

**(ii) Over-specified parameters:** For over-specified parameters $\boldsymbol{\omega}^*_{j_2|j_1}, \boldsymbol{\eta}^*_{j_1 j_2}, \tau^*_{j_1 j_2}, \nu^*_{j_1 j_2}$ which are fitted by more than one parameter, i.e. $|\mathcal{V}_{j_2|j_1}| > 1$, their estimation rates are not homogeneous. In particular, the rates for estimating $\boldsymbol{\omega}^*_{j_2|j_1}$ are of order $\widetilde{\mathcal{O}}_P(n^{-1/4})$. At the same time, those for $\boldsymbol{\eta}^*_{j_1 j_2}, \tau^*_{j_1 j_2}, \nu^*_{j_1 j_2}$ depend on their number of fitted parameters $|\mathcal{V}_{j_2|j_1}|$ and the solvability of the polynomial equation system in equation (4), standing at the orders of $\widetilde{\mathcal{O}}_P(n^{-1/r^{SS}(|\mathcal{V}_{j_2|j_1}|)}), \widetilde{\mathcal{O}}_P(n^{-1/2r^{SS}(|\mathcal{V}_{j_2|j_1}|)}), \widetilde{\mathcal{O}}_P(n^{-1/r^{SS}(|\mathcal{V}_{j_2|j_1}|)})$, respectively. For instance, when $|\mathcal{V}_{j_2|j_1}| = 3$, these rates become $\widetilde{\mathcal{O}}_P(n^{-1/6}), \widetilde{\mathcal{O}}_P(n^{-1/12}), \widetilde{\mathcal{O}}_P(n^{-1/6})$, which are significantly slower than those for exact-specified parameters. These slow rates occur due to the interactions mentioned in the "Challenges" paragraph.

**(iii) Expert estimation:** Recall that expert specialization is an essential problem where we learn how fast an expert specializes in some tasks or some aspects of the data. Therefore, it is important to understand the convergence behavior of the expert estimation, particularly its data-dependent term $(\boldsymbol{\eta}^*_{j_1 j_2})^\top \boldsymbol{x}$. According to the Cauchy-Schwarz inequality, we have

$$|(\hat{\boldsymbol{\eta}}_{i_1 i_2}^{SS,n})^\top \boldsymbol{x} - (\boldsymbol{\eta}^*_{j_1 j_2})^\top \boldsymbol{x}| \le \|\hat{\boldsymbol{\eta}}_{i_1 i_2}^{SS,n} - \boldsymbol{\eta}^*_{j_1 j_2}\| \cdot \|\boldsymbol{x}\|, \tag{6}$$

where $\hat{\boldsymbol{\eta}}_{i_1 i_2}^{SS,n}$ is an MLE of $\boldsymbol{\eta}^*_{j_1 j_2}$. Since the input space is bounded and from the estimation rate of $\boldsymbol{\eta}^*_{j_1 j_2}$ in the above two remarks, we deduce that $(\boldsymbol{\eta}^*_{j_1 j_2})^\top \boldsymbol{x}$ admits an estimation rate of order $\widetilde{\mathcal{O}}_P(n^{-1/2})$ when $|\mathcal{V}_{j_2|j_1}| = 1$ or $\widetilde{\mathcal{O}}_P(n^{-1/r^{SS}(|\mathcal{V}_{j_2|j_1}|)})$ when $|\mathcal{V}_{j_2|j_1}| > 1$. Note that the latter rate is significantly slow since the term $r^{SS}(|\mathcal{V}_{j_2|j_1}|)$ grows as the number of fitted experts $|\mathcal{V}_{j_2|j_1}|$ increases.

## 2.2 SOFTMAX-LAPLACE HMoE

Moving to this section, we study the effects of replacing the softmax gating in the second level with the Laplace gating on the convergence of expert estimation under the Gaussian HMoE. In particular, the conditional density function in equation (1) becomes

$$p_{G_*}^{SL}(y|\boldsymbol{x}) := \sum_{i_1=1}^{k_1^*} \sigma((\boldsymbol{a}_{i_1}^*)^\top \boldsymbol{x} + b_{i_1}^*) \sum_{i_2=1}^{k_2^*} \sigma(-\|\boldsymbol{\omega}_{i_2|i_1}^* - \boldsymbol{x}\| + \beta_{i_2|i_1}^*) \pi(y|(\boldsymbol{\eta}_{i_1 i_2}^*)^\top \boldsymbol{x} + \tau_{i_1 i_2}^*, \nu_{i_1 i_2}^*),$$

(7)

where the abbreviation $SL$ stands for "Softmax-Laplace". Additionally, the MLE under this setting, denoted by $\widehat{G}_n^{SL}$, is determined similarly to that in equation (2). The main difference between the density $p_{G_*}^{SL}(y|\boldsymbol{x})$ from its counterpart $p_{G_*}^{SS}(y|\boldsymbol{x})$ is the Laplace gating function $\sigma(-\|\boldsymbol{\omega}_{i_2|i_1}^* - \boldsymbol{x}\| + \beta_{i_2|i_1}^*)$ in the second level. Due to this gating change, the interaction between parameters $\boldsymbol{a}$ and $\boldsymbol{\omega}$ via the PDE $\frac{\partial u}{\partial \boldsymbol{a}} = \frac{\partial u}{\partial \boldsymbol{\omega}}$ in equation (3) no longer holds true, while others still exist. As a consequence, we only need to consider a simpler (fewer variables) system of polynomial equations than that in equation (4). More specifically, for each $m \geq 2$, we define $r^{SL}(m)$ as the smallest natural number $r$ such that the following system does not have any non-trivial solutions for the unknown variables $(p_{i_2}, \boldsymbol{q}_2, \boldsymbol{q}_{3i_2}, q_{4i_2}, q_{5i_2})_{i_2=1}^m$:

$$\sum_{i_2=1}^{m} \sum_{(\boldsymbol{\alpha}_2, \boldsymbol{\alpha}_3, \alpha_4, \alpha_5) \in \mathcal{I}_{\boldsymbol{\rho}_1, \rho_2}^{SL}} \frac{1}{\boldsymbol{\alpha}!} \cdot p_{i_2}^2 \boldsymbol{q}_2^{\boldsymbol{\alpha}_2} \boldsymbol{q}_{3i_2}^{\boldsymbol{\alpha}_3} q_{4i_2}^{\alpha_4} q_{5i_2}^{\alpha_5} = 0, \quad 1 \leq |\boldsymbol{\rho}_1| + \rho_2 \leq r,$$

(8)

where $\mathcal{I}_{\boldsymbol{\rho}_1, \rho_2}^{SL} := \{(\boldsymbol{\alpha}_2, \boldsymbol{\alpha}_3, \alpha_4, \alpha_5) \in \mathbb{R}^d \times \mathbb{R}^d \times \mathbb{R} \times \mathbb{R}_+ : \boldsymbol{\alpha}_2 + \boldsymbol{\alpha}_3 = \boldsymbol{\rho}_1, |\boldsymbol{\alpha}_3| + \alpha_4 + 2\alpha_5 = \rho_2\}$. Here, a solution is called non-trivial if all the values of $p_{i_2}$ are different from zero and at least one among $q_{4i_2}$ is non-zero. This system has been considered in Nguyen et al. (2023) where they show that $r^{SL}(2) = 4$ and $r^{SL}(3) = 6$. We observe that the function $r^{SL}$ admits identical behavior to the function $r^{SS}$ in Lemma 1 at some particular points. Nevertheless, it is challenging to make an explicit comparison between these two functions, which requires further technical tools in algebraic geometry Sturmfels (2002) to be developed.

Next, note that we can achieve the density estimation rate $\mathbb{E}_{\boldsymbol{X}}[h(p_{\widehat{G}_n^{SL}}^{SL}(\cdot|\boldsymbol{X}), p_{G_*}^{SL}(\cdot|\boldsymbol{X}))] = \widetilde{\mathcal{O}}_P(n^{-1/2})$ using similar arguments for Theorem 1 (see Appendix F). Thus, we will present only the convergence of parameter and expert estimation under the setting of this section in Theorem 3.

**Theorem 3.** *The following Hellinger lower bounds hold true for any $G \in \mathcal{G}_{k_1^*, k_2}(\Theta)$:*

$$\mathbb{E}_{\boldsymbol{X}}[h(p_G^{SL}(\cdot|\boldsymbol{X}), p_{G_*}^{SL}(\cdot|\boldsymbol{X}))] \gtrsim \mathcal{L}_{(\frac{1}{2}r^{SL}, r^{SL}, \frac{1}{2}r^{SL})}(G, G_*).$$

*As a result, we obtain that $\mathcal{L}_{(\frac{1}{2}r^{SL}, r^{SL}, \frac{1}{2}r^{SL})}(\widehat{G}_n^{SL}, G_*) = \widetilde{\mathcal{O}}_P(n^{-1/2})$.*

Proof of Theorem 3 is in Appendix E. From the above results, it can be seen that the parameter and expert estimation when using the softmax gating and Laplace gating in the first and second levels of the Gaussian HMoE share the same convergence behavior as those when using the softmax gating in both levels in Theorem 2. In particular, by arguing analogously to equation (6), we get that the data-dependent term of expert $(\boldsymbol{\eta}_{j_1 j_2}^*)^\top \boldsymbol{x}$ has an estimation rate of order $\widetilde{\mathcal{O}}_P(n^{-1/2})$ when $|\mathcal{V}_{j_2|j_1}| = 1$ or $\widetilde{\mathcal{O}}_P(n^{-1/r^{SL}(|\mathcal{V}_{j_2|j_1}|)})$ when $|\mathcal{V}_{j_2|j_1}| > 1$. Thus, we can see that substituting the softmax gating with the Laplace gating in the second level is not enough to accelerate the expert estimation rate (see Table 1). This is because the interaction $\frac{\partial u}{\partial \boldsymbol{\eta}} = \frac{\partial^2 u}{\partial \boldsymbol{a} \partial \tau}$ between $\boldsymbol{\eta}$ and other parameters in equation (3) still occurs under the setting of softmax-Laplace gating Gaussian HMoE.

## 2.3 LAPLACE-LAPLACE HMoE

In this section, we consider the two-level Gaussian HMoE equipped with the Laplace gating in both levels. More specifically, the conditional density function in equation (7) turns into

$$p_{G_*}^{LL}(y|\boldsymbol{x}) := \sum_{i_1=1}^{k_1^*} \sigma(-\|\boldsymbol{a}_{i_1}^* - \boldsymbol{x}\| + b_{i_1}^*) \sum_{i_2=1}^{k_2^*} \sigma(-\|\boldsymbol{\omega}_{i_2|i_1}^* - \boldsymbol{x}\| + \beta_{i_2|i_1}^*) \pi(y|(\boldsymbol{\eta}_{i_1 i_2}^*)^\top \boldsymbol{x} + \tau_{i_1 i_2}^*, \nu_{i_1 i_2}^*),$$

(9)

where the abbreviation $LL$ stands for "Laplace-Laplace". Furthermore, the definition of the MLE under this setting, denoted by $\widehat{G}_n^{LL}$, is determined similarly to that in equation (2). Under this setting, the first-level softmax gating $\sigma((\boldsymbol{a}_{i_1}^*)^\top \boldsymbol{x} + b_{i_1}^*)$ used in previous sections is replaced with the Laplace gating $\sigma(-\|\boldsymbol{a}_{i_1}^* - \boldsymbol{x}\| + b_{i_1}^*)$, leading to the disappearance of the interaction $\frac{\partial u}{\partial \boldsymbol{\eta}} = \frac{\partial^2 u}{\partial \boldsymbol{a} \partial \tau}$ between $\boldsymbol{\eta}$ and other parameters mentioned in equation (3). Therefore, we only need to cope with $\frac{\partial u}{\partial \nu} = \frac{1}{2} \cdot \frac{\partial^2 u}{\partial \tau^2}$ as in Ho et al. (2022). Consequently, it is sufficient to take account of the following system of polynomial equations with substantially fewer variables than those in equations (4) and (8). In particular, for each $m \geq 2$, we define $r^{LL}(m)$ as the smallest natural number $r$ such that the following system does not have any non-trivial solutions for the unknown variables $(p_{i_2}, q_{4i_2}, q_{5i_2})_{i_2=1}^m$:

$$\sum_{i_2=1}^m \sum_{(\alpha_4, \alpha_5) \in \mathcal{I}_\rho^{LL}} \frac{1}{\boldsymbol{\alpha}!} \cdot p_{i_2}^2 q_{4i_2}^{\alpha_4} q_{5i_2}^{\alpha_5} = 0, \quad 1 \leq \rho \leq r, \tag{10}$$

where $\mathcal{I}_\rho^{LL} := \{(\alpha_4, \alpha_5) \in \mathbb{R} \times \mathbb{R}_+ : \alpha_4 + 2\alpha_5 = \rho\}$. Here, a solution is called non-trivial if all the values of $p_{i_2}$ are different from zero and at least one among $q_{4i_2}$ is non-zero. The above system has been studied in Ho & Nguyen (2016) which show that $r^{LL}(2) = 4$ and $r^{LL}(3) = 6$. These values are similar to those of the aforementioned functions $r^{SS}$ and $r^{SL}$.

As demonstrated in Appendix F, we also obtain the convergence rate of density estimation $\mathbb{E}_{\boldsymbol{X}}[h(p_{\widehat{G}_n^{LL}}^{LL}(\cdot|\boldsymbol{X}), p_{G_*}^{LL}(\cdot|\boldsymbol{X}))] = \widetilde{\mathcal{O}}_P(n^{-1/2})$ under this setting. Given that result, we are ready to investigate the impacts of using the Laplace gating in both levels on the convergence behavior of parameter and expert estimation in the below theorem.

**Theorem 4.** *The following Hellinger lower bounds hold true for any $G \in \mathcal{G}_{k_1^*, k_2}(\Theta)$:*

$$\mathbb{E}_{\boldsymbol{X}}[h(p_G^{LL}(\cdot|\boldsymbol{X}), p_{G_*}^{LL}(\cdot|\boldsymbol{X}))] \gtrsim \mathcal{L}_{(2, r^{LL}, \frac{1}{2}r^{LL})}(G, G_*).$$

*As a result, we obtain that $\mathcal{L}_{(2, r^{LL}, \frac{1}{2}r^{LL})}(\widehat{G}_n^{LL}, G_*) = \widetilde{\mathcal{O}}_P(n^{-1/2})$.*

Proof of Theorem 4 is in Appendix E. From the formulation of the loss function $\mathcal{L}_{(2, r^{LL}, \frac{1}{2}r^{LL})}$ in equation (5), we observe that all the parameter estimations share the same convergence behavior as those under the previous two settings, except for the estimation of $\boldsymbol{\eta}_{j_1j_2}^*$ which enjoys a convergence rate of order $\widetilde{\mathcal{O}}_P(n^{-1/2})$ when $|\mathcal{V}_{j_2|j_1}| = 1$ or $\widetilde{\mathcal{O}}_P(n^{-1/4})$ when $|\mathcal{V}_{j_2|j_1}| > 1$. By employing the same arguments as in equation (6), we deduce that the data-dependent term of expert $(\boldsymbol{\eta}_{j_1j_2}^*)^\top \boldsymbol{x}$ also admits these rates. Compared to those when using the softmax gating in either level or both levels, the expert estimation rates when using the Laplace gating in both levels are improved significantly as they no longer depend on the term $r^{LL}(|\mathcal{V}_{j_2|j_1}|)$ (see Table 1). This rate acceleration occurs since the interaction $\frac{\partial u}{\partial \boldsymbol{\eta}} = \frac{\partial^2 u}{\partial \boldsymbol{a} \partial \tau}$ between $\boldsymbol{\eta}$ and other parameters mentioned in equation (3) does not exist under this setting. As a result, we claim that the convergence of expert estimation under the two-level Gaussian HMoE is benefited the most when equipped with the Laplace gating in both levels.

Table 1: Summary of estimation rates for the data-dependent term $(\boldsymbol{\eta}_{j_1j_2}^*)^\top \boldsymbol{x}$ in experts. Below, experts are called exact-specified when $|\mathcal{V}_{j_2|j_1}| = 1$ and over-specified when $|\mathcal{V}_{j_2|j_1}| > 1$.

| **Gating** | | Softmax-Softmax | Softmax-Laplace | Laplace-Laplace |
|---|---|---|---|---|
| **Expert estima-tion rates** | Exact-specified | $\widetilde{\mathcal{O}}_P(n^{-1/2})$ | $\widetilde{\mathcal{O}}_P(n^{-1/2})$ | $\widetilde{\mathcal{O}}_P(n^{-1/2})$ |
| | Over-specified | $\widetilde{\mathcal{O}}_P(n^{-1/r^{SS}(|\mathcal{V}_{j_2|j_1}|)})$ | $\widetilde{\mathcal{O}}_P(n^{-1/r^{SL}(|\mathcal{V}_{j_2|j_1}|)})$ | $\boldsymbol{\widetilde{\mathcal{O}}_P(n^{-1/4})}$ |

## 3 EXPERIMENTS

In this section, we empirically demonstrate the effects of employing various combinations of gating functions in HMoE to validate our theoretical findings and discuss empirical insights. First, we show that HMoE outperforms standard MoE and other alternatives, particularly in cases with inherent subgroups or multilevel structures, where HMoE excels. We then conduct comprehensive ablation studies to analyze the impact of different gating function combinations and perform case studies across various scenarios. Beyond performance improvements, these experiments provide valuable insights into how different gating function combinations influence the distribution of input modules, offering explanations for the performance variations observed with different gating configurations.

Table 2: Comparison of HMoE-based methods (gray) and baselines, utilizing vital signs and clinical notes of MIMIC-IV (Johnson et al., 2020). The best results are highlighted in **bold font**, and the second-best results are underlined. All results are averaged across 5 random experiments.

| | | MulT | MAG | TFN | HAIM | MISTS | MoE | HMoE |
|---|---|---|---|---|---|---|---|---|
| 48-IHM | AUROC | $75.56 \pm 0.34$ | $79.36 \pm 0.25$ | $79.12 \pm 0.56$ | $78.87 \pm 0.00$ | $77.23 \pm 0.82$ | $83.13 \pm 0.36$ | **$85.59 \pm 0.44$** |
| | F1 | $38.65 \pm 0.25$ | $40.87 \pm 0.17$ | $40.96 \pm 0.37$ | $39.78 \pm 0.00$ | $45.98 \pm 0.49$ | $46.82 \pm 0.28$ | **$47.57 \pm 0.32$** |
| LOS | AUROC | $82.12 \pm 0.98$ | $81.94 \pm 0.36$ | $81.65 \pm 0.43$ | $82.46 \pm 0.00$ | $80.34 \pm 0.61$ | $83.76 \pm 0.59$ | **$86.26 \pm 0.61$** |
| | F1 | $73.16 \pm 0.51$ | $72.78 \pm 0.22$ | $73.89 \pm 0.52$ | $72.75 \pm 0.00$ | $73.22 \pm 0.43$ | $74.32 \pm 0.44$ | **$76.07 \pm 0.29$** |
| 25-PHE | AUROC | $70.41 \pm 0.44$ | $71.17 \pm 0.36$ | $72.26 \pm 0.27$ | $63.57 \pm 0.00$ | $71.49 \pm 0.59$ | **$73.87 \pm 0.71$** | $73.81 \pm 0.51$ |
| | F1 | $32.33 \pm 0.62$ | $32.86 \pm 0.19$ | $34.24 \pm 0.14$ | $42.80 \pm 0.00$ | $33.29 \pm 0.23$ | **$35.96 \pm 0.23$** | $35.64 \pm 0.18$ |

**HMoE Implementation.** We implement the two-level HMoE module, inspired by Lepikhin et al. (2020). Algorithm 1 in Appendix outlines the procedure, which employs a recursive computation strategy to process inputs in a coarse-to-fine manner. The inputs are first partitioned by the outer dispatcher, followed by the inner dispatcher, into subgroups, which are then sent to specialized groups and experts for independent processing. The outputs from the experts are recursively combined using inner and outer combination tensors to produce the final output. Gating losses from both levels are integrated and scaled to regularize training, promoting balanced expert utilization.

### 3.1 PRIMARY RESULTS

**HMoE Improves Multimodal Fusion.** We first evaluate the effectiveness of HMoE on the MIMIC-IV dataset, a comprehensive database containing records from nearly 300k patients admitted to a medical center between 2008 and 2019, focusing on a subset of 73,181 ICU stays. We integrated diverse patient modalities, including vital signs (time series), clinical notes, and CXR (chest X-ray images). Our tasks of interest in the MIMIC dataset include 48-hour in-hospital mortality prediction (48-IHM), 25-type phenotype classification (25-PHE), and length-of-stay (LOS) prediction. The baselines include: (1) the Multimodal Transformer (MulT), which models modality interactions (Tsai et al., 2019); (2) the Multimodal Adaptation Gate (MAG), which addresses consistency and differences across modalities (Rahman et al., 2020); (3) the early fusion method Tensor Fusion Network (TFN) (Zadeh et al., 2017); (4) the HAIM data pipeline (Soenksen et al., 2022), specifically designed for integrating multimodal data from MIMIC-IV; (5) MISTS, a cross-attention approach combined with irregular sequence modeling (Zhang et al., 2023); and (6) multimodal fusion using MoE (Han et al., 2024). The data is first processed by modality-specific encoders, with the obtained modality embeddings then fed into 12 stacked HMoE modules with residual connections to produce the outcome. Details of the building blocks are provided in the appendix. Table 2 presents the outcomes of integrating time series, clinical notes, and CXR data into various prediction tasks. The HMoE (Laplace-Laplace) outperforms the baselines in most scenarios, often by a significant margin. While the MoE-based fusion method (Han et al., 2024) has proven effective in multimodal fusion, the inherent hierarchical structure of the HMoE module further enhances its ability to process multimodal inputs, allowing for more specialized expert assignment and improved performance.

**HMoE Enhances Clinical Latent Domain Discovery.** Many datasets in high-stakes applications can be categorized into different latent domains. For instance, in clinical prediction tasks, patients can be grouped based on latent domains such as age, medical history, treatment, and symptoms. Training a generic model on heterogeneous patient data is often less effective than using a domain-specific model, as demonstrated by the SLDG method proposed by Wu et al. (2024). However, SLDG assigns a fixed classifier to each domain without considering the interactions between them. Moreover, it relies heavily on a separate hierarchical clustering process, which is separated from model training and limits input data to low-dimensional forms like short time series, failing to utilize a broader range of patient modalities. We extend this framework by evaluating HMoE for latent domain modeling tasks, using the HMoE module as a substitute for domain-specific classifiers. The HMoE module partitions inputs based on the similarity-driven top-$k$ routing mechanism, allowing tokens from each patient sample to be shared across multiple inner and outer experts simultaneously. In addition to MIMIC-IV, we also evaluated our methods on the eICU dataset (Pollard et al., 2018), which covers over 139k patients admitted to ICUs across the United States between 2014 and 2015. We followed the experimental settings used by Wu et al. (2024). For predictive tasks, we tested our method on readmission prediction and mortality prediction, and included representative baselines: Oracle (trained directly on the target test data), Base (trained solely on the source training data), as well as

Table 3: We apply HMoE to multi-domain and multi-modal patient data. HMoE delivers customized predictions for each group, while effectively accounting for the interactions and uniqueness of each group. This approach greatly improves results compared to current state-of-the-art methods.

| Dataset | eICU | | | | MIMIC-IV | | | |
|---|---|---|---|---|---|---|---|---|
| Task | Readmission | | Mortality | | Readmission | | Mortality | |
| Metric | AUPRC | AUROC | AUPRC | AUROC | AUPRC | AUROC | AUPRC | AUROC |
| *Oracle* | *21.92 ± 0.15* | *67.72 ± 0.42* | *27.14 ± 0.06* | *83.87 ± 0.57* | *28.21 ± 0.34* | *69.31 ± 0.53* | *42.83 ± 0.48* | *89.82 ± 0.75* |
| Base | 10.41 ± 0.12 | 51.01 ± 0.31 | 23.02 ± 0.24 | 80.31 ± 0.43 | 23.70 ± 0.23 | 66.54 ± 0.41 | 37.40 ± 0.20 | 86.10 ± 0.64 |
| DANN | 13.50 ± 0.09 | 53.79 ± 0.19 | 24.47 ± 0.08 | 80.82 ± 0.27 | 24.68 ± 0..09 | 67.31 ± 0.33 | 38.01 ± 0.17 | 87.34 ± 0.39 |
| MLDG | 10.41 ± 0.07 | 52.54 ± 0.43 | 22.41 ± 0.12 | 79.73 ± 0.39 | 20.50 ± 0.14 | 63.72 ± 0.29 | 35.98 ± 0.31 | 85.72 ± 0.68 |
| IRM | 13.62 ± 0.13 | 53.78 ± 0.22 | 25.18 ± 0.09 | 80.09 ± 0.47 | 24.23 ± 0.21 | 66.80 ± 0.22 | 38.72 ± 0.19 | 87.59 ± 0.43 |
| SLDG | 18.57 ± 0.10 | 62.30 ± 0.46 | **26.79 ± 0.16** | **82.44 ± 0.19** | 27.41 ± 0.10 | 69.02 ± 0.40 | 41.56 ± 0.12 | **89.85 ± 0.59** |
| HMoE | **19.39 ± 0.05** | **63.61 ± 0.23** | 26.60 ± 0.08 | 81.92 ± 0.28 | 27.82 ± 0.24 | 69.13 ± 0.21 | 42.23 ± 0.32 | 89.47 ± 0.18 |
| HMoE-M | - | - | - | - | **27.97 ± 0.18** | **69.19 ± 0.26** | **42.47 ± 0.35** | 89.65 ± 0.13 |

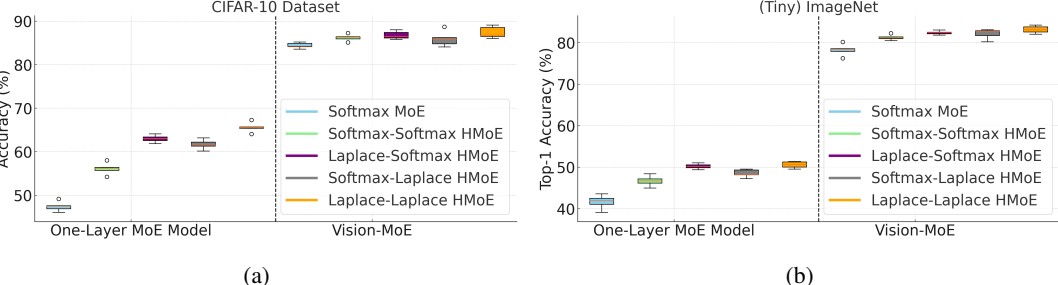

(a)                      (b)

Figure 2: We evaluate the impact of using different gating function combinations in HMoE and compare it with standard MoE on (a) CIFAR-10 and (b) ImageNet. First, we present the results of one-layer MoE models (left side of each figure), where the model contains only the module of that specific setting. For the one-layer results, we use Tiny-ImageNet as a substitute for the full ImageNet. Next, we integrate these MoE modules into the state-of-the-art Vision MoE model (right) (Ruiz et al., 2021) and compare the performance on the full datasets.

domain generalization methods that require domain IDs: DANN (Ganin et al., 2016) and MLDG (Li et al., 2018), and those that do not require IDs: IRM (Arjovsky et al., 2019). Table 3 presents the results for both datasets. Among all the tested methods, HMoE with the Softmax-Laplace gating combination achieved the best overall performance on both tasks. Given HMoE's advantage in processing multimodal inputs, we further added clinical notes and CXR modalities to the MIMIC-IV dataset (HMoE-M in Table 3), which led to additional performance improvements thanks to the joint benefit of customized modeling and the inclusion of extra modality information.

## 3.2 QUANTATITIVE ANALYSIS

**Combinations of Different Gating Mechanisms.** Figure 2 compares the performance of different gating function combinations on the commonly used CIFAR-10 and ImageNet datasets. We first evaluate a single module (i.e., a one-layer MoE model) on CIFAR-10 and Tiny-ImageNet, followed by integrating these modules into the Vision-MoE framework (Riquelme et al., 2021): in the Vision Transformer (ViT) models, we selectively replace an even number of FFN layers with targeted MoE layers and test the models on the full datasets. The performance gap between different gating functions is more pronounced in the one-layer MoE models due to the amplified effect of the module differences, while the difference becomes smaller after incorporating them into Vision MoE. The results show that the Laplace-Laplace gating combination achieves the best performance, while the combination of Laplace and Softmax gating also yields competitive results. Overall, HMoE demonstrates its potential to enhance the capacity of image classification models.

**Multimodal Routing Distributions.** We then analyze how modality tokens are distributed across different experts and groups. Figure 3 displays the distribution of three modality tokens in the best-performing HMoE block for corresponding tasks from the MIMIC-IV dataset. The HMoE module consists of two expert groups, each containing four experts. The results are taken from the final HMoE block of the trained model, using the first batch of data. Most vital signs and clinical notes tokens are routed to expert group 1, while CXR tokens are predominantly routed to expert group 2. For tasks (a) and (b), vital signs and clinical notes contribute more heavily to the overall HMoE

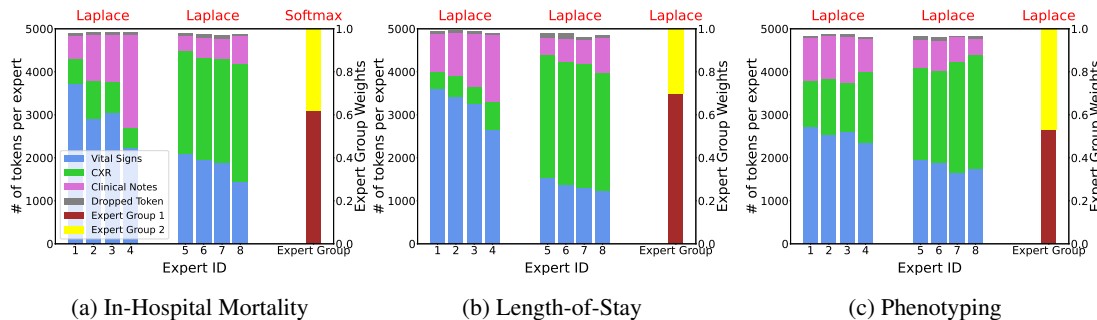

(a) In-Hospital Mortality  (b) Length-of-Stay  (c) Phenotyping

Figure 3: Token distribution (time series, CXR, clinical notes) of HMoE blocks of a multimodal transformer. We present the best-performing gating combinations for three tasks evaluated on MIMIC-IV, where the HMoE block comprises 2 outer expert groups, each containing 4 inner experts. Expert IDs 1 to 4 (left section of each figure) represent token distributions from expert group 1, and expert IDs 5 to 8 (middle section) represent token distributions from expert group 2. The right section shows the relative weights assigned to each expert group.

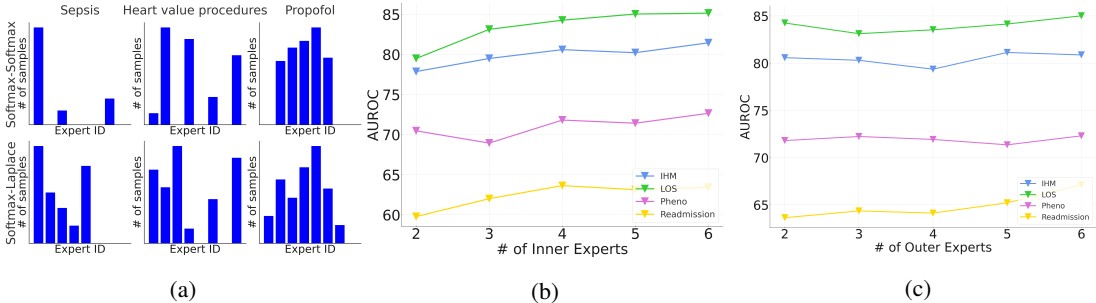

(a)  (b)  (c)

Figure 4: (a) Distribution of top clinical events across expert IDs under heterogeneous versus homogeneous gating functions. (b)/(c) Performance variations as the number of inner/outer experts increases.

prediction, particularly in task (b). However, for task (c), CXR tokens play a more significant role, contributing almost as much as vital signs, despite being present in smaller quantities. Additionally, due to the load-balancing loss applied during training, the total token count is nearly uniformly distributed among experts, with minimal token dropping because of exceeding capacity limits.

**Distribution of Clinical Events.** Given that the number of clinical event categories is much larger than the number of modalities, it is more intuitive to visualize the impact of different gating function combinations on the distribution of clinical events. Figure 4 (a) illustrates the routing distribution for the most commonly observed clinical events using the best-performing Softmax-Laplace gating combination of HMoE in latent domain discovery, compared to the Softmax gating function. The results indicate that the Softmax-Laplace combination promotes greater diversification in routing clinical event samples to experts while encouraging expert sharing across different categories. We further conduct ablation studies by varying the number of inner and outer experts in the best-performing HMoE across four tasks, as shown in Figure 4 (b) and (c), where their number of outer and inner experts is fixed at 2 and 4, respectively. The results demonstrate that increasing the number of experts has a positive impact on performance, particularly for inner experts, though this improvement comes with an increase in computational demands.

## 4  DISCUSSIONS AND LIMITATIONS

In this work, we explored diverse gating function combinations beyond Softmax in a two-level hierarchical mixture of experts (HMoE). Our theoretical analysis demonstrated that using Laplace gating in HMoE improves convergence behavior, and employing Laplace gating at both levels significantly optimizes performance. We validated this theoretical finding on multiple real-world tasks, while also showcasing the effectiveness of HMoE in handling complex inputs, such as multimodal and multidomain data. However, the enhanced ability to process complex inputs comes with increased computational demands, which is a key limitation of HMoE. For future work, we plan to explore techniques like pruning to reduce computational costs in large-scale multimodal tasks and to identify more suitable downstream applications for HMoE.

## REPRODUCIBILITY STATEMENT

To ensure the reproducibility of our empirical results, we provide comprehensive descriptions of the data, preprocessing steps, and implementation details in Appendices B, C, and D. Additionally, the code is included in the supplementary materials for submission. All datasets utilized in this study are publicly accessible online, though access to the MIMIC-IV and eICU datasets requires an additional approval process following their regulations.

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

# Supplement to "On Expert Estimation in Hierarchical Mixture of Experts: Beyond Softmax Gating Functions"

In this supplementary material, we first introduce some related works to this paper in Appendix A. The dataset information, preprocessing procedures, and implementation details can be found in Appendices B, C, and D, respectively. Next, we provide the proof for the convergence of expert estimation in Appendix E, while that for the convergence of density estimation is presented in Appendix F. Then, we continue to streamline the proof of Lemma 1 in Appendix G before investigating the identifiability of the Gaussian HMoE in Appendix H.

## A  RELATED WORKS

MoE (Jacobs et al., 1991; Xu et al., 1994) has gained significant popularity for managing complex tasks since its introduction three decades ago. Unlike traditional models that reuse the same parameters for all inputs, MoE selects distinct parameters for each specific input. This results in a sparsely activated layer, enabling a substantial scaling of model capacity without a corresponding increase in computational cost. Recent studies (Shazeer et al., 2017; Fedus et al., 2022; Mustafa et al., 2022; Zhou et al., 2023; Shen et al., 2023; Han et al., 2024) have demonstrated the effectiveness of integrating MoE with cutting-edge models across a diverse range of tasks. Nie et al. (2021); Zhou et al. (2022); Puigcerver et al. (2023) have also tackled key challenges such as accuracy and training instability. With the growing prevalence of MoE, the HMoE architecture has also been utilized to enhance model generalization performance in complex data structures. For instance, Ng & McLachlan (2007) leveraged HMoE to more effectively manage hierarchical data, thereby improving classification accuracy in medical datasets. Similarly, Peralta & Soto (2014) introduced regularized HMoE models with embedded local feature selection, which enhanced model performance in high-dimensional scenarios. Due to its ability to assign input partitions to specialized experts, HMoE is particularly well-suited for multi-modal or multi-domain applications (Zhao et al., 2021). Prior research has demonstrated that HMoE can ensure robust generalization capabilities (Azran & Meir, 2004). However, existing studies have primarily assessed HMoE in small-scale experiments and have not shown its effectiveness in large-scale real-world settings.

While MoE has been widely employed to scale up large models, its theoretical foundations have remained relatively underdeveloped. First of all, Mendes & Jiang (2011) studied the maximum likelihood estimator for parameters of the MoE with each expert being a polynomial regression model. In particular, they investigated the convergence rate of the estimated density to the true density under the Kullback-Leibler (KL) divergence and gave some insights on how many experts should be chosen. Next, Ho et al. (2022) conducted a similar convergence analysis for input-free gating Gaussian MoE but using the Hellinger distance for the density estimation problem instead of the KL divergence. Additionally, they utilized the generalized Wasserstein distance to capture the parameter estimation rates which were negatively affected by the algebraic interactions among parameters. Nguyen et al. (2023) then generalized these results to a more popular setting known as softmax gating Gaussian MoE. Rather than leveraging the generalized Wasserstein distance for the parameter estimation problem, they proposed novel Voronoi-based loss functions which were shown to characterize the parameter estimation rates more accurately. Recently, Han et al. (2024) advocated using a new Laplace gating function which induced faster convergence rates than the softmax gating functions due to a reduced number of parameter interactions. However, to the best of our knowledge, a comprehensive convergence analysis for HMoE has remained elusive in the literature.

## B  DATASET INFORMATION

### B.1  MIMIC-IV

MIMIC-IV (Johnson et al., 2020) is a comprehensive database containing records from nearly 300,000 patients admitted to a medical center between 2008 and 2019, focusing on a subset of 73,181 ICU stays. We linked core ICU records, including lab results and vital signs, with corresponding chest X-rays (Johnson et al., 2019b), radiological notes (Johnson et al., 2023), and electrocardiogram (ECG) data (Gow et al., 2022) recorded during the same ICU stay.

**Tasks of Interest.** We design an in-hospital mortality prediction task (referred to as **48-IHM**) to assess our method's capability in forecasting short-term patient deterioration. Additionally, accurately predicting patient discharge times is vital for improving patient outcomes and managing hospital resources efficiently Bertsimas et al. (2022), leading us to implement the length-of-stay (**LOS**) task. Both the 48-IHM and LOS tasks are framed as binary classification problems, utilizing a 48-hour observation window (for patients staying at least 48 hours in the ICU) to predict in-hospital mortality (48-IHM) and patient discharge (without death) within the subsequent 48 hours (LOS). Moreover, recognizing the presence of specific acute care conditions in patient records is key for several clinical goals, such as forming cohorts for studies and identifying comorbidities Agarwal et al. (2016). Traditional approaches, which often rely on manual chart reviews or billing codes, are increasingly being complemented by machine learning models Harutyunyan et al. (2019). Automating this process demands high-accuracy classifications, which drives the development of our 25-type phenotype classification (**25-PHE**) task. This multilabel classification problem involves predicting one of 25 acute care conditions using data from the entire ICU stay. We summarize the details of these tasks below:

- **48-IHM**: This is a binary classification task where we aim to predict in-hospital mortality based on data collected during the first 48 hours of ICU admission, applicable only to patients who remained in the ICU for at least 48 hours.

- **LOS**: The length-of-stay task is structured similarly to 48-IHM. For patients who stayed in the ICU for a minimum of 48 hours, the objective is to predict whether they will be discharged (without death) within the next 48 hours.

- **25-PHE**: This multilabel classification task involves predicting one of 25 acute care conditions Elixhauser (2009); Lovaasen & Schwerdtfeger (2012), such as congestive heart failure, pneumonia, or shock, at the conclusion of each patient's ICU stay. Since the original task was developed for diagnoses based on ICD-9 codes, and MIMIC-IV includes both ICD-9 and ICD-10 codes, we convert diagnoses coded in ICD-10 using the conversion database from Butler (2007).

**Evaluation.** We concentrated on patients with complete data across all modalities, which yielded a dataset of 8,770 ICU stays for the 48-IHM and LOS tasks, and 14,541 stays for the 25-PHE task. To assess the performance of the single-label tasks, 48-IHM and LOS, we utilize the F1-score and AUROC as our evaluation metrics. For the 25-PHE task, following prior research (Zhang et al., 2023; Lin et al., 2019; Arbabi et al., 2019), we rely on macro-averaged F1-score and AUROC as the primary measures of evaluation. For the multimodal fusion task, we allocated 70% data for training, while the remaining 30% was evenly divided between validation and testing. For clinical latent domain discovery, similar to Wu et al. (2024), we segment the dataset into four temporal groups: 2008-2010, 2011-2013, 2014-2016, and 2017-2019. Each group is then divided into training, validation, and testing sets, following a 70%, 10%, and 20% split, respectively. Patients admitted after 2014 are treated as the target test data, while all earlier patients are used as the source training data.

### B.2 EICU

The eICU dataset (Pollard et al., 2018) includes over 200,000 visits from 139,000 patients admitted to ICUs in 208 hospitals across the United States. The data was gathered between 2014 and 2015. The 208 hospitals are categorized into four regions based on their geographic location: Midwest, Northeast, West, and South. We define our cohorts by excluding visits from patients younger than 18 or older than 89, as well as visits exceeding 10 days in length or containing fewer than 3 or more than 256 timestamps. Additionally, we omit visits shorter than 12 hours, since predictions are made 12 hours post-admission.

**Tasks of Interest.** For the readmission task using the eICU dataset, our goal is to predict whether a patient will be readmitted within 15 days after discharge. Similar to the MIMIC-IV dataset, the mortality prediction task focuses on determining whether a patient will pass away following discharge.

**Evaluation.** The eICU dataset is divided into four regional groups: Midwest, Northeast, West, and South. Each region is further split into 70% for training, 10% for validation, and 20% for testing. To assess the performance gap between regions, we compare the backbone model's performance when trained on data from the same region versus data from other regions, as proposed by Wu et al. (2024).

The region with the largest performance gap (Midwest) is selected as the target test data, while the remaining regions (Northeast, West, and South) are used as the source training data. To compare with baselines from Wu et al. (2024), we use the same evaluation metrics: Area Under the Precision-Recall Curve (AUPRC) and the Area Under the Receiver Operating Characteristic Curve (AUROC) scores.

### B.3 Image Classification Datasets

**CIFAR-10.** CIFAR-10 Krizhevsky et al. (2009) is a well-known dataset in computer vision, commonly used for object recognition tasks. It contains 60,000 color images, each with a resolution of 32x32 pixels, representing one of 10 object categories ("plane," "car," "bird," "cat," "deer," "dog," "frog," "horse," "ship," "truck"), with 6,000 images per class.

**ImageNet.** We use the ImageNet database from ILSVRC2012 (Russakovsky et al., 2015), where the task is to classify images into 1,000 distinct categories, using a vast dataset of over 1.2 million training images and 150,000 validation and test images sourced from the ImageNet database.

**Tiny-ImageNet.** The Tiny-ImageNet is a smaller, more manageable subset of the ImageNet dataset. It contains 100,000 images and 200 classes selected from full ImageNet dataset. All images are resized to 64×64 pixels to reduce computational demands.

## C Data Preprocessing for Clinical Tasks

During preprocessing, we selected 30 relevant lab and chart events from each patient's ICU records to capture vital sign measurements. For chest X-rays, we employed a pre-trained DenseNet-121 model (Cohen et al., 2022), which had been fine-tuned on the CheXpert dataset (Irvin et al., 2019), to extract 1024-dimensional image embeddings. Additionally, we used the BioClinicalBERT model (Alsentzer et al., 2019) to generate 768-dimensional embeddings for the radiological notes.

**Time Series.** We selected 30 time-series events for analysis, as outlined in (Soenksen et al., 2022). This included nine vital signs: heart rate, mean/systolic/diastolic blood pressure, respiratory rate, oxygen saturation, and Glasgow Coma Scale (GCS) verbal, eye, and motor response. Additionally, 21 laboratory values were incorporated: potassium, sodium, chloride, creatinine, urea nitrogen, bicarbonate, anion gap, hemoglobin, hematocrit, magnesium, platelet count, phosphate, white blood cell count, total calcium, MCH, red blood cell count, MCHC, MCV, RDW, platelet count, neutrophil count, and vancomycin. Each time series value was standardized to have a mean of 0 and a standard deviation of 1, based on values from the training set. We use the Transformer as an encoder for time series data.

**Chest X-Rays.** To integrate medical imaging into our analysis, we use the MIMIC-CXR-JPG module (Johnson et al., 2019a) available through Physionet (Goldberger et al., 2000), which contains 377,110 JPG images derived from the DICOM-based MIMIC-CXR database (Johnson et al., 2019b). As described in Soenksen et al. (2022), each image is resized to $224 \times 224$ pixels, and we extract embeddings from the final layer of the DenseNet121 model. To identify X-rays taken during the patient's ICU stay, we match subject IDs from MIMIC-CXR-JPG with the core MIMIC-IV database and then filter the X-rays to those captured between the ICU admission and discharge times.

**Clinical Notes** To incorporate text data, we use the MIMIC-IV-Note module (Johnson et al., 2023), which includes 2,321,355 deidentified radiology reports for 237,427 patients. These reports can be linked to patients in the main MIMIC-IV dataset using a similar matching method as employed for chest X-rays. It is important to note that we were unable to access intermediate clinical notes (i.e., notes recorded by clinicians during the patient's stay), as they have not yet been made publicly available. We extract note embeddings using the Bio-Clinical BERT model (Alsentzer et al., 2019).

## D Implementation Details

### D.1 Model Architecture

Once embeddings from each input modality or domain are generated, we address the issue of irregularity in the data. To do this, we use a discretized multi-time attention (mTAND) module (Shukla & Marlin, 2021), which applies a time attention mechanism (Kazemi et al., 2019) to convert

---

**Algorithm 1** Computation Procedure for the 2-Level Hierarchical MoE Module

---

1: **Input**: $\mathbf{x} \in \mathbb{R}^{B \times N \times D}$; batch size $B$, sequence length $N$, embedding dimension $D$, number of outer/inner experts $E_o/E_i$, capacity per outer/inner expert $\mathcal{C}_o, \mathcal{C}_i$, dispatch tensor $\mathbf{D}$, combine tensor $\mathbf{C}$
2: $\mathbf{D}_o, \mathbf{C}_o, \mathbf{L}_o = \mathsf{Gate}_{\mathsf{outer}}(\mathbf{x})$ $\triangleright$ compute outer dispatch, outer combine tensors, and outer gating loss
3: $\mathbf{x}_{\mathsf{outer}}^{(e,b,c,d)} = \sum_n \mathbf{D}_o^{(b,n,e,c)} \cdot \mathbf{x}^{(b,n,d)}$ $\triangleright$ dispatch inputs to outer experts using dispatch tensor
4: $\mathbf{D}_i, \mathbf{C}_i, \mathbf{L}_i = \mathsf{Gate}_{\mathsf{inner}}(\mathbf{x}_{\mathsf{outer}})$ $\triangleright$ compute inner dispatch, inner combine tensors, and inner gating loss
5: $\mathbf{x}_{\mathsf{experts}}^{(e_o,e_i,b,c_i,d)} = \sum_{c_o} \mathbf{D}_i^{(e_o,b,c_o,e_i,c_i)} \cdot \mathbf{x}_{\mathsf{outer}}^{(e_o,b,c_o,d)}$ $\triangleright$ dispatch inputs to the inner experts
6: $\mathbf{y}_{\mathsf{experts}} = \mathsf{Experts}(\mathbf{x}_{\mathsf{experts}})$ $\triangleright$ expert processing
7: $\mathbf{y}_{\mathsf{outer}}^{(e_o,b,n,d)} = \sum_{e_i,c_i} \mathbf{C}_i^{(e_o,b,c_o,e_i,c_i)} \cdot \mathbf{y}_{\mathsf{experts}}^{(e_o,e_i,b,c_i,d)}$ $\triangleright$ combine inner expert outputs
8: $\mathbf{y}^{(b,n,d)} = \sum_{e,c} \mathbf{C}_o^{(b,n,e,c)} \cdot \mathbf{y}_{\mathsf{outer}}^{(e,b,c,d)}$ $\triangleright$ combine outer expert outputs
9: $\mathcal{L} = \lambda(\mathcal{L}_o + \mathcal{L}_i)$ $\triangleright$ compute total loss
10: **Return**: $\mathbf{y}, \mathcal{L}$

---

irregularly sampled observations into discrete time intervals. This approach has been employed in previous works such as (Zhang et al., 2023; Han et al., 2024). The mTAND module transforms the irregular sequences into fixed-length representations, which are then passed into the MoE fusion layer with a residual connection. This fusion layer comprises multi-head self-attention followed by the HMoE module. In total, there are 12 MoE fusion layers, and the output from this layer is optimized using task-specific loss and load imbalance loss. We apply a dropout rate of 0.1 and use the Adam optimizer with a learning rate of 1e-4 and a weight decay of 1e-5. All models are trained for 100 epochs. For the multimodal experiment, we use a batch size of 2, while for the latent domain discovery experiment, the batch size is set to 256.

### D.2 HMoE Module

The detailed implementation procedure of the two-level HMoE module of the MoE fusion layer can be found in Algorithm 1. We have also provided Python code as part of the supplementary material.

## E PROOFS FOR CONVERGENCE OF EXPERT ESTIMATION

*Proof of Theorems 2, 3 and 4.* **Overview.** We will focus on establishing the following inequality:

$$\inf_{G \in \mathcal{G}_{k_1^*,k_2}(\Theta)} \mathbb{E}_{\boldsymbol{X}}[h(p_G^{type}(\cdot|\boldsymbol{X}), p_{G_*}^{type}(\cdot|\boldsymbol{X}))]/\mathcal{L}_{(r_1,r_2,r_3)}(G, G_*) > 0,$$

where the value of $(r_1, r_2, r_3)$ varies with the variable $type \in \{SS, SL, LL\}$. Note that the Hellinger distance $h$ is lower bounded by the Total Variation distance $V$, that is, $h \geq V$, it suffices to demonstrate that

$$\inf_{G \in \mathcal{G}_{k_1^*,k_2}(\Theta)} \mathbb{E}_{\boldsymbol{X}}[V(p_G^{type}(\cdot|\boldsymbol{X}), p_{G_*}^{type}(\cdot|\boldsymbol{X}))]/\mathcal{L}_{(r_1,r_2,r_3)}(G, G_*) > 0. \tag{11}$$

To this end, we first show that

$$\lim_{\varepsilon \to 0} \inf_{G \in \mathcal{G}_{k_1^*,k_2}(\Theta): \mathcal{L}_{(r_1,r_2,r_3)}(G,G_*) \leq \varepsilon} \mathbb{E}_{\boldsymbol{X}}[V(p_G^{type}(\cdot|\boldsymbol{X}), p_{G_*}^{type}(\cdot|\boldsymbol{X}))]/\mathcal{L}_{(r_1,r_2,r_3)}(G, G_*) > 0. \tag{12}$$

The proof of this result will be presented later. Now, suppose that it holds true, then there exists a positive constant $\varepsilon'$ that satisfies

$$\inf_{G \in \mathcal{G}_{k_1^*,k_2}(\Theta): \mathcal{L}_1(G,G_*) \leq \varepsilon'} \mathbb{E}_{\boldsymbol{X}}[V(p_G^{type}(\cdot|\boldsymbol{X}), p_{G_*}^{type}(\cdot|\boldsymbol{X}))]/\mathcal{L}_{(r_1,r_2,r_3)}(G, G_*) > 0.$$

Thus, it suffices to establish the following inequality:

$$\inf_{G \in \mathcal{G}_{k_1^*,k_2}(\Theta): \mathcal{L}_1(G,G_*) > \varepsilon'} \mathbb{E}_{\boldsymbol{X}}[V(p_G^{type}(\cdot|\boldsymbol{X}), p_{G_*}^{type}(\cdot|\boldsymbol{X}))]/\mathcal{L}_{(r_1,r_2,r_3)}(G, G_*) > 0. \tag{13}$$

Assume by contrary that the inequality (13) does not hold true, then we can seek a sequence of mixing measures $G_n' \in \mathcal{G}_{k_1^*,k_2}(\Theta)$ that satisfy $\mathcal{L}_1(G_n', G_*) > \varepsilon'$ and

$$\lim_{n \to \infty} \mathbb{E}_{\boldsymbol{X}}[V(p_{G_n'}^{type}(\cdot|\boldsymbol{X}), p_{G_*}^{type}(\cdot|\boldsymbol{X}))]/\mathcal{L}_{(r_1,r_2,r_3)}(G_n', G_*) = 0.$$

Thus, we deduce that $\mathbb{E}_{\boldsymbol{X}}[V(p_{G'_n}^{type}(\cdot|\boldsymbol{X}), p_{G_*}^{type}(\cdot|\boldsymbol{X}))] \to 0$ as $n \to \infty$. Since $\Theta$ is a compact set, we can substitute the sequence $(G'_n)$ by one of its subsequences that converges to a mixing measure $G' \in \mathcal{G}_{k_1^*, k_2}(\Theta)$. Recall that $\mathcal{L}_{(r_1, r_2, r_3)}(G'_n, G_*) > \varepsilon'$, then we deduce that $\mathcal{L}_{(r_1, r_2, r_3)}(G', G_*) > \varepsilon'$. By employing the Fatou's lemma, it follows that

$$
\begin{aligned}
0 &= \lim_{n \to \infty} \mathbb{E}_{\boldsymbol{X}}[V(p_{G'_n}^{type}(\cdot|\boldsymbol{X}), p_{G_*}^{type}(\cdot|\boldsymbol{X}))]/\mathcal{L}_{(r_1, r_2, r_3)}(G'_n, G_*) \\
&\geq \frac{1}{2} \int \liminf_{n \to \infty} \left| p_{G'_n}^{type}(y|\boldsymbol{x}) - p_{G_*}^{type}(y|\boldsymbol{x}) \right|^2 \mathrm{d}(\boldsymbol{x}, y).
\end{aligned}
$$

Thus, we obtain that $p_{G'}^{type}(y|\boldsymbol{x}) = p_{G_*}^{type}(y|\boldsymbol{x})$ for almost surely $(\boldsymbol{x}, y)$. According to Proposition 1, we get that $G' \equiv G_*$, which yields that $\mathcal{L}_{(r_1, r_2, r_3)}(G', G_*) = 0$. This result contradicts the fact that $\mathcal{L}_{(r_1, r_2, r_3)}(G', G_*) > \varepsilon' > 0$. Hence, we obtain the result in equation (13), which together with the inequality (12) leads to the conclusion in equation (11).

Now, we are going back to the proof of the inequality (12).

**Proof of the inequality** (12) Suppose that the inequality (12) does not hold, then we can find a sequence of mixing measures $(G_n)$ in $\mathcal{G}_{k_1^*, k_2}(\Theta)$ that satisfies $\mathcal{L}_{(r_1, r_2, r_3)}(G_n, G_*) \to 0$ and

$$
\mathbb{E}_{\boldsymbol{X}}[V(p_{G_n}^{type}(\cdot|\boldsymbol{X}), p_{G_*}^{type}(\cdot|\boldsymbol{X}))]/\mathcal{L}_{(r_1, r_2, r_3)}(G_n, G_*) \to 0, \tag{14}
$$

as $n \to \infty$. For each $j_1 \in [k_1^*]$, let $\mathcal{V}_{j_1}^n := \mathcal{V}_{j_1}(G_n)$ be a Voronoi cell of $G_n$ generated by the $j_1$-th components of $G_*$. As the Voronoi loss $\mathcal{V}_{j_1}^n$ has only one element and our arguments are asymptotic, we may assume WLOG that $\mathcal{V}_{j_1}^n = \mathcal{V}_{j_1} = \{j_1\}$ for any $j_1 \in [k_1^*]$. Then, the Voronoi loss becomes

$$
\begin{aligned}
\mathcal{L}_{(r_1, r_2, r_3)}(G_n, G_*) &= \sum_{j_1=1}^{k_1^*} \left| \exp(b_{j_1}^n) - \exp(b_{j_1}^*) \right| + \sum_{j_1=1}^{k_1^*} \exp(b_{j_1}^n) \|\Delta \boldsymbol{a}_{j_1}^n\| + \sum_{j_1=1}^{k_1^*} \exp(b_{j_1}^n) \\
&\times \Bigg[ \sum_{j_2 : |\mathcal{V}_{j_2|j_1}|=1} \sum_{i_2 \in \mathcal{V}_{j_2|j_1}} \exp(\beta_{i_2|j_1}^n) \Big( \|\Delta \boldsymbol{\omega}_{i_2 j_2|j_1}^n\| + \|\Delta \boldsymbol{\eta}_{j_1 i_2 j_2}^n\| + |\Delta \tau_{j_1 i_2 j_2}^n| + |\Delta \nu_{j_1 i_2 j_2}^n| \Big) \\
&+ \sum_{j_2 : |\mathcal{V}_{j_2|j_1}|>1} \sum_{i_2 \in \mathcal{V}_{j_2|j_1}} \exp(\beta_{i_2|j_1}^n) \Big( \|\Delta \boldsymbol{\omega}_{i_2 j_2|j_1}^n\|^2 + \|\Delta \boldsymbol{\eta}_{j_1 i_2 j_2}^n\|^{r_1} + |\Delta \tau_{j_1 i_2 j_2}^n|^{r_2} \\
&+ |\Delta \nu_{j_1 i_2 j_2}^n|^{r_3} \Big) \Bigg] + \sum_{j_1=1}^{k_1^*} \exp(b_{j_1}^n) \sum_{j_2=1}^{k_2^*} \left| \sum_{i_2 \in \mathcal{V}_{j_2|j_1}} \exp(\beta_{i_2|j_1}^n) - \exp(\beta_{j_2|j_1}^*) \right|. \tag{15}
\end{aligned}
$$

Since $\mathcal{L}_{(r_1, r_2, r_3)}(G_n, G_*) \to 0$ as $n \to \infty$, it follows that $\exp(b_{j_1}^n) \to \exp(b_{j_1}^*)$, $\boldsymbol{a}_{j_1}^n \to \boldsymbol{a}_{j_1}^*$, $\exp(\beta_{i_2|j_1}^n) \to \exp(\beta_{j_2|j_1}^*)$, $\boldsymbol{\omega}_{i_2|j_1}^n \to \boldsymbol{\omega}_{j_2|j_1}^*$, $\boldsymbol{\eta}_{j_1 i_2}^n \to \boldsymbol{\eta}_{j_1 j_2}^*$, $\tau_{j_1 i_2}^n \to \tau_{j_1 j_2}^*$ and $\nu_{j_1 i_2}^n \to \nu_{j_1 j_2}^*$ for all $j_1 \in [k_1^*]$, $j_2 \in [k_2^*]$ and $i_2 \in \mathcal{V}_{j_2|j_1}$.

Subsequently, we consider three different settings where the variable $type$ takes the value in the set $\{SS, SL, LL\}$ in Appendices E.1, E.2 and E.3, respectively. In each appendix, the proof will be divided into three main stages.

### E.1 WHEN $type = SS$

When $type = SS$, the corresponding Voronoi loss function is $\mathcal{L}_{(\frac{1}{2}r^{SS}, r^{SS}, \frac{1}{2}r^{SS})}(G_n, G_*) = \mathcal{L}_{1n}$ where we define

$$
\mathcal{L}_{1n} := \sum_{j_1=1}^{k_1^*} \left| \exp(b_{j_1}^n) - \exp(b_{j_1}^*) \right| + \sum_{j_1=1}^{k_1^*} \exp(b_{j_1}^n) \|\Delta \boldsymbol{a}_{j_1}^n\| + \sum_{j_1=1}^{k_1^*} \exp(b_{j_1}^n)
$$

$$
\times \left[ \sum_{j_2 : |\mathcal{V}_{j_2|j_1}|=1} \sum_{i_2 \in \mathcal{V}_{j_2|j_1}} \exp(\beta_{i_2|j_1}^n) \Big( \|\Delta \boldsymbol{\omega}_{i_2 j_2|j_1}^n\| + \|\Delta \boldsymbol{\eta}_{j_1 i_2 j_2}^n\| + |\Delta \tau_{j_1 i_2 j_2}^n| + |\Delta \nu_{j_1 i_2 j_2}^n| \Big) \right.
$$

$$
+ \sum_{j_2 : |\mathcal{V}_{j_2|j_1}|>1} \sum_{i_2 \in \mathcal{V}_{j_2|j_1}} \exp(\beta_{i_2|j_1}^n) \Big( \|\Delta \boldsymbol{\omega}_{i_2 j_2|j_1}^n\|^2 + \|\Delta \boldsymbol{\eta}_{j_1 i_2 j_2}^n\|^{\frac{r_{j_2|j_1}^{SS}}{2}} + |\Delta \tau_{j_1 i_2 j_2}^n|^{r_{j_2|j_1}^{SS}}
$$

$$
\left. + |\Delta \nu_{j_1 i_2 j_2}^n|^{\frac{r_{j_2|j_1}^{SS}}{2}} \Big) \right] + \sum_{j_1=1}^{k_1^*} \exp(b_{j_1}^n) \sum_{j_2=1}^{k_2^*} \left| \sum_{i_2 \in \mathcal{V}_{j_2|j_1}} \exp(\beta_{i_2|j_1}^n) - \exp(\beta_{j_2|j_1}^*) \right|. \tag{16}
$$

**Step 1 - Taylor expansion:** In this stage, we aim to decompose the term

$$
Q_n := \left[ \sum_{j_1=1}^{k_1^*} \exp((\boldsymbol{a}_{j_1}^*)^\top \boldsymbol{x} + b_{j_1}^*) \right] [p_{G_n}^{SS}(y|\boldsymbol{x}) - p_{G_*}^{SS}(y|\boldsymbol{x})]
$$

into a combination of linearly independent terms using the Taylor expansion. For that purpose, let us denote

$$
p_{j_1}^{SS,n}(y|\boldsymbol{x}) := \sum_{j_2=1}^{k_2^*} \sum_{i_2 \in \mathcal{V}_{j_2|j_1}} \sigma((\boldsymbol{\omega}_{i_2|j_1}^n)^\top \boldsymbol{x} + \beta_{i_2|j_1}^n) \pi(y|(\boldsymbol{\eta}_{j_1 i_2}^n)^\top \boldsymbol{x} + \tau_{j_1 i_2}^n, \nu_{j_1 i_2}^n),
$$

$$
p_{j_1}^{SS,*}(y|\boldsymbol{x}) := \sum_{j_2=1}^{k_2^*} \sigma((\boldsymbol{\omega}_{j_2|j_1}^*)^\top \boldsymbol{x} + \beta_{j_2|j_1}^*) \pi(y|(\boldsymbol{\eta}_{j_1 j_2}^*)^\top \boldsymbol{x} + \tau_{j_1 j_2}^*, \nu_{j_1 j_2}^*).
$$

Then, it can be checked that the quantity $Q_n$ is divided as

$$
Q_n = \sum_{j_1=1}^{k_1^*} \exp(b_{j_1}^n) \left[ \exp((\boldsymbol{a}_{j_1}^n)^\top \boldsymbol{x}) p_{j_1}^{SS,n}(y|\boldsymbol{x}) - \exp((\boldsymbol{a}_{j_1}^*)^\top \boldsymbol{x}) p_{j_1}^{SS,*}(y|\boldsymbol{x}) \right]
$$

$$
- \sum_{j_1=1}^{k_1^*} \exp(b_{j_1}^n) \left[ \exp((\boldsymbol{a}_{j_1}^n)^\top \boldsymbol{x}) - \exp((\boldsymbol{a}_{j_1}^*)^\top \boldsymbol{x}) \right] p_{G_n}^{SS}(y|\boldsymbol{x})
$$

$$
+ \sum_{j_1=1}^{k_1^*} \left( \exp(b_{j_1}^n) - \exp(b_{j_1}^*) \right) \exp((\boldsymbol{a}_{j_1}^*)^\top \boldsymbol{x}) \left[ p_{j_1}^{SS,n}(y|\boldsymbol{x}) - p_{G_n}^{SS}(y|\boldsymbol{x}) \right]
$$

$$
:= A_n - B_n + C_n. \tag{17}
$$

**Step 1A - Decompose $A_n$:** Using the same techniques for decomposing $Q_n$, we can decompose $A_n$ as follows:

$$
A_n := \sum_{j_1=1}^{k_1^*} \frac{\exp(b_{j_1}^n)}{\sum_{j_2'=1}^{k_2^*} \exp((\boldsymbol{\omega}_{j_2'|j_1}^*)^\top \boldsymbol{x} + \beta_{j_2'|j_1}^*)} [A_{n,j_1,1} + A_{n,j_1,2} + A_{n,j_1,3}],
$$

where

$$A_{n,j_1,1} := \sum_{j_2=1}^{k_2^*} \sum_{i_2 \in \mathcal{V}_{j_2|j_1}} \exp(\beta_{i_2|j_1}^n) \Big[ \exp((\boldsymbol{\omega}_{i_2|j_1}^n)^\top \boldsymbol{x}) \exp((\boldsymbol{a}_{j_1}^n)^\top \boldsymbol{x}) \pi(y|(\boldsymbol{\eta}_{j_1 i_2}^n)\top \boldsymbol{x} + \tau_{j_1 i_2}^n, \nu_{j_1 i_2}^n)$$

$$- \exp((\boldsymbol{\omega}_{j_2|j_1}^*)^\top \boldsymbol{x}) \exp((\boldsymbol{a}_{j_1}^*)^\top \boldsymbol{x}) \pi(y|(\boldsymbol{\eta}_{j_1 j_2}^*)^\top \boldsymbol{x} + \tau_{j_1 j_2}^*, \nu_{j_1 j_2}^*) \Big],$$

$$A_{n,j_1,2} := \sum_{j_2=1}^{k_2^*} \sum_{i_2 \in \mathcal{V}_{j_2|j_1}} \exp(\beta_{i_2|j_1}^n) \Big[ \exp((\boldsymbol{\omega}_{i_2|j_1}^n)^\top \boldsymbol{x}) - \exp((\boldsymbol{\omega}_{j_2|j_1}^*)^\top \boldsymbol{x}) \Big]$$

$$\times \exp((\boldsymbol{a}_{j_1}^n)^\top \boldsymbol{x}) p_{j_1}^{SS,n}(y|\boldsymbol{x}),$$

$$A_{n,j_1,3} := \sum_{j_2=1}^{k_2^*} \Big( \sum_{i_2 \in \mathcal{V}_{j_2|j_1}} \exp(\beta_{i_2|j_1}^n) - \exp(\beta_{j_2|j_1}^*) \Big) \exp((\boldsymbol{\omega}_{j_2|j_1}^*)^\top \boldsymbol{x})$$

$$\times [\exp((\boldsymbol{a}_{j_1}^*)^\top \boldsymbol{x}) \pi(y|(\boldsymbol{\eta}_{j_1 j_2}^*)^\top \boldsymbol{x} + \tau_{j_1 j_2}^*, \nu_{j_1 j_2}^*) - \exp((\boldsymbol{a}_{j_1}^n)^\top \boldsymbol{x}) p_{j_1}^{SS,n}(y|\boldsymbol{x})].$$

Based on the cardinality of the Voronoi cells $\mathcal{V}_{j_2|j_1}$, we continue to divide the term $A_{n,j_1,1}$ into two parts as

$$A_{n,j_1,1} = \sum_{j_2:|\mathcal{V}_{j_2|j_1}|=1} \sum_{i_2 \in \mathcal{V}_{j_2|j_1}} \exp(\beta_{i_2|j_1}^n) \Big[ \exp((\boldsymbol{\omega}_{i_2|j_1}^n)^\top \boldsymbol{x}) \exp((\boldsymbol{a}_{j_1}^n)^\top \boldsymbol{x}) \pi(y|(\boldsymbol{\eta}_{j_1 i_2}^n)\top \boldsymbol{x} + \tau_{j_1 i_2}^n, \nu_{j_1 i_2}^n)$$

$$- \exp((\boldsymbol{\omega}_{j_2|j_1}^*)^\top \boldsymbol{x}) \exp((\boldsymbol{a}_{j_1}^*)^\top \boldsymbol{x}) \pi(y|(\boldsymbol{\eta}_{j_1 j_2}^*)^\top \boldsymbol{x} + \tau_{j_1 j_2}^*, \nu_{j_1 j_2}^*) \Big],$$

$$+ \sum_{j_2:|\mathcal{V}_{j_2|j_1}|>1} \sum_{i_2 \in \mathcal{V}_{j_2|j_1}} \exp(\beta_{i_2|j_1}^n) \Big[ \exp((\boldsymbol{\omega}_{i_2|j_1}^n)^\top \boldsymbol{x}) \exp((\boldsymbol{a}_{j_1}^n)^\top \boldsymbol{x}) \pi(y|(\boldsymbol{\eta}_{j_1 i_2}^n)\top \boldsymbol{x} + \tau_{j_1 i_2}^n, \nu_{j_1 i_2}^n)$$

$$- \exp((\boldsymbol{\omega}_{j_2|j_1}^*)^\top \boldsymbol{x}) \exp((\boldsymbol{a}_{j_1}^*)^\top \boldsymbol{x}) \pi(y|(\boldsymbol{\eta}_{j_1 j_2}^*)^\top \boldsymbol{x} + \tau_{j_1 j_2}^*, \nu_{j_1 j_2}^*) \Big]$$

$$:= A_{n,j_1,1,1} + A_{n,j_1,1,2}.$$

Let $\xi(\boldsymbol{\eta}, \tau) = \boldsymbol{\eta}^\top \boldsymbol{x} + \tau$. By applying the first-order Taylor expansion, the term $A_{n,j_1,1,1}$ can be rewritten as

$$A_{n,j_1,1,1} = \sum_{j_2:|\mathcal{V}_{j_2|j_1}|=1} \sum_{i_2 \in \mathcal{V}_{j_2|j_1}} \sum_{|\boldsymbol{\alpha}|=1} \frac{\exp(\beta_{i_2|j_1}^n)}{2^{\alpha_5} \boldsymbol{\alpha}!} (\Delta \boldsymbol{\omega}_{i_2 j_2|j_1}^n)^{\boldsymbol{\alpha}_1} (\Delta \boldsymbol{a}_{j_1}^n)^{\boldsymbol{\alpha}_2} (\Delta \boldsymbol{\eta}_{j_1 i_2 j_2}^n)^{\boldsymbol{\alpha}_3} (\Delta \tau_{j_1 i_2 j_2}^n)^{\alpha_4}$$

$$\times (\Delta \nu_{j_1 i_2 j_2}^n)^{\alpha_5} \boldsymbol{x}^{\boldsymbol{\alpha}_1 + \boldsymbol{\alpha}_2 + \boldsymbol{\alpha}_3} \exp((\boldsymbol{\omega}_{j_2|j_1}^*)^\top \boldsymbol{x}) \exp((\boldsymbol{a}_{j_1}^*)^\top \boldsymbol{x}) \frac{\partial^{|\boldsymbol{\alpha}_3| + \alpha_4 + 2\alpha_5} \pi}{\partial \xi^{|\boldsymbol{\alpha}_3| + \alpha_4 + 2\alpha_5}} (y|(\boldsymbol{\eta}_{j_1 j_2}^*)^\top \boldsymbol{x} + \tau_{j_1 j_2}^*, \nu_{j_1 j_2}^*)$$

$$+ R_{n,1,1}(\boldsymbol{x})$$

$$= \sum_{j_2:|\mathcal{V}_{j_2|j_1}|=1} \sum_{|\boldsymbol{\rho}_1| + \rho_2 = 1}^{2} S_{n,j_2|j_1,\boldsymbol{\rho}_1,\rho_2} \cdot \boldsymbol{x}^{\boldsymbol{\rho}_1} \cdot \exp((\boldsymbol{\omega}_{j_2|j_1}^*)^\top \boldsymbol{x}) \exp((\boldsymbol{a}_{j_1}^*)^\top \boldsymbol{x})$$

$$\times \frac{\partial^{\rho_2} \pi}{\partial \xi^{\rho_2}} (y|(\boldsymbol{\eta}_{j_1 j_2}^*)^\top \boldsymbol{x} + \tau_{j_1 j_2}^*, \nu_{j_1 j_2}^*) + R_{n,1,1}(\boldsymbol{x}),$$

where $R_{n,1,1}(\boldsymbol{x})$ is a Taylor remainder satisfying $R_{n,1,1}(\boldsymbol{x})/\mathcal{L}_{1n} \to 0$ as $n \to \infty$, and

$$S_{n,j_2|j_1,\boldsymbol{\rho}_1,\rho_2} := \sum_{i_2 \in \mathcal{V}_{j_2|j_1}} \sum_{(\boldsymbol{\alpha}_1, \boldsymbol{\alpha}_2, \boldsymbol{\alpha}_3, \alpha_4, \alpha_5) \in \mathcal{I}_{\boldsymbol{\rho}_1, \rho_2}^{SS}} \frac{\exp(\beta_{i_2|j_1}^n)}{2^{\alpha_5} \boldsymbol{\alpha}!} (\Delta \boldsymbol{\omega}_{i_2 j_2|j_1}^n)^{\boldsymbol{\alpha}_1} (\Delta \boldsymbol{a}_{j_1}^n)^{\boldsymbol{\alpha}_2} (\Delta \boldsymbol{\eta}_{j_1 i_2 j_2}^n)^{\boldsymbol{\alpha}_3}$$

$$\times (\Delta \tau_{j_1 i_2 j_2}^n)^{\alpha_4} (\Delta \nu_{j_1 i_2 j_2}^n)^{\alpha_5},$$

for any $(\boldsymbol{\rho}_1, \rho_2) \neq (\boldsymbol{0}_d, 0)$ and $j_1 \in [k_1^*], j_2 \in [k_2^*]$ in which

$$\mathcal{I}_{\boldsymbol{\rho}_1, \rho_2}^{SS} := \{(\boldsymbol{\alpha}_1, \boldsymbol{\alpha}_2, \boldsymbol{\alpha}_3, \alpha_4, \alpha_5) \in \mathbb{R}^d \times \mathbb{R}^d \times \mathbb{R}^d \times \mathbb{R} : \boldsymbol{\alpha}_1 + \boldsymbol{\alpha}_2 + \boldsymbol{\alpha}_3 = \boldsymbol{\rho}_1, |\boldsymbol{\alpha}_3| + \alpha_4 + 2\alpha_5 = \rho_2\}.$$

For each $(j_1, j_2) \in [k_1^*] \times [k_2^*]$, by applying the Taylor expansion of order $r^{SS}(|\mathcal{V}_{j_2|j_1}|) := r_{j_2|j_1}^{SS}$, we can represent the term $A_{n,j_1,1,2}$ as

$$A_{n,j_1,1,2} = \sum_{j_2:|\mathcal{V}_{j_2|j_1}|>1} \sum_{|\boldsymbol{\rho}_1|+\boldsymbol{\rho}_2=1}^{2r_{j_2|j_1}^{SS}} S_{n,j_2|j_1,\boldsymbol{\rho}_1,\boldsymbol{\rho}_2} \cdot \boldsymbol{x}^{\boldsymbol{\rho}_1} \cdot \exp((\boldsymbol{\omega}_{j_2|j_1}^*)^\top \boldsymbol{x}) \exp((\boldsymbol{a}_{j_1}^*)^\top \boldsymbol{x})$$

$$\times \frac{\partial^{\rho_2}\pi}{\partial \xi^{\rho_2}}(y|(\boldsymbol{\eta}_{j_1 j_2}^*)^\top \boldsymbol{x} + \tau_{j_1 j_2}^*, \nu_{j_1 j_2}^*) + R_{n,1,2}(\boldsymbol{x}),$$

where $R_{n,1,2}(\boldsymbol{x})$ is a Taylor remainder such that $R_{n,1,2}(\boldsymbol{x})/\mathcal{L}_{1n} \to 0$ as $n \to \infty$.

Subsequently, we rewrite the term $A_{n,j_1,2}$ as follows:

$$\sum_{j_2:|\mathcal{V}_{j_2|j_1}|=1} \sum_{i_2 \in \mathcal{V}_{j_2|j_1}} \exp(\beta_{i_2|j_1}^n)\Big[\exp((\boldsymbol{\omega}_{i_2|j_1}^n)^\top \boldsymbol{x}) - \exp((\boldsymbol{\omega}_{j_2|j_1}^*)^\top \boldsymbol{x})\Big] \exp((\boldsymbol{a}_{j_1}^n)^\top \boldsymbol{x}) p_{j_1}^{SS,n}(y|\boldsymbol{x})$$

$$+ \sum_{j_2:|\mathcal{V}_{j_2|j_1}|>1} \sum_{i_2 \in \mathcal{V}_{j_2|j_1}} \exp(\beta_{i_2|j_1}^n)\Big[\exp((\boldsymbol{\omega}_{i_2|j_1}^n)^\top \boldsymbol{x}) - \exp((\boldsymbol{\omega}_{j_2|j_1}^*)^\top \boldsymbol{x})\Big] \exp((\boldsymbol{a}_{j_1}^n)^\top \boldsymbol{x}) p_{j_1}^{SS,n}(y|\boldsymbol{x})$$

$$:= A_{n,j_1,2,1} + A_{n,j_1,2,2}.$$

By means of the first-order Taylor expansion, we have

$$A_{n,j_1,2,1} = \sum_{j_2:|\mathcal{V}_{j_2|j_1}|=1} \sum_{i_2 \in \mathcal{V}_{j_2|j_1}} \sum_{|\boldsymbol{\psi}|=1} \frac{\exp(\beta_{i_2|j_1}^n)}{\boldsymbol{\psi}!}(\Delta\boldsymbol{\omega}_{i_2 j_2|j_1}^n)^{\boldsymbol{\psi}}$$

$$\times \boldsymbol{x}^{\boldsymbol{\psi}} \exp((\boldsymbol{\omega}_{j_2|j_1}^*)^\top \boldsymbol{x}) \exp((\boldsymbol{a}_{j_1}^n)^\top \boldsymbol{x}) p_{j_1}^{SS,n}(y|\boldsymbol{x}) + R_{n,2,1}(\boldsymbol{x}),$$

$$= \sum_{j_2:|\mathcal{V}_{j_2|j_1}|=1} \sum_{|\boldsymbol{\psi}|=1} T_{n,j_2|j_1,\boldsymbol{\psi}} \cdot \boldsymbol{x}^{\boldsymbol{\psi}} \exp((\boldsymbol{\omega}_{j_2|j_1}^*)^\top \boldsymbol{x}) \exp((\boldsymbol{a}_{j_1}^n)^\top \boldsymbol{x}) p_{j_1}^{SS,n}(y|\boldsymbol{x}) + R_{n,2,1}(\boldsymbol{x}),$$

where $R_{n,2,1}(\boldsymbol{x})$ is a Taylor remainder such that $R_{n,2,1}(\boldsymbol{x})/\mathcal{L}_{1n} \to 0$ as $n \to \infty$, and

$$T_{n,j_2|j_1,\boldsymbol{\psi}} := \sum_{i_2 \in \mathcal{V}_{j_2|j_1}} \frac{\exp(\beta_{i_2|j_1}^n)}{\boldsymbol{\psi}!}(\Delta\boldsymbol{\omega}_{i_2 j_2|j_1}^n)^{\boldsymbol{\psi}},$$

for any $j_2 \in [k_2^*]$ and $\boldsymbol{\psi} \neq \mathbf{0}_d$.

At the same time, we apply the second-order Taylor expansion to $A_{n,j_1,2,2}$:

$$A_{n,j_1,2,2} = \sum_{j_2:|\mathcal{V}_{j_2|j_1}|>1} \sum_{|\boldsymbol{\psi}|=1}^{2} T_{n,j_2|j_1,\boldsymbol{\psi}} \cdot \boldsymbol{x}^{\boldsymbol{\psi}} \exp((\boldsymbol{\omega}_{j_2|j_1}^*)^\top \boldsymbol{x}) \exp((\boldsymbol{a}_{j_1}^n)^\top \boldsymbol{x}) p_{j_1}^{SS,n}(y|\boldsymbol{x}) + R_{n,2,2}(\boldsymbol{x}),$$

where $R_{n,2,2}(\boldsymbol{x})$ is a Taylor remainder such that $R_{n,2,2}(\boldsymbol{x})/\mathcal{L}_{1n} \to 0$ as $n \to \infty$.

As a result, the term $A_n$ can be rewritten as

$$A_n = \sum_{j_1=1}^{k_1^*} \sum_{j_2=1}^{k_2^*} \frac{\exp(b_{j_1}^n)}{\sum_{j_2'=1}^{k_2^*} \exp((\boldsymbol{\omega}_{j_2'|j_1}^*)^\top \boldsymbol{x} + \beta_{j_2'|j_1}^*)} \Bigg[ \sum_{|\boldsymbol{\rho}_1|+\boldsymbol{\rho}_2=1}^{2r_{j_2|j_1}^{SS}} S_{n,j_2|j_1,\boldsymbol{\rho}_1,\boldsymbol{\rho}_2} \cdot \boldsymbol{x}^{\boldsymbol{\rho}_1} \cdot \exp((\boldsymbol{\omega}_{j_2|j_1}^*)^\top \boldsymbol{x})$$

$$\times \exp((\boldsymbol{a}_{j_1}^*)^\top \boldsymbol{x}) \frac{\partial^{\rho_2}\pi}{\partial \xi^{\rho_2}}(y|(\boldsymbol{\eta}_{j_1 j_2}^*)^\top \boldsymbol{x} + \tau_{j_1 j_2}^*, \nu_{j_1 j_2}^*) + R_{n,1,1}(\boldsymbol{x}) + R_{n,1,2}(\boldsymbol{x})$$

$$- \sum_{|\boldsymbol{\psi}|=0}^{2} T_{n,j_2|j_1,\boldsymbol{\psi}} \cdot \boldsymbol{x}^{\boldsymbol{\psi}} \exp((\boldsymbol{\omega}_{j_2|j_1}^*)^\top \boldsymbol{x}) \exp((\boldsymbol{a}_{j_1}^n)^\top \boldsymbol{x}) p_{j_1}^{SS,n}(y|\boldsymbol{x}) - R_{n,2,1}(\boldsymbol{x}) - R_{n,2,2}(\boldsymbol{x}) \Bigg], \tag{18}$$

where $S_{n,j_2|j_1,\boldsymbol{\rho}_1,\boldsymbol{\rho}_2} = T_{n,j_2|j_1,\boldsymbol{\psi}} = \sum_{i_2 \in \mathcal{V}_{j_2|j_1}} \exp(\beta_{i_2|j_1}^n) - \exp(\beta_{j_2|j_1}^*)$ for any $j_2 \in [k_2^*]$ where $(\boldsymbol{\alpha}_1, \boldsymbol{\rho}_1, \rho_2) = (\mathbf{0}_d, \mathbf{0}_d, 0)$ and $\boldsymbol{\psi} = \mathbf{0}_d$.

**Step 1B - Decompose $B_n$:** By invoking the first-order Taylor expansion, the term $B_n$ defined in equation (17) can be rewritten as

$$B_n = \sum_{j_1=1}^{k_1^*} \exp(b_{j_1}^n) \sum_{|\boldsymbol{\gamma}|=1} (\Delta \boldsymbol{a}_{j_1}^n)^{\boldsymbol{\gamma}} \cdot \boldsymbol{x}^{\boldsymbol{\gamma}} \exp((\boldsymbol{a}_{j_1}^*)^\top \boldsymbol{x}) p_{G_n}^{SS}(y|\boldsymbol{x}) + R_{n,3}(\boldsymbol{x}), \qquad (19)$$

where $R_{n,3}(\boldsymbol{x})$ is a Taylor remainder such that $R_{n,3}(\boldsymbol{x})/\mathcal{L}_{1n} \to 0$ as $n \to \infty$.

From the decomposition in equations (17), (18) and (19), we realize that $A_n$, $B_n$ and $C_n$ can be viewed as a combination of elements from the following set union:

$$\left\{ \boldsymbol{x}^{\boldsymbol{\rho}_1} \cdot \exp((\boldsymbol{\omega}_{j_2|j_1}^*)^\top \boldsymbol{x}) \exp((\boldsymbol{a}_{j_1}^*)^\top \boldsymbol{x}) \frac{\partial^{\rho_2} \pi}{\partial \xi^{\rho_2}} (y|(\boldsymbol{\eta}_{j_1 j_2}^*)^\top \boldsymbol{x} + \tau_{j_1 j_2}^*, \nu_{j_1 j_2}^*) : j_1 \in [k_1^*],\ j_2 \in [k_2^*], \right.$$

$$\left. 0 \le |\boldsymbol{\rho}_1| + \rho_2 \le 2 r_{j_2|j_1}^{SS} \right\}$$

$$\cup \left\{ \frac{\boldsymbol{x}^{\boldsymbol{\psi}} \exp((\boldsymbol{\omega}_{j_2|j_1}^*)^\top \boldsymbol{x}) \exp((\boldsymbol{a}_{j_1}^n)^\top \boldsymbol{x}) p_{j_1}^{SS,n}(y|\boldsymbol{x})}{\sum_{j_2'=1}^{k_2^*} \exp((\boldsymbol{\omega}_{j_2'|j_1}^*)^\top \boldsymbol{x} + \beta_{j_2'|j_1}^*)} : j_1 \in [k_1^*],\ j_2 \in [k_2^*],\ 0 \le |\boldsymbol{\psi}| \le 2 \right\}$$

$$\cup \left\{ \boldsymbol{x}^{\boldsymbol{\gamma}} \exp((\boldsymbol{a}_{j_1}^*)^\top \boldsymbol{x}) p_{j_1}^{SS,n}(y|\boldsymbol{x}),\ \boldsymbol{x}^{\boldsymbol{\gamma}} \exp((\boldsymbol{a}_{j_1}^*)^\top \boldsymbol{x}) p_{G_n}^{SS}(y|\boldsymbol{x}) : j_1 \in [k_1^*],\ 0 \le |\boldsymbol{\gamma}| \le 1 \right\}.$$

**Step 2 - Non-vanishing coefficients:** In this stage, we show that not all the coefficients in the representation of $A_n/\mathcal{L}_{1n}$, $B_n/\mathcal{L}_{1n}$ and $C_n/\mathcal{L}_{1n}$ go to zero as $n \to \infty$. Assume that all of them approach zero, then by looking into the coefficients associated with the term

- $\exp((\boldsymbol{a}_{j_1}^*)^\top \boldsymbol{x}) p_{j_1}^{SS,n}(y|\boldsymbol{x})$ in $C_n/\mathcal{L}_{1n}$, we have

$$\frac{1}{\mathcal{L}_{1n}} \cdot \sum_{j_1=1}^{k_1^*} \left| \exp(b_{j_1}^n) - \exp(b_{j_1}^*) \right| \to 0. \qquad (20)$$

- $\dfrac{\exp((\boldsymbol{\omega}_{j_2|j_1}^*)^\top \boldsymbol{x}) \exp((\boldsymbol{a}_{j_1}^*)^\top \boldsymbol{x}) \pi(y|(\boldsymbol{\eta}_{j_1 j_2}^*)^\top \boldsymbol{x} + \tau_{j_1 j_2}^*, \nu_{j_1 j_2}^*)}{\sum_{j_2'=1}^{k_2^*} \exp((\boldsymbol{\omega}_{j_2'|j_1}^*)^\top \boldsymbol{x} + \beta_{j_2'|j_1}^*)}$ in $A_n/\mathcal{L}_{1n}$, we get that

$$\frac{1}{\mathcal{L}_{1n}} \cdot \sum_{j_1=1}^{k_1^*} \exp(b_{j_1}^n) \sum_{j_2=1}^{k_2^*} \left| \sum_{i_2 \in \mathcal{V}_{j_2|j_1}} \exp(\beta_{i_2|j_1}^n) - \exp(\beta_{j_2|j_1}^*) \right| \to 0. \qquad (21)$$

- $\dfrac{\boldsymbol{x}^{\boldsymbol{\psi}} \exp((\boldsymbol{\omega}_{j_2|j_1}^*)^\top \boldsymbol{x}) \exp((\boldsymbol{a}_{j_1}^n)^\top \boldsymbol{x}) p_{j_1}^{SS,n}(y|\boldsymbol{x})}{\sum_{j_2'=1}^{k_2^*} \exp((\boldsymbol{\omega}_{j_2'|j_1}^*)^\top \boldsymbol{x} + \beta_{j_2'|j_1}^*)}$ in $A_n/\mathcal{L}_{1n}$ for $j_1 \in [k_1^*], j_2 \in [k_2^*]$ : $|\mathcal{V}_{j_2|j_1}| = 1$ and $\boldsymbol{\psi} = e_{d,u}$ where $e_{d,u} := (0,\ldots,0,\underbrace{1}_{u\text{-}th},0,\ldots,0) \in \mathbb{N}^d$, we receive

$$\frac{1}{\mathcal{L}_{1n}} \cdot \sum_{j_1=1}^{k_1^*} \exp(b_{j_1}^n) \sum_{j_2 \in [k_2^*]:|\mathcal{V}_{j_2|j_1}|=1} \sum_{i_2 \in \mathcal{V}_{j_2|j_1}} \exp(\beta_{i_2|j_1}^n) \| \boldsymbol{\omega}_{i_2|j_1}^n - \boldsymbol{\omega}_{j_2|j_1}^* \|_1 \to 0.$$

Note that since the norm-1 is equivalent to the norm-2, then we can replace the norm-1 with the norm-2, that is,

$$\frac{1}{\mathcal{L}_{1n}} \cdot \sum_{j_1=1}^{k_1^*} \exp(b_{j_1}^n) \sum_{j_2 \in [k_2^*]:|\mathcal{V}_{j_2|j_1}|=1} \sum_{i_2 \in \mathcal{V}_{j_2|j_1}} \exp(\beta_{i_2|j_1}^n) \| \boldsymbol{\omega}_{i_2|j_1}^n - \boldsymbol{\omega}_{j_2|j_1}^* \| \to 0. \qquad (22)$$

- $\dfrac{\exp((\boldsymbol{\omega}^*_{j_2|j_1})^\top \boldsymbol{x})\exp((\boldsymbol{a}^*_{j_1})^\top \boldsymbol{x})\frac{\partial^{\rho_2}\pi}{\partial \xi^{\rho_2}}(y|(\boldsymbol{\eta}^*_{j_1 j_2})^\top \boldsymbol{x} + \tau^*_{j_1 j_2}, \nu^*_{j_1 j_2})}{\sum_{j'_2=1}^{k^*_2}\exp((\boldsymbol{\omega}^*_{j'_2|j_1})^\top \boldsymbol{x} + \beta^*_{j'_2|j_1})}$ in $A_n/\mathcal{L}_{1n}$ for $j_1 \in [k^*_1], j_2 \in [k^*_2]: |\mathcal{V}_{j_2|j_1}| = 1$ and $\rho_2 = 1$, we have that

$$\frac{1}{\mathcal{L}_{1n}} \cdot \sum_{j_1=1}^{k^*_1} \exp(b^n_{j_1}) \sum_{j_2 \in [k^*_2]:|\mathcal{V}_{j_2|j_1}|=1} \exp(\beta^n_{j_2|j_1})|\tau^n_{j_1 j_2} - \tau^*_{j_1 j_2}| \to 0. \tag{23}$$

- $\dfrac{\boldsymbol{x}^{\boldsymbol{\rho}_1}\exp((\boldsymbol{\omega}^*_{j_2|j_1})^\top \boldsymbol{x})\exp((\boldsymbol{a}^*_{j_1})^\top \boldsymbol{x})\frac{\partial^{\rho_2}\pi}{\partial \xi^{\rho_2}}(y|(\boldsymbol{\eta}^*_{j_1 j_2})^\top \boldsymbol{x} + \tau^*_{j_1 j_2}, \nu^*_{j_1 j_2})}{\sum_{j'_2=1}^{k^*_2}\exp((\boldsymbol{\omega}^*_{j'_2|j_1})^\top \boldsymbol{x} + \beta^*_{j'_2|j_1})}$ in $A_n/\mathcal{L}_{1n}$ for $j_1 \in [k^*_1], j_2 \in [k^*_2]: |\mathcal{V}_{j_2|j_1}| = 1, \boldsymbol{\rho}_1 = e_{d,u}$ and $\rho_2 = 1$, we have that

$$\frac{1}{\mathcal{L}_{1n}} \cdot \sum_{j_1=1}^{k^*_1} \exp(b^n_{j_1}) \sum_{j_2 \in [k^*_2]:|\mathcal{V}_{j_2|j_1}|=1} \sum_{i_2 \in \mathcal{V}_{j_2|j_1}} \exp(\beta^n_{j_2|j_1})\|\boldsymbol{\eta}^n_{j_1 i_2} - \boldsymbol{\eta}^*_{j_1 j_2}\| \to 0. \tag{24}$$

- $\dfrac{\exp((\boldsymbol{\omega}^*_{j_2|j_1})^\top \boldsymbol{x})\exp((\boldsymbol{a}^*_{j_1})^\top \boldsymbol{x})\frac{\partial^{\rho_2}\pi}{\partial \xi^{\rho_2}}(y|(\boldsymbol{\eta}^*_{j_1 j_2})^\top \boldsymbol{x} + \tau^*_{j_1 j_2}, \nu^*_{j_1 j_2})}{\sum_{j'_2=1}^{k^*_2}\exp((\boldsymbol{\omega}^*_{j'_2|j_1})^\top \boldsymbol{x} + \beta^*_{j'_2|j_1})}$ in $A_n/\mathcal{L}_{1n}$ for $j_1 \in [k^*_1], j_2 \in [k^*_2]: |\mathcal{V}_{j_2|j_1}| = 1$ and $\rho_2 = 2$, we have that

$$\frac{1}{\mathcal{L}_{1n}} \cdot \sum_{j_1=1}^{k^*_1} \exp(b^n_{j_1}) \sum_{j_2 \in [k^*_2]:|\mathcal{V}_{j_2|j_1}|=1} \exp(\beta^n_{j_2|j_1})|\nu^n_{j_1 j_2} - \nu^*_{j_1 j_2}| \to 0. \tag{25}$$

- $\boldsymbol{x}^{\boldsymbol{\gamma}}\exp((\boldsymbol{a}^*_{j_1})^\top \boldsymbol{x})p^{SS}_{G_n}(y|\boldsymbol{x})$ in $B_n/\mathcal{L}_{1n}$ for $j_1 \in [k^*_1]$ and $\boldsymbol{\gamma} = e_{d,u}$, we obtain

$$\frac{1}{\mathcal{L}_{1n}} \cdot \sum_{j_1=1}^{k^*_1} \exp(b^n_{j_1})\|\boldsymbol{a}^n_{j_1} - \boldsymbol{a}^*_{j_1}\| \to 0. \tag{26}$$

- $\dfrac{\boldsymbol{x}^{\boldsymbol{\psi}}\exp((\boldsymbol{\omega}^*_{j_2|j_1})^\top \boldsymbol{x})\exp((\boldsymbol{a}^n_{j_1})^\top \boldsymbol{x})p^{SS,n}_{j_1}(y|\boldsymbol{x})}{\sum_{j'_2=1}^{k^*_2}\exp((\boldsymbol{\omega}^*_{j'_2|j_1})^\top \boldsymbol{x} + \beta^*_{j'_2|j_1})}$ in $A_n/\mathcal{L}_{1n}$ for $j_1 \in [k^*_1], j_2 \in [k^*_2]: |\mathcal{V}_{j_2|j_1}| > 1$ and $\boldsymbol{\psi} = 2e_{d,u}$, we receive that

$$\frac{1}{\mathcal{L}_{1n}} \cdot \sum_{j_1=1}^{k^*_1} \exp(b^n_{j_1}) \sum_{j_2 \in [k^*_2]:|\mathcal{V}_{j_2|j_1}|>1} \sum_{i_2 \in \mathcal{V}_{j_2|j_1}} \exp(\beta^n_{i_2|j_1})\|\boldsymbol{\omega}^n_{i_2|j_1} - \boldsymbol{\omega}^*_{j_2|j_1}\|^2 \to 0. \tag{27}$$

Combine the above limits together with the loss $\mathcal{L}_{1n}$ in equation (16), it yields that

$$\frac{1}{\mathcal{L}_{1n}} \cdot \sum_{j_1=1}^{k^*_1} \exp(b^n_{j_1})\Bigg[\sum_{j_2:|\mathcal{V}_{j_2|j_1}|>1} \sum_{i_2 \in \mathcal{V}_{j_2|j_1}} \exp(\beta^n_{i_2|j_1})\Big(\|\Delta\boldsymbol{\eta}^n_{j_1 i_2 j_2}\|^{\frac{r^{SS}_{j_2|j_1}}{2}} + |\Delta\tau^n_{j_1 i_2 j_2}|^{r^{SS}_{j_2|j_1}}$$
$$+ |\Delta\nu^n_{j_1 i_2 j_2}|^{\frac{r^{SS}_{j_2|j_1}}{2}}\Big)\Bigg] \not\to 0,$$

which indicates that

$$\frac{1}{\mathcal{L}_{1n}} \cdot \sum_{j_1=1}^{k^*_1} \exp(b^n_{j_1})\Bigg[\sum_{j_2:|\mathcal{V}_{j_2|j_1}|>1} \sum_{i_2 \in \mathcal{V}_{j_2|j_1}} \exp(\beta^n_{i_2|j_1})\Big(\|\Delta\boldsymbol{\omega}^n_{i_2 j_2|j_1}\|^{r^{SS}_{j_2|j_1}} + \|\Delta\boldsymbol{a}^n_{j_1}\|^{r^{SS}_{j_2|j_1}}$$
$$+ \|\Delta\boldsymbol{\eta}^n_{j_1 i_2 j_2}\|^{\frac{r^{SS}_{j_2|j_1}}{2}} + |\Delta\tau^n_{j_1 i_2 j_2}|^{r^{SS}_{j_2|j_1}} + |\Delta\nu^n_{j_1 i_2 j_2}|^{\frac{r^{SS}_{j_2|j_1}}{2}}\Big)\Bigg] \not\to 0,$$

as $n \to \infty$. Therefore, there exist indices $j_1^* \in [k_1^*]$ and $j_2^* \in [k_2^*] : |\mathcal{V}_{j_2^*|j_1^*}| > 1$ such that

$$\frac{1}{\mathcal{L}_{1n}} \cdot \sum_{i_2 \in \mathcal{V}_{j_2^*|j_1^*}} \exp(\beta_{i_2|j_1^*}^n) \Big( \|\boldsymbol{\omega}_{i_2|j_1^*}^n - \boldsymbol{\omega}_{j_2^*|j_1^*}^*\|^{r_{j_2^*|j_1^*}^{SS}} + \|\boldsymbol{a}_{j_1^*}^n - \boldsymbol{a}_{j_1^*}^*\|^{r_{j_2^*|j_1^*}^{SS}} + \|\boldsymbol{\eta}_{j_1^* i_2}^n - \boldsymbol{\eta}_{j_1^* j_2^*}^*\|^{\frac{r_{j_2^*|j_1^*}^{SS}}{2}}$$

$$+ |\tau_{j_1^* i_2}^n - \tau_{j_1^* j_2^*}^*|^{r_{j_2^*|j_1^*}^{SS}} + |\nu_{j_1^* i_2}^n - \nu_{j_1^* j_2^*}^*|^{\frac{r_{j_2^*|j_1^*}^{SS}}{2}} \Big) \not\to 0. \tag{28}$$

WLOG, we may assume that $j_1^* = j_2^* = 1$. By examining the coefficients of the terms
$$\frac{\boldsymbol{x}^{\boldsymbol{\rho}_1} \exp((\boldsymbol{\omega}_{j_2|j_1}^*)^\top \boldsymbol{x}) \exp((\boldsymbol{a}_{j_1}^*)^\top \boldsymbol{x}) \frac{\partial^{\rho_2} \pi}{\partial \xi^{\rho_2}}(y|(\boldsymbol{\eta}_{j_1 j_2}^*)^\top \boldsymbol{x} + \tau_{j_1 j_2}^*, \nu_{j_1 j_2}^*)}{\sum_{j_2'=1}^{k_2^*} \exp((\boldsymbol{\omega}_{j_2'|j_1}^*)^\top \boldsymbol{x} + \beta_{j_2'|j_1}^*)} \text{ in } A_n/\mathcal{L}_{1n} \text{ for } j_1 = j_2 = 1,$$
we have $\exp(b_1^n) S_{n,1|1,\boldsymbol{0}_d,\boldsymbol{\rho}_1,\rho_2}/\mathcal{L}_{1n} \to 0$, or equivalently,

$$\frac{1}{\mathcal{L}_{1n}} \cdot \sum_{i_2 \in \mathcal{V}_{1|1}} \sum_{(\boldsymbol{\alpha}_1, \boldsymbol{\alpha}_2, \boldsymbol{\alpha}_3, \alpha_4, \alpha_5) \in \mathcal{I}_{\boldsymbol{\rho}_1, \rho_2}^{SS}} \frac{\exp(\beta_{i_2|1}^n)}{2^{\alpha_5} \boldsymbol{\alpha}!} \cdot (\Delta \boldsymbol{\omega}_{1 i_2 1}^n)^{\boldsymbol{\alpha}_1} (\Delta \boldsymbol{a}_1^n)^{\boldsymbol{\alpha}_2} (\Delta \boldsymbol{\eta}_{1 i_2 1}^n)^{\boldsymbol{\alpha}_3}$$

$$\times (\Delta \tau_{1 i_2 1}^n)^{\alpha_4} (\Delta \nu_{1 i_2 1}^n)^{\alpha_5} \to 0. \tag{29}$$

By dividing the left hand side of equation (29) by that of equation (28), we get

$$\frac{\sum_{i_2 \in \mathcal{V}_{1|1}} \sum_{(\boldsymbol{\alpha}_1, \boldsymbol{\alpha}_2, \boldsymbol{\alpha}_3, \alpha_4, \alpha_5) \in \mathcal{I}_{\boldsymbol{\rho}_1, \rho_2}^{SS}} \frac{\exp(\beta_{i_2|1}^n)}{2^{\alpha_5} \boldsymbol{\alpha}!} \cdot (\Delta \boldsymbol{\omega}_{1 i_2 1}^n)^{\boldsymbol{\alpha}_1} (\Delta \boldsymbol{a}_1^n)^{\boldsymbol{\alpha}_2} (\Delta \boldsymbol{\eta}_{1 i_2 1}^n)^{\boldsymbol{\alpha}_3} (\Delta \tau_{1 i_2 1}^n)^{\alpha_4} (\Delta \nu_{1 i_2 1}^n)^{\alpha_5}}{\sum_{i_2 \in \mathcal{V}_{1|1}} \exp(\beta_{i_2|1}^n) \Big( \|\Delta \boldsymbol{\omega}_{1 i_2 1}^n\|^{r_{1|1}^{SS}} + \|\Delta \boldsymbol{a}_1^n\|^{r_{1|1}^{SS}} + \|\Delta \boldsymbol{\eta}_{1 i_2 i}^n\|^{\frac{r_{1|1}^{SS}}{2}} + |\Delta \tau_{1 i_2 1}^n|^{r_{1|1}^{SS}} + |\Delta \nu_{1 i_2 1}^n|^{\frac{r_{1|1}^{SS}}{2}} \Big)} \to 0. \tag{30}$$

Let us define $\overline{M}_n := \max\{\|\Delta \boldsymbol{\omega}_{1 i_2 1}^n\|, \|\Delta \boldsymbol{a}_1^n\|, \|\Delta \boldsymbol{\eta}_{1 i_2 1}^n\|^{1/2}, \|\Delta \tau_{1 i_2 1}^n\|, \|\Delta \nu_{1 i_2 1}^n\|^{1/2} : i_2 \in \mathcal{V}_{1|1}\}$, and $\overline{\beta}_n := \max_{i_2 \in \mathcal{V}_{1|1}} \exp(\beta_{i_2|1}^n)$. Since the sequence $\exp(\beta_{i_2|1}^n)/\overline{\beta}_n$ is bounded, we can replace it by its subsequence which has a positive limit $p_{i_2}^2 := \lim_{n \to \infty} \exp(\beta_{i_2|1}^n)/\overline{\beta}_n$. Note that at least one among the limits $p_{i_2}^2$ must be equal to one. Next, let us define

$$(\Delta \boldsymbol{\omega}_{1 i_2 1}^n)/\overline{M} \to \boldsymbol{q}_{1 i_2} \quad (\Delta \boldsymbol{a}_1^n)/\overline{M}_n \to \boldsymbol{q}_2, \qquad (\Delta \boldsymbol{\eta}_{1 i_2 1}^n)/\overline{M}_n \to \boldsymbol{q}_{3 i_2},$$
$$(\Delta \tau_{1 i_2 1}^n)/\overline{M}_n \to q_{4 i_2}, \quad (\Delta \nu_{1 i_2 1}^n)/2\overline{M}_n \to q_{5 i_2} \qquad \qquad .$$

Note that at least one among $\boldsymbol{q}_{1 i_2}, \boldsymbol{q}_2, \boldsymbol{q}_{3 i_2}, q_{4 i_2}, q_{5 i_2}$ must be equal to either 1 or $-1$.

By dividing both the numerator and the denominator of the term in equation (30) by $\overline{\beta}_n \overline{M}_n^{|\boldsymbol{\rho}_1| + \rho_2}$, we obtain the system of polynomial equations:

$$\sum_{i_2 \in \mathcal{V}_{1|1}} \sum_{(\boldsymbol{\alpha}_1, \boldsymbol{\alpha}_2, \boldsymbol{\alpha}_3, \alpha_4, \alpha_5) \in \mathcal{I}_{\boldsymbol{\rho}_1, \rho_2}^{SS}} \frac{1}{\boldsymbol{\alpha}!} \cdot p_{i_2}^2 \boldsymbol{q}_{1 i_2}^{\boldsymbol{\alpha}_1} \boldsymbol{q}_2^{\boldsymbol{\alpha}_2} \boldsymbol{q}_{3 i_2}^{\boldsymbol{\alpha}_3} q_{4 i_2}^{\alpha_4} q_{5 i_2}^{\alpha_5} = 0, \quad 1 \leq |\boldsymbol{\rho}_1| + \rho_2 \leq r_{1|1}^{SS}.$$

According to the definition of the term $r_{1|1}^{SS}$, the above system does not have any non-trivial solutions, which is a contradiction. Consequently, at least one among the coefficients in the representation of $A_n/\mathcal{L}_{1n}$, $B_n/\mathcal{L}_{1n}$ and $C_n/\mathcal{L}_{1n}$ must not converge to zero as $n \to \infty$.

**Step 3 - Application of the Fatou's lemma.** In this stage, we show that all the coefficients in the formulations of $A_n/\mathcal{L}_{1n}$, $B_n/\mathcal{L}_{1n}$ and $C_n/\mathcal{L}_{1n}$ go to zero as $n \to \infty$. Denote by $m_n$ the maximum of the absolute values of those coefficients, the result from Step 2 induces that $1/m_n \not\to \infty$. By employing the Fatou's lemma, we have

$$0 = \lim_{n \to \infty} \frac{\mathbb{E}_{\boldsymbol{X}}[V(p_{G_n}^{SS}(\cdot|\boldsymbol{X}), p_{G_*}^{SS}(\cdot|\boldsymbol{X}))]}{m_n \mathcal{L}_{1n}} \geq \int \liminf_{n \to \infty} \frac{|p_{G_n}^{SS}(y|\boldsymbol{x}) - p_{G_*}^{SS}(y|\boldsymbol{x})|}{2 m_n \mathcal{L}_{1n}} \mathrm{d}(\boldsymbol{x}, y).$$

Thus, we deduce that

$$\frac{|p_{G_n}^{SS}(y|\boldsymbol{x}) - p_{G_*}^{SS}(y|\boldsymbol{x})|}{2 m_n \mathcal{L}_{1n}} \to 0,$$

which results in $Q_n/[m_n \mathcal{L}_{1n}] \to 0$ as $n \to \infty$ for almost surely $(\boldsymbol{x}, y)$.

Next, we denote

$$\frac{\exp(b_{j_1}^n)S_{n,j_2|j_1,\boldsymbol{\rho}_1,\rho_2}}{m_n\mathcal{L}_{1n}} \to \phi_{j_2|j_1,\boldsymbol{\rho}_1,\rho_2}, \qquad\qquad \frac{\exp(b_{j_1}^n)T_{n,j_2|j_1,\boldsymbol{\psi}}}{m_n\mathcal{L}_{1n}} \to \varphi_{j_2|j_1,\boldsymbol{\psi}},$$

$$\frac{\exp(b_{j_1}^n)(\Delta \boldsymbol{a}_{j_1}^n)^{\boldsymbol{\gamma}}}{m_n\mathcal{L}_{1n}} \to \lambda_{j_1,\boldsymbol{\gamma}}, \qquad\qquad \frac{\exp(b_{j_1}^n) - \exp(b_{j_1}^*)}{m_n\mathcal{L}_{1n}} \to \chi_{j_1}$$

with a note that at least one among them is non-zero. Then, the decomposition of $Q_n$ in equation (17) indicates that

$$\lim_{n\to\infty}\frac{Q_n}{m_n\mathcal{L}_{1n}} = \lim_{n\to\infty}\frac{A_n}{m_n\mathcal{L}_{1n}} - \lim_{n\to\infty}\frac{B_n}{m_n\mathcal{L}_{1n}} + \lim_{n\to\infty}\frac{C_n}{m_n\mathcal{L}_{1n}},$$

in which

$$\lim_{n\to\infty}\frac{A_n}{m_n\mathcal{L}_{1n}} = \sum_{j_1=1}^{k_1^*}\sum_{j_2=1}^{k_2^*}\left[\sum_{|\boldsymbol{\rho}_1|+\rho_2=0}^{2r_{j_2|j_1}^{SS}} S_{n,j_2|j_1,\boldsymbol{\rho}_1,\rho_2} \cdot \boldsymbol{x}^{\boldsymbol{\rho}_1}\exp((\boldsymbol{\omega}_{j_2|j_1}^*)^\top \boldsymbol{x})\right.$$

$$\times \exp((\boldsymbol{a}_{j_1}^*)^\top\boldsymbol{x})\frac{\partial^{\rho_2}\pi}{\partial\xi^{\rho_2}}(y|(\boldsymbol{\eta}_{j_1j_2}^*)^\top\boldsymbol{x}+\tau_{j_1j_2}^*, \nu_{j_1j_2}^*) - \sum_{|\boldsymbol{\psi}|=0}^{2}\varphi_{j_2|j_1,\boldsymbol{\psi}}\cdot\boldsymbol{x}^{\boldsymbol{\psi}}\exp((\boldsymbol{\omega}_{j_2|j_1}^*)^\top\boldsymbol{x})$$

$$\left.\times \exp((\boldsymbol{a}_{j_1}^*)^\top\boldsymbol{x})p_{j_1}^{SS,*}(y|\boldsymbol{x})\right]\frac{1}{\sum_{j_2'=1}^{k_2^*}\exp((\boldsymbol{\omega}_{j_2'|j_1}^*)^\top\boldsymbol{x}+\beta_{j_2'|j_1}^*)},$$

$$\lim_{n\to\infty}\frac{B_n}{m_n\mathcal{L}_{1n}} = \sum_{j_1=1}^{k_1^*}\sum_{|\boldsymbol{\gamma}|=1}\lambda_{j_1,\boldsymbol{\gamma}}\cdot\boldsymbol{x}^{\boldsymbol{\gamma}}\exp((\boldsymbol{a}_{j_1}^*)^\top\boldsymbol{x})p_{G_*}^{SS}(y|\boldsymbol{x}),$$

$$\lim_{n\to\infty}\frac{C_n(\boldsymbol{x})}{m_n\mathcal{L}_{1n}} = \sum_{j_1=1}^{k_1^*}\chi_{j_1}\exp((\boldsymbol{a}_{j_1}^*)^\top\boldsymbol{x})\left[p_{j_1}^{SS,*}(y|\boldsymbol{x}) - p_{G_*}^{SS}(y|\boldsymbol{x})\right].$$

Since the set

$$\left\{\frac{\boldsymbol{x}^{\boldsymbol{\rho}_1}\exp((\boldsymbol{\omega}_{j_2|j_1}^*)^\top\boldsymbol{x})\exp((\boldsymbol{a}_{j_1}^*)^\top\boldsymbol{x})\frac{\partial^{\rho_2}\pi}{\partial\xi^{\rho_2}}(y|(\boldsymbol{\eta}_{j_1j_2}^*)^\top\boldsymbol{x}+\tau_{j_1j_2}^*,\nu_{j_1j_2}^*)}{\sum_{j_2'=1}^{k_2^*}\exp((\boldsymbol{\omega}_{j_2'|j_1}^*)^\top\boldsymbol{x}+\beta_{j_2'|j_1}^*)} : j_1\in[k_1^*], j_2\in[k_2^*],\right.$$

$$\left. 0\le|\boldsymbol{\rho}_1|+\rho_2\le 2r_{j_2|j_1}^{SS}\right\}$$

$$\cup\left\{\frac{\boldsymbol{x}^{\boldsymbol{\psi}}\exp((\boldsymbol{\omega}_{j_2|j_1}^*)^\top\boldsymbol{x})\exp((\boldsymbol{a}_{j_1}^*)^\top\boldsymbol{x})p_{j_1}^{SS,*}(y|\boldsymbol{x})}{\sum_{j_2'=1}^{k_2^*}\exp((\boldsymbol{\omega}_{j_2'|j_1}^*)^\top\boldsymbol{x}+\beta_{j_2'|j_1}^*)} : j_1\in[k_1^*], j_2\in[k_2^*], 0\le|\boldsymbol{\psi}|\le 2\right\}$$

$$\cup\left\{\boldsymbol{x}^{\boldsymbol{\gamma}}\exp((\boldsymbol{a}_{j_1}^*)^\top\boldsymbol{x})p_{G_*}^{SS}(y|\boldsymbol{x}),\ \exp((\boldsymbol{a}_{j_1}^*)^\top\boldsymbol{x})p_{j_1}^{SS,*}(y|\boldsymbol{x}),\ \exp((\boldsymbol{a}_{j_1}^*)^\top\boldsymbol{x})p_{G_*}^{SS}(y|\boldsymbol{x})\right.$$

$$\left. : j_1\in[k_1^*], 0\le|\boldsymbol{\gamma}|\le 2\right\}$$

is linearly independent, we obtain that $\phi_{j_2|j_1,\boldsymbol{\rho}_1,\rho_2} = \varphi_{j_2|j_1,\boldsymbol{\psi}} = \lambda_{j_1,\boldsymbol{\gamma}} = \chi_{j_1} = 0$ for all $j_1\in[k_1^*]$, $j_2\in[k_2^*]$, $0\le|\boldsymbol{\rho}_1|+\rho_2\le 2r_{j_2|j_1}^{SS}$, $0\le|\boldsymbol{\psi}|\le 2$ and $0\le|\boldsymbol{\gamma}|\le 1$, which is a contradiction. As a consequence, we obtain the inequality in equation (12). Hence, the proof is completed.

## E.2 WHEN $type = SL$

When $type = SL$, the corresponding Voronoi loss function is $\mathcal{L}_{(\frac{1}{2}r^{SL}, r^{SL}, \frac{1}{2}r^{SL})}(G_n, G_*) = \mathcal{L}_{2n}$ where we define

$$
\mathcal{L}_{2n} := \sum_{j_1=1}^{k_1^*} \left| \exp(b_{j_1}^n) - \exp(b_{j_1}^*) \right| + \sum_{j_1=1}^{k_1^*} \exp(b_{j_1}^n) \|\Delta \boldsymbol{a}_{j_1}^n\| + \sum_{j_1=1}^{k_1^*} \exp(b_{j_1}^n)
$$

$$
\times \left[ \sum_{j_2: |\mathcal{V}_{j_2|j_1}|=1} \sum_{i_2 \in \mathcal{V}_{j_2|j_1}} \exp(\beta_{i_2|j_1}^n) \left( \|\Delta \boldsymbol{\omega}_{i_2 j_2|j_1}^n\| + \|\Delta \boldsymbol{\eta}_{j_1 i_2 j_2}^n\| + |\Delta \tau_{j_1 i_2 j_2}^n| + |\Delta \nu_{j_1 i_2 j_2}^n| \right) \right.
$$

$$
+ \sum_{j_2: |\mathcal{V}_{j_2|j_1}|>1} \sum_{i_2 \in \mathcal{V}_{j_2|j_1}} \exp(\beta_{i_2|j_1}^n) \left( \|\Delta \boldsymbol{\omega}_{i_2 j_2|j_1}^n\|^2 + \|\Delta \boldsymbol{\eta}_{j_1 i_2 j_2}^n\|^{\frac{r_{j_2|j_1}^{SL}}{2}} + |\Delta \tau_{j_1 i_2 j_2}^n|^{r_{j_2|j_1}^{SL}} \right.
$$

$$
\left. \left. + |\Delta \nu_{j_1 i_2 j_2}^n|^{\frac{r_{j_2|j_1}^{SL}}{2}} \right) \right] + \sum_{j_1=1}^{k_1^*} \exp(b_{j_1}^n) \sum_{j_2=1}^{k_2^*} \left| \sum_{i_2 \in \mathcal{V}_{j_2|j_1}} \exp(\beta_{i_2|j_1}^n) - \exp(\beta_{j_2|j_1}^*) \right|. \tag{31}
$$

**Step 1 - Taylor expansion:** In this step, we use the Taylor expansion to decompose the term

$$
Q_n := \left[ \sum_{j_1=1}^{k_1^*} \exp((\boldsymbol{a}_{j_1}^*)^\top \boldsymbol{x} + b_{j_1}^*) \right] [p_{G_n}^{SL}(y|\boldsymbol{x}) - p_{G_*}^{SL}(y|\boldsymbol{x})].
$$

Prior to that, let us denote

$$
p_{j_1}^{SL,n}(y|\boldsymbol{x}) := \sum_{j_2=1}^{k_2^*} \sum_{i_2 \in \mathcal{V}_{j_2|j_1}} \sigma(-\|\boldsymbol{\omega}_{i_2|j_1}^n - \boldsymbol{x}\| + \beta_{i_2|j_1}^n) \pi(y|(\boldsymbol{\eta}_{j_1 i_2}^n)^\top \boldsymbol{x} + \tau_{j_1 i_2}^n, \nu_{j_1 i_2}^n),
$$

$$
p_{j_1}^{SL,*}(y|\boldsymbol{x}) := \sum_{j_2=1}^{k_2^*} \sigma(-\|\boldsymbol{\omega}_{j_2|j_1}^* - \boldsymbol{x}\| + \beta_{j_2|j_1}^*) \pi(y|(\boldsymbol{\eta}_{j_1 j_2}^*)^\top \boldsymbol{x} + \tau_{j_1 j_2}^*, \nu_{j_1 j_2}^*).
$$

Then, the quantity $Q_n$ is divided into three terms as

$$
Q_n = \sum_{j_1=1}^{k_1^*} \exp(b_{j_1}^n) \left[ \exp((\boldsymbol{a}_{j_1}^n)^\top \boldsymbol{x}) p_{j_1}^{SL,n}(y|\boldsymbol{x}) - \exp((\boldsymbol{a}_{j_1}^*)^\top \boldsymbol{x}) p_{j_1}^{SL,*}(y|\boldsymbol{x}) \right]
$$

$$
- \sum_{j_1=1}^{k_1^*} \exp(b_{j_1}^n) \left[ \exp((\boldsymbol{a}_{j_1}^n)^\top \boldsymbol{x}) - \exp((\boldsymbol{a}_{j_1}^*)^\top \boldsymbol{x}) \right] p_{G_n}^{SL}(y|\boldsymbol{x})
$$

$$
+ \sum_{j_1=1}^{k_1^*} \left( \exp(b_{j_1}^n) - \exp(b_{j_1}^*) \right) \exp((\boldsymbol{a}_{j_1}^*)^\top \boldsymbol{x}) \left[ p_{j_1}^{SL,n}(y|\boldsymbol{x}) - p_{G_n}^{SL}(y|\boldsymbol{x}) \right]
$$

$$
:= A_n - B_n + C_n. \tag{32}
$$

**Step 1A - Decompose $A_n$:** We continue to decompose $A_n$:

$$
A_n := \sum_{j_1=1}^{k_1^*} \frac{\exp(b_{j_1}^n)}{\sum_{j_2'=1}^{k_2^*} \exp(-\|\boldsymbol{\omega}_{j_2'|j_1}^* - \boldsymbol{x}\| + \beta_{j_2'|j_1}^*)} [A_{n,j_1,1} + A_{n,j_1,2} + A_{n,j_1,3}],
$$

in which

$$A_{n,j_1,1} := \sum_{j_2=1}^{k_2^*} \sum_{i_2 \in \mathcal{V}_{j_2|j_1}} \exp(\beta_{i_2|j_1}^n) \Big[ \exp(-\|\boldsymbol{\omega}_{i_2|j_1}^n - \boldsymbol{x}\|) \exp((\boldsymbol{a}_{j_1}^n)^\top \boldsymbol{x}) \pi(y|(\boldsymbol{\eta}_{j_1 i_2}^n)^\top \boldsymbol{x} + \tau_{j_1 i_2}^n, \nu_{j_1 i_2}^n)$$

$$- \exp(-\|\boldsymbol{\omega}_{j_2|j_1}^* - \boldsymbol{x}\|) \exp((\boldsymbol{a}_{j_1}^*)^\top \boldsymbol{x}) \pi(y|(\boldsymbol{\eta}_{j_1 j_2}^*)^\top \boldsymbol{x} + \tau_{j_1 j_2}^*, \nu_{j_1 j_2}^*) \Big],$$

$$A_{n,j_1,2} := \sum_{j_2=1}^{k_2^*} \sum_{i_2 \in \mathcal{V}_{j_2|j_1}} \exp(\beta_{i_2|j_1}^n) \Big[ \exp(-\|\boldsymbol{\omega}_{i_2|j_1}^n - \boldsymbol{x}\|) - \exp(-\|\boldsymbol{\omega}_{j_2|j_1}^* - \boldsymbol{x}\|) \Big]$$

$$\times \exp((\boldsymbol{a}_{j_1}^n)^\top \boldsymbol{x}) p_{j_1}^{SL,n}(y|\boldsymbol{x}),$$

$$A_{n,j_1,3} := \sum_{j_2=1}^{k_2^*} \Big( \sum_{i_2 \in \mathcal{V}_{j_2|j_1}} \exp(\beta_{i_2|j_1}^n) - \exp(\beta_{j_2|j_1}^*) \Big) \exp(-\|\boldsymbol{\omega}_{j_2|j_1}^* - \boldsymbol{x}\|)$$

$$\times [\exp((\boldsymbol{a}_{j_1}^*)^\top \boldsymbol{x}) \pi(y|(\boldsymbol{\eta}_{j_1 j_2}^*)^\top \boldsymbol{x} + \tau_{j_1 j_2}^*, \nu_{j_1 j_2}^*) - \exp((\boldsymbol{a}_{j_1}^n)^\top \boldsymbol{x}) p_{j_1}^{SL,n}(y|\boldsymbol{x})].$$

Based on the cardinality of the Voronoi cells $\mathcal{V}_{j_2|j_1}$, we proceed to divide the term $A_{n,j_1,1}$ into two parts as

$$A_{n,j_1,1} = \sum_{j_2:|\mathcal{V}_{j_2|j_1}|=1} \sum_{i_2 \in \mathcal{V}_{j_2|j_1}} \exp(\beta_{i_2|j_1}^n) \Big[ \exp(-\|\boldsymbol{\omega}_{i_2|j_1}^n - \boldsymbol{x}\|) \exp((\boldsymbol{a}_{j_1}^n)^\top \boldsymbol{x}) \pi(y|(\boldsymbol{\eta}_{j_1 i_2}^n)^\top \boldsymbol{x} + \tau_{j_1 i_2}^n, \nu_{j_1 i_2}^n)$$

$$- \exp(-\|\boldsymbol{\omega}_{j_2|j_1}^* - \boldsymbol{x}\|) \exp((\boldsymbol{a}_{j_1}^*)^\top \boldsymbol{x}) \pi(y|(\boldsymbol{\eta}_{j_1 j_2}^*)^\top \boldsymbol{x} + \tau_{j_1 j_2}^*, \nu_{j_1 j_2}^*) \Big],$$

$$+ \sum_{j_2:|\mathcal{V}_{j_2|j_1}|>1} \sum_{i_2 \in \mathcal{V}_{j_2|j_1}} \exp(\beta_{i_2|j_1}^n) \Big[ \exp(-\|\boldsymbol{\omega}_{i_2|j_1}^n - \boldsymbol{x}\|) \exp((\boldsymbol{a}_{j_1}^n)^\top \boldsymbol{x}) \pi(y|(\boldsymbol{\eta}_{j_1 i_2}^n)^\top \boldsymbol{x} + \tau_{j_1 i_2}^n, \nu_{j_1 i_2}^n)$$

$$- \exp(-\|\boldsymbol{\omega}_{j_2|j_1}^* - \boldsymbol{x}\|) \exp((\boldsymbol{a}_{j_1}^*)^\top \boldsymbol{x}) \pi(y|(\boldsymbol{\eta}_{j_1 j_2}^*)^\top \boldsymbol{x} + \tau_{j_1 j_2}^*, \nu_{j_1 j_2}^*) \Big]$$

$$:= A_{n,j_1,1,1} + A_{n,j_1,1,2}.$$

Let us denote $F(\boldsymbol{x}; \boldsymbol{\omega}) := \exp(-\|\boldsymbol{\omega} - \boldsymbol{x}\|)$ and $\xi(\boldsymbol{\eta}, \tau) = \boldsymbol{\eta}^\top \boldsymbol{x} + \tau$. By means of the first-order Taylor expansion, $A_{n,j_1,1,1}$ can be represented as

$$A_{n,j_1,1,1} = \sum_{j_2:|\mathcal{V}_{j_2|j_1}|=1} \sum_{i_2 \in \mathcal{V}_{j_2|j_1}} \sum_{|\boldsymbol{\alpha}|=1} \frac{\exp(\beta_{i_2|j_1}^n)}{2^{\alpha_5} \boldsymbol{\alpha}!} (\Delta \boldsymbol{\omega}_{i_2 j_2|j_1}^n)^{\boldsymbol{\alpha}_1} (\Delta \boldsymbol{a}_{j_1}^n)^{\boldsymbol{\alpha}_2} (\Delta \boldsymbol{\eta}_{j_1 i_2 j_2}^n)^{\boldsymbol{\alpha}_3} (\Delta \tau_{j_1 i_2 j_2}^n)^{\alpha_4}$$

$$\times (\Delta \nu_{j_1 i_2 j_2}^n)^{\alpha_5} \boldsymbol{x}^{\boldsymbol{\alpha}_2 + \boldsymbol{\alpha}_3} \frac{\partial^{|\boldsymbol{\alpha}_1|} F}{\partial \boldsymbol{\omega}^{\boldsymbol{\alpha}_1}} (\boldsymbol{x}; \boldsymbol{\omega}_{j_2|j_1}^*) \exp((\boldsymbol{a}_{j_1}^*)^\top \boldsymbol{x}) \frac{\partial^{|\boldsymbol{\alpha}_3| + \alpha_4 + 2\alpha_5} \pi}{\partial \xi^{|\boldsymbol{\alpha}_3| + \alpha_4 + 2\alpha_5}} (y|(\boldsymbol{\eta}_{j_1 j_2}^*)^\top \boldsymbol{x} + \tau_{j_1 j_2}^*, \nu_{j_1 j_2}^*) + R_{n,1,1}(\boldsymbol{x})$$

$$= \sum_{j_2:|\mathcal{V}_{j_2|j_1}|=1} \sum_{|\boldsymbol{\alpha}_1|=0}^{1} \sum_{|\boldsymbol{\rho}_1|+\rho_2=0 \vee 1-|\boldsymbol{\alpha}_1|}^{2(1-|\boldsymbol{\alpha}_1|)} S_{n,j_2|j_1,\boldsymbol{\alpha}_1,\boldsymbol{\rho}_1,\rho_2} \cdot \boldsymbol{x}^{\boldsymbol{\rho}_1} \cdot \frac{\partial^{|\boldsymbol{\alpha}_1|} F}{\partial \boldsymbol{\omega}^{\boldsymbol{\alpha}_1}} (\boldsymbol{x}; \boldsymbol{\omega}_{j_2|j_1}^*) \exp((\boldsymbol{a}_{j_1}^*)^\top \boldsymbol{x})$$

$$\times \frac{\partial^{\rho_2} \pi}{\partial \xi^{\rho_2}} (y|(\boldsymbol{\eta}_{j_1 j_2}^*)^\top \boldsymbol{x} + \tau_{j_1 j_2}^*, \nu_{j_1 j_2}^*) + R_{n,1,1}(\boldsymbol{x}),$$

where $R_{n,1,1}(\boldsymbol{x})$ is a Taylor remainder such that $R_{n,1,1}(\boldsymbol{x})/\mathcal{L}_{2n} \to 0$ as $n \to \infty$, and

$$S_{n,j_2|j_1,\boldsymbol{\alpha}_1,\boldsymbol{\rho}_1,\rho_2} := \sum_{i_2 \in \mathcal{V}_{j_2|j_1}} \sum_{(\boldsymbol{\alpha}_2,\boldsymbol{\alpha}_3,\alpha_4,\alpha_5) \in \mathcal{I}_{\boldsymbol{\rho}_1,\rho_2}^{SL}} \frac{\exp(\beta_{i_2|j_1}^n)}{2^{\alpha_5} \boldsymbol{\alpha}!} (\Delta \boldsymbol{\omega}_{i_2 j_2|j_1}^n)^{\boldsymbol{\alpha}_1} (\Delta \boldsymbol{a}_{j_1}^n)^{\boldsymbol{\alpha}_2} (\Delta \boldsymbol{\eta}_{j_1 i_2 j_2}^n)^{\boldsymbol{\alpha}_3}$$

$$\times (\Delta \tau_{j_1 i_2 j_2}^n)^{\alpha_4} (\Delta \nu_{j_1 i_2 j_2}^n)^{\alpha_5},$$

for any $(\boldsymbol{\alpha}_1, \boldsymbol{\rho}_1, \rho_2) \neq (\boldsymbol{0}_d, \boldsymbol{0}_d, 0)$ and $j_1 \in [k_1^*], j_2 \in [k_2^*]$ in which

$$\mathcal{I}_{\boldsymbol{\rho}_1,\rho_2}^{SL} := \{(\boldsymbol{\alpha}_2, \boldsymbol{\alpha}_3, \alpha_4, \alpha_5) \in \mathbb{R}^d \times \mathbb{R}^d \times \mathbb{R}^d \times \mathbb{R} : \boldsymbol{\alpha}_2 + \boldsymbol{\alpha}_3 = \boldsymbol{\rho}_1, |\boldsymbol{\alpha}_3| + \alpha_4 + 2\alpha_5 = \rho_2\}.$$

For each $(j_1, j_2) \in [k_1^*] \times [k_2^*]$, by applying the Taylor expansion of order $r^{SL}(|\mathcal{V}_{j_2|j_1}|) := r_{j_2|j_1}^{SL}$, the term $A_{n,j_1,1,2}$ can be rewritten as

$$A_{n,j_1,1,2} = \sum_{j_2:|\mathcal{V}_{j_2|j_1}|>1} \sum_{|\boldsymbol{\alpha}_1|=1}^{r_{j_2|j_1}^{SL}} \sum_{|\boldsymbol{\rho}_1|+\boldsymbol{\rho}_2=0\vee 1-|\boldsymbol{\alpha}_1|}^{2(r_{j_2|j_1}^{SL}-|\boldsymbol{\alpha}_1|)} S_{n,j_2|j_1,\boldsymbol{\alpha}_1,\boldsymbol{\rho}_1,\boldsymbol{\rho}_2} \cdot \boldsymbol{x}^{\boldsymbol{\rho}_1} \cdot \frac{\partial^{|\boldsymbol{\alpha}_1|}F}{\partial \boldsymbol{\omega}^{\boldsymbol{\alpha}_1}}(\boldsymbol{x};\boldsymbol{\omega}_{j_2|j_1}^*)\exp((\boldsymbol{a}_{j_1}^*)^\top \boldsymbol{x})$$

$$\times \frac{\partial^{\boldsymbol{\rho}_2}\pi}{\partial \xi^{\boldsymbol{\rho}_2}}(y|(\boldsymbol{\eta}_{j_1 j_2}^*)^\top \boldsymbol{x} + \tau_{j_1 j_2}^*, \nu_{j_1 j_2}^*) + R_{n,1,2}(\boldsymbol{x}),$$

where $R_{n,1,2}(\boldsymbol{x})$ is a Taylor remainder such that $R_{n,1,2}(\boldsymbol{x})/\mathcal{L}_{2n} \to 0$ as $n \to \infty$.

Next, we rewrite the term $A_{n,j_1,2}$ as follows:

$$\sum_{j_2:|\mathcal{V}_{j_2|j_1}|=1} \sum_{i_2 \in \mathcal{V}_{j_2|j_1}} \exp(\beta_{i_2|j_1}^n)\Big[\exp(-\|\boldsymbol{\omega}_{i_2|j_1}^n - \boldsymbol{x}\|) - \exp(-\|\boldsymbol{\omega}_{j_2|j_1}^* - \boldsymbol{x}\|)\Big]\exp((\boldsymbol{a}_{j_1}^n)^\top \boldsymbol{x})p_{j_1}^{SL,n}(y|\boldsymbol{x})$$

$$+ \sum_{j_2:|\mathcal{V}_{j_2|j_1}|>1} \sum_{i_2 \in \mathcal{V}_{j_2|j_1}} \exp(\beta_{i_2|j_1}^n)\Big[\exp(-\|\boldsymbol{\omega}_{i_2|j_1}^n - \boldsymbol{x}\|) - \exp(-\|\boldsymbol{\omega}_{j_2|j_1}^* - \boldsymbol{x}\|)\Big]\exp((\boldsymbol{a}_{j_1}^n)^\top \boldsymbol{x})p_{j_1}^{SL,n}(y|\boldsymbol{x})$$

$$:= A_{n,j_1,2,1} + A_{n,j_1,2,2}.$$

By applying the first-order Taylor expansion, we have

$$A_{n,j_1,2,1} = \sum_{j_2:|\mathcal{V}_{j_2|j_1}|=1} \sum_{i_2 \in \mathcal{V}_{j_2|j_1}} \sum_{|\boldsymbol{\psi}|=1} \frac{\exp(\beta_{i_2|j_1}^n)}{\boldsymbol{\psi}!}(\Delta \boldsymbol{\omega}_{i_2 j_2|j_1}^n)^{\boldsymbol{\psi}}$$

$$\times \frac{\partial^{|\boldsymbol{\psi}|}F}{\partial \boldsymbol{\omega}^{\boldsymbol{\psi}}}(\boldsymbol{x};\boldsymbol{\omega}_{j_2|j_1}^*)\exp((\boldsymbol{a}_{j_1}^n)^\top \boldsymbol{x})p_{j_1}^{SL,n}(y|\boldsymbol{x}) + R_{n,2,1}(\boldsymbol{x}),$$

$$= \sum_{j_2:|\mathcal{V}_{j_2|j_1}|=1} \sum_{|\boldsymbol{\psi}|=1} T_{n,j_2|j_1,\boldsymbol{\psi}} \cdot \frac{\partial^{|\boldsymbol{\psi}|}F}{\partial \boldsymbol{\omega}^{\boldsymbol{\psi}}}(\boldsymbol{x};\boldsymbol{\omega}_{j_2|j_1}^*)\exp((\boldsymbol{a}_{j_1}^n)^\top \boldsymbol{x})p_{j_1}^{SL,n}(y|\boldsymbol{x}) + R_{n,2,1}(\boldsymbol{x}),$$

where $R_{n,2,1}(\boldsymbol{x})$ is a Taylor remainder such that $R_{n,2,1}(\boldsymbol{x})/\mathcal{L}_{2n} \to 0$ as $n \to \infty$, and

$$T_{n,j_2|j_1,\boldsymbol{\psi}} := \sum_{i_2 \in \mathcal{V}_{j_2|j_1}} \frac{\exp(\beta_{i_2|j_1}^n)}{\boldsymbol{\psi}!}(\Delta \boldsymbol{\omega}_{i_2 j_2|j_1}^n)^{\boldsymbol{\psi}},$$

for any $j_2 \in [k_2^*]$ and $\boldsymbol{\psi} \neq \boldsymbol{0}_d$.

Meanwhile, we employ the second-order Taylor expansion to $A_{n,j_1,2,2}$:

$$A_{n,j_1,2,2} = \sum_{j_2:|\mathcal{V}_{j_2|j_1}|>1} \sum_{|\boldsymbol{\psi}|=1}^{2} T_{n,j_2|j_1,\boldsymbol{\psi}} \cdot \frac{\partial^{|\boldsymbol{\psi}|}F}{\partial \boldsymbol{\omega}^{\boldsymbol{\psi}}}(\boldsymbol{x};\boldsymbol{\omega}_{j_2|j_1}^*)\exp((\boldsymbol{a}_{j_1}^n)^\top \boldsymbol{x})p_{j_1}^{SL,n}(y|\boldsymbol{x}) + R_{n,2,2}(\boldsymbol{x}),$$

where $R_{n,2,2}(\boldsymbol{x})$ is a Taylor remainder such that $R_{n,2,2}(\boldsymbol{x})/\mathcal{L}_{2n} \to 0$ as $n \to \infty$.

As a result, the term $A_n$ can be rewritten as

$$A_n = \sum_{j_1=1}^{k_1^*} \sum_{j_2=1}^{k_2^*} \frac{\exp(b_{j_1}^n)}{\sum_{j_2'=1}^{k_2^*} \exp(-\|\boldsymbol{\omega}_{j_2'|j_1}^* - \boldsymbol{x}\| + \beta_{j_2'|j_1}^*)}\Bigg[\sum_{|\boldsymbol{\alpha}_1|=0}^{r_{j_2|j_1}^{SL}} \sum_{|\boldsymbol{\rho}_1|+\boldsymbol{\rho}_2=0\vee 1-|\boldsymbol{\alpha}_1|}^{2(r_{j_2|j_1}^{SL}-|\boldsymbol{\alpha}_1|)} S_{n,j_2|j_1,\boldsymbol{\alpha}_1,\boldsymbol{\rho}_1,\boldsymbol{\rho}_2}$$

$$\times \boldsymbol{x}^{\boldsymbol{\rho}_1} \cdot \frac{\partial^{|\boldsymbol{\alpha}_1|}F}{\partial \boldsymbol{\omega}^{\boldsymbol{\alpha}_1}}(\boldsymbol{x};\boldsymbol{\omega}_{j_2|j_1}^*)\exp((\boldsymbol{a}_{j_1}^*)^\top \boldsymbol{x})\frac{\partial^{\boldsymbol{\rho}_2}\pi}{\partial \xi^{\boldsymbol{\rho}_2}}(y|(\boldsymbol{\eta}_{j_1 j_2}^*)^\top \boldsymbol{x} + \tau_{j_1 j_2}^*, \nu_{j_1 j_2}^*) + R_{n,1,1}(\boldsymbol{x}) + R_{n,1,2}(\boldsymbol{x})$$

$$- \sum_{|\boldsymbol{\psi}|=0}^{2} T_{n,j_2|j_1,\boldsymbol{\psi}} \cdot \frac{\partial^{|\boldsymbol{\psi}|}F}{\partial \boldsymbol{\omega}^{\boldsymbol{\psi}}}(\boldsymbol{x};\boldsymbol{\omega}_{j_2|j_1}^*)\exp((\boldsymbol{a}_{j_1}^n)^\top \boldsymbol{x})p_{j_1}^{SL,n}(y|\boldsymbol{x}) - R_{n,2,1}(\boldsymbol{x}) - R_{n,2,2}(\boldsymbol{x})\Bigg],$$

$$(33)$$

where $S_{n,j_2|j_1,\boldsymbol{\alpha}_1,\boldsymbol{\rho}_1,\boldsymbol{\rho}_2} = T_{n,j_2|j_1,\boldsymbol{\psi}} = \sum_{i_2 \in \mathcal{V}_{j_2|j_1}} \exp(\beta_{i_2|j_1}^n) - \exp(\beta_{j_2|j_1}^*)$ for any $j_2 \in [k_2^*]$ where $(\boldsymbol{\alpha}_1, \boldsymbol{\rho}_1, \boldsymbol{\rho}_2) = (\boldsymbol{0}_d, \boldsymbol{0}_d, 0)$ and $\boldsymbol{\psi} = \boldsymbol{0}_d$.

**Step 1B - Decompose $B_n$:** By invoking the first-order Taylor expansion, we decompose the term $B_n$ defined in equation (32) as

$$B_n = \sum_{j_1=1}^{k_1^*} \exp(b_{j_1}^n) \sum_{|\boldsymbol{\gamma}|=1} (\Delta \boldsymbol{a}_{j_1}^n)^{\boldsymbol{\gamma}} \cdot \boldsymbol{x}^{\boldsymbol{\gamma}} \exp((\boldsymbol{a}_{j_1}^*)^\top \boldsymbol{x}) p_{G_n}^{SL}(y|\boldsymbol{x}) + R_{n,3}(\boldsymbol{x}), \qquad (34)$$

where $R_{n,3}(\boldsymbol{x})$ is a Taylor remainder such that $R_{n,3}(\boldsymbol{x})/\mathcal{L}_{2n} \to 0$ as $n \to \infty$.

It can be seen from the decomposition in equations (32), (33) and (34) that $A_n$, $B_n$ and $C_n$ can be treated as a linear combination of elements from the following set union:

$$\left\{ \boldsymbol{x}^{\boldsymbol{\rho}_1} \cdot \frac{\partial^{|\boldsymbol{\alpha}_1|} F}{\partial \boldsymbol{\omega}^{\boldsymbol{\alpha}_1}}(\boldsymbol{x}; \boldsymbol{\omega}_{j_2|j_1}^*) \exp((\boldsymbol{a}_{j_1}^*)^\top \boldsymbol{x}) \frac{\partial^{\rho_2} \pi}{\partial \xi^{\rho_2}}(y|(\boldsymbol{\eta}_{j_1 j_2}^*)^\top \boldsymbol{x} + \tau_{j_1 j_2}^*, \nu_{j_1 j_2}^*) : j_1 \in [k_1^*], \; j_2 \in [k_2^*], \right.$$

$$\left. 0 \le |\boldsymbol{\alpha}_1| \le r_{j_2|j_1}^{SL}, \; 0 \le |\boldsymbol{\rho}_1| + \rho_2 \le 2(r_{j_2|j_1}^{SL} - |\boldsymbol{\alpha}_1|) \right\}$$

$$\cup \left\{ \frac{\frac{\partial^{|\boldsymbol{\psi}|} F}{\partial \boldsymbol{\omega}^{\boldsymbol{\psi}}}(\boldsymbol{x}; \boldsymbol{\omega}_{j_2|j_1}^*) \exp((\boldsymbol{a}_{j_1}^n)^\top \boldsymbol{x}) p_{j_1}^{SL,n}(y|\boldsymbol{x})}{\sum_{j_2'=1}^{k_2^*} \exp(-\|\boldsymbol{\omega}_{j_2'|j_1}^* - \boldsymbol{x}\| + \beta_{j_2'|j_1}^*)} : j_1 \in [k_1^*], \; j_2 \in [k_2^*], \; 0 \le |\boldsymbol{\psi}| \le 2 \right\}$$

$$\cup \left\{ \boldsymbol{x}^{\boldsymbol{\gamma}} \exp((\boldsymbol{a}_{j_1}^*)^\top \boldsymbol{x}) p_{j_1}^{SL,n}(y|\boldsymbol{x}), \; \boldsymbol{x}^{\boldsymbol{\gamma}} \exp((\boldsymbol{a}_{j_1}^*)^\top \boldsymbol{x}) p_{G_n}^{SL}(y|\boldsymbol{x}) : j_1 \in [k_1^*], \; 0 \le |\boldsymbol{\gamma}| \le 1 \right\}.$$

**Step 2 - Non-vanishing coefficients:** In this stage, we illustrate that not all the coefficients in the representation of $A_n/\mathcal{L}_{2n}$, $B_n/\mathcal{L}_{2n}$ and $C_n/\mathcal{L}_{2n}$ go to zero as $n \to \infty$. Suppose that all of them approach zero, then we examine the coefficients associated with the term

- $\exp((\boldsymbol{a}_{j_1}^*)^\top \boldsymbol{x}) p_{j_1}^{SL,n}(y|\boldsymbol{x})$ in $C_n/\mathcal{L}_{2n}$, we have

$$\frac{1}{\mathcal{L}_{2n}} \cdot \sum_{j_1=1}^{k_1^*} \left| \exp(b_{j_1}^n) - \exp(b_{j_1}^*) \right| \to 0. \qquad (35)$$

- $\dfrac{F(\boldsymbol{x}; \boldsymbol{\omega}_{j_2|j_1}^*) \exp((\boldsymbol{a}_{j_1}^*)^\top \boldsymbol{x}) \pi(y|(\boldsymbol{\eta}_{j_1 j_2}^*)^\top \boldsymbol{x} + \tau_{j_1 j_2}^*, \nu_{j_1 j_2}^*)}{\sum_{j_2'=1}^{k_2^*} \exp(-\|\boldsymbol{\omega}_{j_2'|j_1}^* - \boldsymbol{x}\| + \beta_{j_2'|j_1}^*)}$ in $A_n/\mathcal{L}_{2n}$, we get that

$$\frac{1}{\mathcal{L}_{2n}} \cdot \sum_{j_1=1}^{k_1^*} \exp(b_{j_1}^n) \sum_{j_2=1}^{k_2^*} \left| \sum_{i_2 \in \mathcal{V}_{j_2|j_1}} \exp(\beta_{i_2|j_1}^n) - \exp(\beta_{j_2|j_1}^*) \right| \to 0. \qquad (36)$$

- $\dfrac{\frac{\partial^{|\boldsymbol{\alpha}_1|} F}{\partial \boldsymbol{\omega}^{\boldsymbol{\alpha}_1}}(\boldsymbol{x}; \boldsymbol{\omega}_{j_2|j_1}^*) \exp((\boldsymbol{a}_{j_1}^n)^\top \boldsymbol{x}) \pi(y|(\boldsymbol{\eta}_{j_1 j_2}^*)^\top \boldsymbol{x} + \tau_{j_1 j_2}^*, \nu_{j_1 j_2}^*)}{\sum_{j_2'=1}^{k_2^*} \exp(-\|\boldsymbol{\omega}_{j_2'|j_1}^* - \boldsymbol{x}\| + \beta_{j_2'|j_1}^*)}$ in $A_n/\mathcal{L}_{2n}$ for $j_1 \in$ $[k_1^*], j_2 \in [k_2^*] : |\mathcal{V}_{j_2|j_1}| = 1$ and $\boldsymbol{\alpha}_1 = e_{d,u}$ where $e_{d,u} := (0, \ldots, 0, \underbrace{1}_{u\text{-}th}, 0, \ldots, 0) \in \mathbb{N}^d$, we receive

$$\frac{1}{\mathcal{L}_{2n}} \cdot \sum_{j_1=1}^{k_1^*} \exp(b_{j_1}^n) \sum_{j_2 \in [k_2^*] : |\mathcal{V}_{j_2|j_1}|=1} \sum_{i_2 \in \mathcal{V}_{j_2|j_1}} \exp(\beta_{i_2|j_1}^n) \|\boldsymbol{\omega}_{i_2|j_1}^n - \boldsymbol{\omega}_{j_2|j_1}^*\|_1 \to 0.$$

Note that since the norm-1 is equivalent to the norm-2, then we can replace the norm-1 with the norm-2, that is,

$$\frac{1}{\mathcal{L}_{2n}} \cdot \sum_{j_1=1}^{k_1^*} \exp(b_{j_1}^n) \sum_{j_2 \in [k_2^*] : |\mathcal{V}_{j_2|j_1}|=1} \sum_{i_2 \in \mathcal{V}_{j_2|j_1}} \exp(\beta_{i_2|j_1}^n) \|\boldsymbol{\omega}_{i_2|j_1}^n - \boldsymbol{\omega}_{j_2|j_1}^*\| \to 0. \qquad (37)$$

- $\dfrac{F(\boldsymbol{x};\boldsymbol{\omega}^*_{j_2|j_1})\exp((\boldsymbol{a}^*_{j_1})^\top\boldsymbol{x})\frac{\partial^{\rho_2}\pi}{\partial\xi^{\rho_2}}(y|(\boldsymbol{\eta}^*_{j_1j_2})^\top\boldsymbol{x}+\tau^*_{j_1j_2},\nu^*_{j_1j_2})}{\sum_{j'_2=1}^{k^*_2}\exp(-\|\boldsymbol{\omega}^*_{j'_2|j_1}-\boldsymbol{x}\|+\beta^*_{j'_2|j_1})}$ in $A_n/\mathcal{L}_{2n}$ for $j_1 \in [k^*_1], j_2 \in [k^*_2] : |\mathcal{V}_{j_2|j_1}| = 1$ and $\rho_2 = 1$, we have that

$$\frac{1}{\mathcal{L}_{2n}}\cdot\sum_{j_1=1}^{k^*_1}\exp(b^n_{j_1})\sum_{j_2\in[k^*_2]:|\mathcal{V}_{j_2|j_1}|=1}\exp(\beta^n_{j_2|j_1})|\tau^n_{j_1j_2}-\tau^*_{j_1j_2}|\to 0. \tag{38}$$

- $\dfrac{\boldsymbol{x}^{\boldsymbol{\rho}_1}F(\boldsymbol{x};\boldsymbol{\omega}^*_{j_2|j_1})\exp((\boldsymbol{a}^*_{j_1})^\top\boldsymbol{x})\frac{\partial^{\rho_2}\pi}{\partial\xi^{\rho_2}}(y|(\boldsymbol{\eta}^*_{j_1j_2})^\top\boldsymbol{x}+\tau^*_{j_1j_2},\nu^*_{j_1j_2})}{\sum_{j'_2=1}^{k^*_2}\exp(-\|\boldsymbol{\omega}^*_{j'_2|j_1}-\boldsymbol{x}\|+\beta^*_{j'_2|j_1})}$ in $A_n/\mathcal{L}_{2n}$ for $j_1 \in [k^*_1], j_2 \in [k^*_2] : |\mathcal{V}_{j_2|j_1}| = 1, \boldsymbol{\rho}_1 = e_{d,u}$ and $\rho_2 = 1$, we have that

$$\frac{1}{\mathcal{L}_{2n}}\cdot\sum_{j_1=1}^{k^*_1}\exp(b^n_{j_1})\sum_{j_2\in[k^*_2]:|\mathcal{V}_{j_2|j_1}|=1}\sum_{i_2\in\mathcal{V}_{j_2|j_1}}\exp(\beta^n_{j_2|j_1})\|\boldsymbol{\eta}^n_{j_1i_2}-\boldsymbol{\eta}^*_{j_1j_2}\|\to 0. \tag{39}$$

- $\dfrac{F(\boldsymbol{x};\boldsymbol{\omega}^*_{j_2|j_1})\exp((\boldsymbol{a}^*_{j_1})^\top\boldsymbol{x})\frac{\partial^{\rho_2}\pi}{\partial\xi^{\rho_2}}(y|(\boldsymbol{\eta}^*_{j_1j_2})^\top\boldsymbol{x}+\tau^*_{j_1j_2},\nu^*_{j_1j_2})}{\sum_{j'_2=1}^{k^*_2}\exp(-\|\boldsymbol{\omega}^*_{j'_2|j_1}-\boldsymbol{x}\|+\beta^*_{j'_2|j_1})}$ in $A_n/\mathcal{L}_{2n}$ for $j_1 \in [k^*_1], j_2 \in [k^*_2] : |\mathcal{V}_{j_2|j_1}| = 1$ and $\rho_2 = 2$, we have that

$$\frac{1}{\mathcal{L}_{2n}}\cdot\sum_{j_1=1}^{k^*_1}\exp(b^n_{j_1})\sum_{j_2\in[k^*_2]:|\mathcal{V}_{j_2|j_1}|=1}\exp(\beta^n_{j_2|j_1})|\nu^n_{j_1j_2}-\nu^*_{j_1j_2}|\to 0. \tag{40}$$

- $\boldsymbol{x}^{\boldsymbol{\gamma}}\exp((\boldsymbol{a}^*_{j_1})^\top\boldsymbol{x})p^{SL}_{G_n}(y|\boldsymbol{x})$ in $B_n/\mathcal{L}_{2n}$ for $j_1 \in [k^*_1]$ and $\boldsymbol{\gamma} = e_{d,u}$, we obtain

$$\frac{1}{\mathcal{L}_{2n}}\cdot\sum_{j_1=1}^{k^*_1}\exp(b^n_{j_1})\|\boldsymbol{a}^n_{j_1}-\boldsymbol{a}^*_{j_1}\|\to 0. \tag{41}$$

- $\dfrac{\frac{\partial^{|\boldsymbol{\alpha}_1|}F}{\partial\boldsymbol{\omega}^{\boldsymbol{\alpha}_1}}(\boldsymbol{x};\boldsymbol{\omega}^*_{j_2|j_1})\exp((\boldsymbol{a}^*_{j_1})^\top\boldsymbol{x})\pi(y|(\boldsymbol{\eta}^*_{j_1j_2})^\top\boldsymbol{x}+\tau^*_{j_1j_2},\nu^*_{j_1j_2})}{\sum_{j'_2=1}^{k^*_2}\exp(-\|\boldsymbol{\omega}^*_{j'_2|j_1}-\boldsymbol{x}\|+\beta^*_{j'_2|j_1})}$ in $A_n/\mathcal{L}_{2n}$ for $j_1 \in [k^*_1], j_2 \in [k^*_2] : |\mathcal{V}_{j_2|j_1}| > 1$ and $\boldsymbol{\alpha}_1 = 2e_{d,u}$, we receive that

$$\frac{1}{\mathcal{L}_{2n}}\cdot\sum_{j_1=1}^{k^*_1}\exp(b^n_{j_1})\sum_{j_2\in[k^*_2]:|\mathcal{V}_{j_2|j_1}|>1}\sum_{i_2\in\mathcal{V}_{j_2|j_1}}\exp(\beta^n_{i_2|j_1})\|\boldsymbol{\omega}^n_{i_2|j_1}-\boldsymbol{\omega}^*_{j_2|j_1}\|^2\to 0. \tag{42}$$

Putting the above limits together with the formulation of the loss $\mathcal{L}_{2n}$ in equation (31), we deduce that

$$\frac{1}{\mathcal{L}_{2n}}\cdot\sum_{j_1=1}^{k^*_1}\exp(b^n_{j_1})\left[\sum_{j_2:|\mathcal{V}_{j_2|j_1}|>1}\sum_{i_2\in\mathcal{V}_{j_2|j_1}}\exp(\beta^n_{i_2|j_1})\left(\|\Delta\boldsymbol{\eta}^n_{j_1i_2j_2}\|^{\frac{r^{SL}_{j_2|j_1}}{2}}+|\Delta\tau^n_{j_1i_2j_2}|^{r^{SL}_{j_2|j_1}}\right.\right.$$
$$\left.\left.+|\Delta\nu^n_{j_1i_2j_2}|^{\frac{r^{SL}_{j_2|j_1}}{2}}\right)\right]\not\to 0,$$

which also suggests that

$$\frac{1}{\mathcal{L}_{2n}}\cdot\sum_{j_1=1}^{k^*_1}\exp(b^n_{j_1})\left[\sum_{j_2:|\mathcal{V}_{j_2|j_1}|>1}\sum_{i_2\in\mathcal{V}_{j_2|j_1}}\exp(\beta^n_{i_2|j_1})\left(\|\Delta\boldsymbol{a}^n_{j_1}\|^{r^{SL}_{j_2|j_1}}+\|\Delta\boldsymbol{\eta}^n_{j_1i_2j_2}\|^{\frac{r^{SL}_{j_2|j_1}}{2}}\right.\right.$$
$$\left.\left.+|\Delta\tau^n_{j_1i_2j_2}|^{r^{SL}_{j_2|j_1}}+|\Delta\nu^n_{j_1i_2j_2}|^{\frac{r^{SL}_{j_2|j_1}}{2}}\right)\right]\not\to 0,$$

as $n \to \infty$. Thus, we can find indices $j_1^* \in [k_1^*]$ and $j_2^* \in [k_2^*] : |\mathcal{V}_{j_2^*|j_1^*}| > 1$ such that

$$\frac{1}{\mathcal{L}_{2n}} \cdot \sum_{i_2 \in \mathcal{V}_{j_2^*|j_1^*}} \exp(\beta_{i_2|j_1^*}^n) \Big( \|\boldsymbol{a}_{j_1^*}^n - \boldsymbol{a}_{j_1^*}^*\|^{r_{j_2^*|j_1^*}^{SL}} + \|\boldsymbol{\eta}_{j_1^* i_2}^n - \boldsymbol{\eta}_{j_1^* j_2^*}^*\|^{\frac{r_{j_2^*|j_1^*}^{SL}}{2}}$$

$$+ |\tau_{j_1^* i_2}^n - \tau_{j_1^* j_2^*}^*|^{r_{j_2^*|j_1^*}^{SL}} + |\nu_{j_1^* i_2}^n - \nu_{j_1^* j_2^*}^*|^{\frac{r_{j_2^*|j_1^*}^{SL}}{2}} \Big) \not\to 0. \quad (43)$$

WLOG, we may assume that $j_1^* = j_2^* = 1$. By considering the coefficients of the terms $\frac{\boldsymbol{x}^{\boldsymbol{\rho}_1} F(\boldsymbol{x}; \boldsymbol{\omega}_{j_2|j_1}^*) \exp((\boldsymbol{a}_{j_1}^*)^\top \boldsymbol{x}) \frac{\partial^{\rho_2} \pi}{\partial \xi^{\rho_2}} (y | (\boldsymbol{\eta}_{j_1 j_2}^*)^\top \boldsymbol{x} + \tau_{j_1 j_2}^*, \nu_{j_1 j_2}^*)}{\sum_{j_2'=1}^{k_2^*} \exp(-\|\boldsymbol{\omega}_{j_2'|j_1}^* - \boldsymbol{x}\| + \beta_{j_2'|j_1}^*)}$ in $A_n/\mathcal{L}_{2n}$ for $j_1 = j_2 = 1$, we have $\exp(b_1^n) S_{n,1|1,\boldsymbol{0}_d,\boldsymbol{\rho}_1,\rho_2}/\mathcal{L}_{2n} \to 0$, or equivalently,

$$\frac{1}{\mathcal{L}_{2n}} \cdot \sum_{i_2 \in \mathcal{V}_{1|1}} \sum_{(\boldsymbol{\alpha}_2,\boldsymbol{\alpha}_3,\alpha_4,\alpha_5) \in \mathcal{I}_{\boldsymbol{\rho}_1,\rho_2}^{SL}} \frac{\exp(\beta_{i_2|1}^n)}{2^{\alpha_5} \boldsymbol{\alpha}_2! \boldsymbol{\alpha}_3! \alpha_4! \alpha_5!} \cdot (\Delta \boldsymbol{a}_1^n)^{\boldsymbol{\alpha}_2} (\Delta \boldsymbol{\eta}_{1 i_2 1}^n)^{\boldsymbol{\alpha}_3}$$

$$\times (\Delta \tau_{1 i_2 1}^n)^{\alpha_4} (\Delta \nu_{1 i_2 1}^n)^{\alpha_5} \to 0. \quad (44)$$

By dividing the left hand side of equation (44) by that of equation (43), we get

$$\frac{\sum_{i_2 \in \mathcal{V}_{1|1}} \sum_{(\boldsymbol{\alpha}_2,\boldsymbol{\alpha}_3,\alpha_4,\alpha_5) \in \mathcal{I}_{\boldsymbol{\rho}_1,\rho_2}^{SL}} \frac{\exp(\beta_{i_2|1}^n)}{2^{\alpha_5} \boldsymbol{\alpha}_2! \boldsymbol{\alpha}_3! \alpha_4! \alpha_5!} \cdot (\Delta \boldsymbol{a}_1^n)^{\boldsymbol{\alpha}_2} (\Delta \boldsymbol{\eta}_{1 i_2 1}^n)^{\boldsymbol{\alpha}_3} (\Delta \tau_{1 i_2 1}^n)^{\alpha_4} (\Delta \nu_{1 i_2 1}^n)^{\alpha_5}}{\sum_{i_2 \in \mathcal{V}_{1|1}} \exp(\beta_{i_2|1}^n) \Big( \|\Delta \boldsymbol{a}_1^n\|^{r_{1|1}^{SL}} + \|\Delta \boldsymbol{\eta}_{1 i_2 i}^n\|^{\frac{r_{1|1}^{SL}}{2}} + |\Delta \tau_{1 i_2 1}^n|^{r_{1|1}^{SL}} + |\Delta \nu_{1 i_2 1}^n|^{\frac{r_{1|1}^{SL}}{2}} \Big)} \to 0.$$
$$(45)$$

Let us define $\overline{M}_n := \max\{\|\Delta \boldsymbol{a}_1^n\|, \|\Delta \boldsymbol{\eta}_{1 i_2 i}^n\|^{1/2}, \|\Delta \tau_{1 i_2 1}^n\|, \|\Delta \nu_{1 i_2 1}^n\|^{1/2} : i_2 \in \mathcal{V}_{1|1}\}$, and $\overline{\beta}_n := \max_{i_2 \in \mathcal{V}_{1|1}} \exp(\beta_{i_2|1}^n)$. Since the sequence $\exp(\beta_{i_2|1}^n)/\overline{\beta}_n$ is bounded, we can replace it by its subsequence which has a positive limit $p_{i_2}^2 := \lim_{n \to \infty} \exp(\beta_{i_2|1}^n)/\overline{\beta}_n$. Note that at least one among the limits $p_{i_2}^2$ must be equal to one. Next, let us define

$$(\Delta \boldsymbol{a}_1^n)/\overline{M}_n \to \boldsymbol{q}_2, \quad (\Delta \boldsymbol{\eta}_{1 i_2 1}^n)/\overline{M}_n \to \boldsymbol{q}_{3 i_2},$$
$$(\Delta \tau_{1 i_2 1}^n)/\overline{M}_n \to q_{4 i_2}, \quad (\Delta \nu_{1 i_2 1}^n)/2\overline{M}_n \to q_{5 i_2}.$$

Note that at least one among $q_2, q_{3 i_2}, q_{4 i_2}, q_{5 i_2}$ must be equal to either 1 or $-1$.

By dividing both the numerator and the denominator of the term in equation (45) by $\overline{\beta}_n \overline{M}_n^{|\boldsymbol{\rho}_1|+\rho_2}$, we obtain the system of polynomial equations:

$$\sum_{i_2 \in \mathcal{V}_{1|1}} \sum_{(\boldsymbol{\alpha}_2,\boldsymbol{\alpha}_3,\alpha_4,\alpha_5) \in \mathcal{I}_{\boldsymbol{\rho}_1,\rho_2}^{SL}} \frac{1}{\boldsymbol{\alpha}_2! \boldsymbol{\alpha}_3! \alpha_4! \alpha_5!} \cdot p_{i_2}^2 \boldsymbol{q}_2^{\boldsymbol{\alpha}_2} \boldsymbol{q}_{3 i_2}^{\boldsymbol{\alpha}_3} q_{4 i_2}^{\alpha_4} q_{5 i_2}^{\alpha_5} = 0, \quad 1 \le |\boldsymbol{\rho}_1| + \rho_2 \le r_{1|1}^{SL}.$$

According to the definition of the term $r_{1|1}^{SL}$, the above system does not have any non-trivial solutions, which is a contradiction. Consequently, at least one among the coefficients in the representation of $A_n/\mathcal{L}_{2n}$, $B_n/\mathcal{L}_{2n}$ and $C_n/\mathcal{L}_{2n}$ must not converge to zero as $n \to \infty$.

**Step 3 - Application of the Fatou's lemma.** In this stage, we show that all the coefficients in the formulations of $A_n/\mathcal{L}_{2n}$, $B_n/\mathcal{L}_{2n}$ and $C_n/\mathcal{L}_{2n}$ go to zero as $n \to \infty$. Denote by $m_n$ the maximum of the absolute values of those coefficients, the result from Step 2 induces that $1/m_n \not\to \infty$. By employing the Fatou's lemma, we have

$$0 = \lim_{n \to \infty} \frac{\mathbb{E}_{\boldsymbol{X}}[V(p_{G_n}^{SL}(\cdot|\boldsymbol{X}), p_{G_*}^{SL}(\cdot|\boldsymbol{X}))]}{m_n \mathcal{L}_{2n}} \ge \int \liminf_{n \to \infty} \frac{|p_{G_n}^{SL}(y|\boldsymbol{x}) - p_{G_*}^{SL}(y|\boldsymbol{x})|}{2 m_n \mathcal{L}_{2n}} \mathrm{d}(\boldsymbol{x}, y).$$

Thus, we deduce that

$$\frac{|p_{G_n}^{SL}(y|\boldsymbol{x}) - p_{G_*}^{SL}(y|\boldsymbol{x})|}{2 m_n \mathcal{L}_{2n}} \to 0,$$

which results in $Q_n/[m_n \mathcal{L}_{2n}] \to 0$ as $n \to \infty$ for almost surely $(\boldsymbol{x}, y)$.

Next, we denote

$$\frac{\exp(b_{j_1}^n)S_{n,j_2|j_1,\boldsymbol{\alpha}_1,\boldsymbol{\rho}_1,\rho_2}}{m_n\mathcal{L}_{2n}} \to \phi_{j_2|j_1,\boldsymbol{\alpha}_1,\boldsymbol{\rho}_1,\rho_2}, \qquad \frac{\exp(b_{j_1}^n)T_{n,j_2|j_1,\boldsymbol{\psi}}}{m_n\mathcal{L}_{2n}} \to \varphi_{j_2|j_1,\boldsymbol{\psi}},$$

$$\frac{\exp(b_{j_1}^n)(\Delta\boldsymbol{a}_{j_1}^n)^{\boldsymbol{\gamma}}}{m_n\mathcal{L}_{2n}} \to \lambda_{j_1,\boldsymbol{\gamma}}, \qquad \frac{\exp(b_{j_1}^n) - \exp(b_{j_1}^*)}{m_n\mathcal{L}_{2n}} \to \chi_{j_1}$$

with a note that at least one among them is non-zero. Then, the decomposition of $Q_n$ in equation (32) indicates that

$$\lim_{n\to\infty}\frac{Q_n}{m_n\mathcal{L}_{2n}} = \lim_{n\to\infty}\frac{A_n}{m_n\mathcal{L}_{2n}} - \lim_{n\to\infty}\frac{B_n}{m_n\mathcal{L}_{2n}} + \lim_{n\to\infty}\frac{C_n}{m_n\mathcal{L}_{2n}},$$

in which

$$\lim_{n\to\infty}\frac{A_n}{m_n\mathcal{L}_{2n}} = \sum_{j_1=1}^{k_1^*}\sum_{j_2=1}^{k_2^*}\left[\sum_{|\boldsymbol{\alpha}_1|=1}^{r_{j_2|j_1}^{SL}}\sum_{|\boldsymbol{\rho}_1|+\rho_2=0\vee 1-|\boldsymbol{\alpha}_1|}^{2(r_{j_2|j_1}^{SL}-|\boldsymbol{\alpha}_1|)} S_{n,j_2|j_1,\boldsymbol{\alpha}_1,\boldsymbol{\rho}_1,\rho_2} \cdot \boldsymbol{x}^{\boldsymbol{\rho}_1}\frac{\partial^{|\boldsymbol{\alpha}_1|}F}{\partial\boldsymbol{\omega}^{\boldsymbol{\alpha}_1}}(\boldsymbol{x};\boldsymbol{\omega}_{j_2|j_1}^*)\right.$$

$$\times \exp((\boldsymbol{a}_{j_1}^*)^\top\boldsymbol{x})\frac{\partial^{\rho_2}\pi}{\partial\xi^{\rho_2}}(y|(\boldsymbol{\eta}_{j_1j_2}^*)^\top\boldsymbol{x}+\tau_{j_1j_2}^*,\nu_{j_1j_2}^*) - \sum_{|\boldsymbol{\psi}|=0}^{2}\varphi_{j_2|j_1,\boldsymbol{\psi}}\cdot\frac{\partial^{|\boldsymbol{\psi}|}F}{\partial\boldsymbol{\omega}^{\boldsymbol{\psi}}}(\boldsymbol{x};\boldsymbol{\omega}_{j_2|j_1}^*)$$

$$\left.\times \exp((\boldsymbol{a}_{j_1}^*)^\top\boldsymbol{x})p_{j_1}^{SL,*}(y|\boldsymbol{x})\right]\frac{1}{\sum_{j_2'=1}^{k_2^*}\exp(-\|\boldsymbol{\omega}_{j_2'|j_1}^* - \boldsymbol{x}\| + \beta_{j_2'|j_1}^*)},$$

$$\lim_{n\to\infty}\frac{B_n}{m_n\mathcal{L}_{2n}} = \sum_{j_1=1}^{k_1^*}\sum_{|\boldsymbol{\gamma}|=1}\lambda_{j_1,\boldsymbol{\gamma}}\cdot\boldsymbol{x}^{\boldsymbol{\gamma}}\exp((\boldsymbol{a}_{j_1}^*)^\top\boldsymbol{x})p_{G_*}^{SL}(y|\boldsymbol{x}),$$

$$\lim_{n\to\infty}\frac{C_n(\boldsymbol{x})}{m_n\mathcal{L}_{2n}} = \sum_{j_1=1}^{k_1^*}\chi_{j_1}\exp((\boldsymbol{a}_{j_1}^*)^\top\boldsymbol{x})\left[p_{j_1}^{SL,*}(y|\boldsymbol{x}) - p_{G_*}^{SL}(y|\boldsymbol{x})\right].$$

Since the set

$$\left\{\frac{\boldsymbol{x}^{\boldsymbol{\rho}_1}\frac{\partial^{|\boldsymbol{\alpha}_1|}F}{\partial\boldsymbol{\omega}^{\boldsymbol{\alpha}_1}}(\boldsymbol{x};\boldsymbol{\omega}_{j_2|j_1}^*)\exp((\boldsymbol{a}_{j_1}^*)^\top\boldsymbol{x})\frac{\partial^{\rho_2}\pi}{\partial\xi^{\rho_2}}(y|(\boldsymbol{\eta}_{j_1j_2}^*)^\top\boldsymbol{x}+\tau_{j_1j_2}^*,\nu_{j_1j_2}^*)}{\sum_{j_2'=1}^{k_2^*}\exp(-\|\boldsymbol{\omega}_{j_2'|j_1}^* - \boldsymbol{x}\| + \beta_{j_2'|j_1}^*)} : j_1\in[k_1^*], j_2\in[k_2^*],\right.$$

$$\left.0\le|\boldsymbol{\alpha}_1|\le r_{j_2|j_1}^{SL}, 0\le|\boldsymbol{\rho}_1|+\rho_2\le 2(r_{j_2|j_1}^{SL}-|\boldsymbol{\alpha}_1|)\right\}$$

$$\cup\left\{\frac{\frac{\partial^{|\boldsymbol{\psi}|}F}{\partial\boldsymbol{\omega}^{\boldsymbol{\psi}}}(\boldsymbol{x};\boldsymbol{\omega}_{j_2|j_1}^*)\exp((\boldsymbol{a}_{j_1}^*)^\top\boldsymbol{x})p_{j_1}^{SL,*}(y|\boldsymbol{x})}{\sum_{j_2'=1}^{k_2^*}\exp(-\|\boldsymbol{\omega}_{j_2'|j_1}^* - \boldsymbol{x}\| + \beta_{j_2'|j_1}^*)} : j_1\in[k_1^*], j_2\in[k_2^*], 0\le|\boldsymbol{\psi}|\le 2\right\}$$

$$\cup\left\{\boldsymbol{x}^{\boldsymbol{\gamma}}\exp((\boldsymbol{a}_{j_1}^*)^\top\boldsymbol{x})p_{G_*}^{SL}(y|\boldsymbol{x}), \exp((\boldsymbol{a}_{j_1}^*)^\top\boldsymbol{x})p_{j_1}^{SL,*}(y|\boldsymbol{x}), \exp((\boldsymbol{a}_{j_1}^*)^\top\boldsymbol{x})p_{G_*}^{SL}(y|\boldsymbol{x})\right.$$

$$\left.: j_1\in[k_1^*], 0\le|\boldsymbol{\gamma}|\le 2\right\}$$

is linearly independent, we obtain that $\phi_{j_2|j_1,\boldsymbol{\alpha}_1,\boldsymbol{\rho}_1,\rho_2} = \varphi_{j_2|j_1,\boldsymbol{\psi}} = \lambda_{j_1,\boldsymbol{\gamma}} = \chi_{j_1} = 0$ for all $j_1\in[k_1^*]$, $j_2\in[k_2^*]$, $0\le|\boldsymbol{\alpha}_1|\le r_{j_2|j_1}^{SL}$, $0\le|\boldsymbol{\rho}_1|+\rho_2\le 2(r_{j_2|j_1}^{SL}-|\boldsymbol{\alpha}_1|)$, $0\le|\boldsymbol{\psi}|\le 2$ and $0\le|\boldsymbol{\gamma}|\le 1$, which is a contradiction. As a consequence, we obtain the inequality in equation (12). Hence, the proof is completed.

### E.3 WHEN $type = LL$

When $type = LL$, the corresponding Voronoi loss function is $\mathcal{L}_{(2, r^{LL}, \frac{1}{2} r^{LL})}(G_n, G_*) = \mathcal{L}_{3n}$ where we define

$$
\mathcal{L}_{3n} := \sum_{j_1=1}^{k_1^*} \left| \exp(b_{j_1}^n) - \exp(b_{j_1}^*) \right| + \sum_{j_1=1}^{k_1^*} \exp(b_{j_1}^n) \|\Delta \boldsymbol{a}_{j_1}^n\| + \sum_{j_1=1}^{k_1^*} \exp(b_{j_1}^n)
$$

$$
\times \left[ \sum_{j_2: |\mathcal{V}_{j_2|j_1}|=1} \sum_{i_2 \in \mathcal{V}_{j_2|j_1}} \exp(\beta_{i_2|j_1}^n) \left( \|\Delta \boldsymbol{\omega}_{i_2 j_2|j_1}^n\| + \|\Delta \boldsymbol{\eta}_{j_1 i_2 j_2}^n\| + |\Delta \tau_{j_1 i_2 j_2}^n| + |\Delta \nu_{j_1 i_2 j_2}^n| \right) \right.
$$

$$
+ \sum_{j_2: |\mathcal{V}_{j_2|j_1}|>1} \sum_{i_2 \in \mathcal{V}_{j_2|j_1}} \exp(\beta_{i_2|j_1}^n) \left( \|\Delta \boldsymbol{\omega}_{i_2 j_2|j_1}^n\|^2 + \|\Delta \boldsymbol{\eta}_{j_1 i_2 j_2}^n\|^2 + |\Delta \tau_{j_1 i_2 j_2}^n|^{r_{j_2|j_1}^{LL}} \right.
$$

$$
\left. + |\Delta \nu_{j_1 i_2 j_2}^n|^{\frac{r_{j_2|j_1}^{LL}}{2}} \right) \right] + \sum_{j_1=1}^{k_1^*} \exp(b_{j_1}^n) \sum_{j_2=1}^{k_2^*} \left| \sum_{i_2 \in \mathcal{V}_{j_2|j_1}} \exp(\beta_{i_2|j_1}^n) - \exp(\beta_{j_2|j_1}^*) \right|. \tag{46}
$$

**Step 1 - Taylor expansion:** In this step, we use the Taylor expansion to decompose the term

$$
Q_n := \left[ \sum_{j_1=1}^{k_1^*} \exp(-\|\boldsymbol{a}_{j_1}^* - \boldsymbol{x}\| + b_{j_1}^*) \right] [p_{G_n}^{LL}(y|\boldsymbol{x}) - p_{G_*}^{LL}(y|\boldsymbol{x})].
$$

Prior to that, let us denote

$$
p_{j_1}^{LL,n}(y|\boldsymbol{x}) := \sum_{j_2=1}^{k_2^*} \sum_{i_2 \in \mathcal{V}_{j_2|j_1}} \sigma(-\|\boldsymbol{\omega}_{i_2|j_1}^n - \boldsymbol{x}\| + \beta_{i_2|j_1}^n) \pi(y|(\boldsymbol{\eta}_{j_1 i_2}^n)^\top \boldsymbol{x} + \tau_{j_1 i_2}^n, \nu_{j_1 i_2}^n),
$$

$$
p_{j_1}^{LL,*}(y|\boldsymbol{x}) := \sum_{j_2=1}^{k_2^*} \sigma(-\|\boldsymbol{\omega}_{j_2|j_1}^* - \boldsymbol{x}\| + \beta_{j_2|j_1}^*) \pi(y|(\boldsymbol{\eta}_{j_1 j_2}^*)^\top \boldsymbol{x} + \tau_{j_1 j_2}^*, \nu_{j_1 j_2}^*).
$$

Then, the quantity $Q_n$ is divided into three terms as

$$
Q_n = \sum_{j_1=1}^{k_1^*} \exp(b_{j_1}^n) \left[ \exp(-\|\boldsymbol{a}_{j_1}^n - \boldsymbol{x}\|) p_{j_1}^{LL,n}(y|\boldsymbol{x}) - \exp(-\|\boldsymbol{a}_{j_1}^* - \boldsymbol{x}\|) p_{j_1}^{LL,*}(y|\boldsymbol{x}) \right]
$$

$$
- \sum_{j_1=1}^{k_1^*} \exp(b_{j_1}^n) \left[ \exp(-\|\boldsymbol{a}_{j_1}^n - \boldsymbol{x}\|) - \exp(-\|\boldsymbol{a}_{j_1}^* - \boldsymbol{x}\|) \right] p_{G_n}^{LL}(y|\boldsymbol{x})
$$

$$
+ \sum_{j_1=1}^{k_1^*} \left( \exp(b_{j_1}^n) - \exp(b_{j_1}^*) \right) \exp(-\|\boldsymbol{a}_{j_1}^* - \boldsymbol{x}\|) \left[ p_{j_1}^{LL,n}(y|\boldsymbol{x}) - p_{G_n}^{LL}(y|\boldsymbol{x}) \right]
$$

$$
:= A_n - B_n + C_n. \tag{47}
$$

**Step 1A - Decompose $A_n$:** We continue to decompose $A_n$:

$$
A_n := \sum_{j_1=1}^{k_1^*} \frac{\exp(b_{j_1}^n)}{\sum_{j_2'=1}^{k_2^*} \exp(-\|\boldsymbol{\omega}_{j_2'|j_1}^* - \boldsymbol{x}\| + \beta_{j_2'|j_1}^*)} [A_{n,j_1,1} + A_{n,j_1,2} + A_{n,j_1,3}],
$$

in which

$$
A_{n,j_1,1} := \sum_{j_2=1}^{k_2^*} \sum_{i_2 \in \mathcal{V}_{j_2|j_1}} \exp(\beta_{i_2|j_1}^n) \Big[ \exp(-\|\boldsymbol{\omega}_{i_2|j_1}^n - \boldsymbol{x}\|) \exp(-\|\boldsymbol{a}_{j_1}^n - \boldsymbol{x}\|) \pi(y|(\boldsymbol{\eta}_{j_1 i_2}^n)^\top \boldsymbol{x} + \tau_{j_1 i_2}^n, \nu_{j_1 i_2}^n)
$$

$$
- \exp(-\|\boldsymbol{\omega}_{j_2|j_1}^* - \boldsymbol{x}\|) \exp(-\|\boldsymbol{a}_{j_1}^* - \boldsymbol{x}\|) \pi(y|(\boldsymbol{\eta}_{j_1 j_2}^*)^\top \boldsymbol{x} + \tau_{j_1 j_2}^*, \nu_{j_1 j_2}^*) \Big],
$$

$$
A_{n,j_1,2} := \sum_{j_2=1}^{k_2^*} \sum_{i_2 \in \mathcal{V}_{j_2|j_1}} \exp(\beta_{i_2|j_1}^n) \Big[ \exp(-\|\boldsymbol{\omega}_{i_2|j_1}^n - \boldsymbol{x}\|) - \exp(-\|\boldsymbol{\omega}_{j_2|j_1}^* - \boldsymbol{x}\|) \Big]
$$

$$
\times \exp(-\|\boldsymbol{a}_{j_1}^n - \boldsymbol{x}\|) p_{j_1}^{LL,n}(y|\boldsymbol{x}),
$$

$$
A_{n,j_1,3} := \sum_{j_2=1}^{k_2^*} \Big( \sum_{i_2 \in \mathcal{V}_{j_2|j_1}} \exp(\beta_{i_2|j_1}^n) - \exp(\beta_{j_2|j_1}^*) \Big) \exp(-\|\boldsymbol{\omega}_{j_2|j_1}^* - \boldsymbol{x}\|)
$$

$$
\times [\exp(-\|\boldsymbol{a}_{j_1}^* - \boldsymbol{x}\|) \pi(y|(\boldsymbol{\eta}_{j_1 j_2}^*)^\top \boldsymbol{x} + \tau_{j_1 j_2}^*, \nu_{j_1 j_2}^*) - \exp(-\|\boldsymbol{a}_{j_1}^n - \boldsymbol{x}\|) p_{j_1}^{LL,n}(y|\boldsymbol{x})].
$$

Firstly, we separate the term $A_{n,j_1,1}$ into two parts based on the cardinality of the Voronoi cells $\mathcal{V}_{j_2|j_1}$ as

$$
A_{n,j_1,1} = \sum_{j_2:|\mathcal{V}_{j_2|j_1}|=1} \sum_{i_2 \in \mathcal{V}_{j_2|j_1}} \exp(\beta_{i_2|j_1}^n) \Big[ \exp(-\|\boldsymbol{\omega}_{i_2|j_1}^n - \boldsymbol{x}\|) \exp(-\|\boldsymbol{a}_{j_1}^n - \boldsymbol{x}\|) \pi(y|(\boldsymbol{\eta}_{j_1 i_2}^n)^\top \boldsymbol{x} + \tau_{j_1 i_2}^n, \nu_{j_1 i_2}^n)
$$

$$
- \exp(-\|\boldsymbol{\omega}_{j_2|j_1}^* - \boldsymbol{x}\|) \exp(-\|\boldsymbol{a}_{j_1}^* - \boldsymbol{x}\|) \pi(y|(\boldsymbol{\eta}_{j_1 j_2}^*)^\top \boldsymbol{x} + \tau_{j_1 j_2}^*, \nu_{j_1 j_2}^*) \Big],
$$

$$
+ \sum_{j_2:|\mathcal{V}_{j_2|j_1}|>1} \sum_{i_2 \in \mathcal{V}_{j_2|j_1}} \exp(\beta_{i_2|j_1}^n) \Big[ \exp(-\|\boldsymbol{\omega}_{i_2|j_1}^n - \boldsymbol{x}\|) \exp(-\|\boldsymbol{a}_{j_1}^n - \boldsymbol{x}\|) \pi(y|(\boldsymbol{\eta}_{j_1 i_2}^n)^\top \boldsymbol{x} + \tau_{j_1 i_2}^n, \nu_{j_1 i_2}^n)
$$

$$
- \exp(-\|\boldsymbol{\omega}_{j_2|j_1}^* - \boldsymbol{x}\|) \exp(-\|\boldsymbol{a}_{j_1}^* - \boldsymbol{x}\|) \pi(y|(\boldsymbol{\eta}_{j_1 j_2}^*)^\top \boldsymbol{x} + \tau_{j_1 j_2}^*, \nu_{j_1 j_2}^*) \Big]
$$

$$
:= A_{n,j_1,1,1} + A_{n,j_1,1,2}.
$$

By denoting $F(\boldsymbol{x}; \boldsymbol{\omega}) := \exp(-\|\boldsymbol{\omega} - \boldsymbol{x}\|)$ and employing the first-order Taylor expansion, we can represent $A_{n,j_1,1,1}$ as

$$
A_{n,j_1,1,1} = \sum_{j_2:|\mathcal{V}_{j_2|j_1}|=1} \sum_{i_2 \in \mathcal{V}_{j_2|j_1}} \sum_{|\boldsymbol{\alpha}|=1} \frac{\exp(\beta_{i_2|j_1}^n)}{2^{\alpha_5}! \boldsymbol{\alpha}!} (\Delta \boldsymbol{\omega}_{i_2 j_2|j_1}^n)^{\boldsymbol{\alpha}_1} (\Delta \boldsymbol{a}_{j_1}^n)^{\boldsymbol{\alpha}_2} (\Delta \boldsymbol{\eta}_{j_1 i_2 j_2}^n)^{\boldsymbol{\alpha}_3} (\Delta \tau_{j_1 i_2 j_2}^n)^{\alpha_4}
$$

$$
\times (\Delta \nu_{j_1 i_2 j_2}^n)^{\alpha_5} \boldsymbol{x}^{\boldsymbol{\alpha}_3} \frac{\partial^{|\boldsymbol{\alpha}_1|} F}{\partial \boldsymbol{\omega}^{\boldsymbol{\alpha}_1}}(\boldsymbol{x}; \boldsymbol{\omega}_{j_2|j_1}^*) \frac{\partial^{|\boldsymbol{\alpha}_2|} F}{\partial \boldsymbol{a}^{\boldsymbol{\alpha}_2}}(\boldsymbol{x}; \boldsymbol{a}_{j_1}^*) \frac{\partial^{|\boldsymbol{\alpha}_3|+\alpha_4+2\alpha_5} \pi}{\partial \xi^{|\boldsymbol{\alpha}_3|+\alpha_4+2\alpha_5}}(y|(\boldsymbol{\eta}_{j_1 j_2}^*)^\top \boldsymbol{x} + \tau_{j_1 j_2}^*, \nu_{j_1 j_2}^*) + R_{n,1,1}(\boldsymbol{x})
$$

$$
= \sum_{j_2:|\mathcal{V}_{j_2|j_1}|=1} \sum_{|\boldsymbol{\alpha}_1|+|\boldsymbol{\alpha}_2|+|\boldsymbol{\alpha}_3|=0}^{1} \sum_{\rho=0 \vee 1-|\boldsymbol{\alpha}_1|-|\boldsymbol{\alpha}_2|-|\boldsymbol{\alpha}_3|}^{2(1-|\boldsymbol{\alpha}_1|-|\boldsymbol{\alpha}_2|-|\boldsymbol{\alpha}_3|)} S_{n,j_2|j_1,\boldsymbol{\alpha}_1,\boldsymbol{\alpha}_2,\boldsymbol{\alpha}_3,\rho} \cdot \boldsymbol{x}^{\boldsymbol{\alpha}_3} \frac{\partial^{|\boldsymbol{\alpha}_1|} F}{\partial \boldsymbol{\omega}^{\boldsymbol{\alpha}_1}}(\boldsymbol{x}; \boldsymbol{\omega}_{j_2|j_1}^*)
$$

$$
\times \frac{\partial^{|\boldsymbol{\alpha}_2|} F}{\partial \boldsymbol{a}^{\boldsymbol{\alpha}_2}}(\boldsymbol{x}; \boldsymbol{a}_{j_1}^*) \frac{\partial^{|\boldsymbol{\alpha}_3|+\rho} \pi}{\partial \xi^{|\boldsymbol{\alpha}_3|+\rho}}(y|(\boldsymbol{\eta}_{j_1 j_2}^*)^\top \boldsymbol{x} + \tau_{j_1 j_2}^*, \nu_{j_1 j_2}^*) + R_{n,1,1}(\boldsymbol{x}),
$$

where $R_{n,1,1}(\boldsymbol{x}, y)$ is a Taylor remainder such that $R_{n,1,1}(\boldsymbol{x}, y)/\mathcal{L}_{3n} \to 0$ as $n \to \infty$, and

$$
S_{n,j_2|j_1,\boldsymbol{\alpha}_1,\boldsymbol{\alpha}_2,\boldsymbol{\alpha}_3,\rho} := \sum_{i_2 \in \mathcal{V}_{j_2|j_1}} \sum_{\alpha_4+2\alpha_5=\rho} \frac{\exp(\beta_{i_2|j_1}^n)}{2^{\alpha_5} \boldsymbol{\alpha}!} (\Delta \boldsymbol{\omega}_{i_2 j_2|j_1}^n)^{\boldsymbol{\alpha}_1} (\Delta \boldsymbol{a}_{j_1}^n)^{\boldsymbol{\alpha}_2} (\Delta \boldsymbol{\eta}_{j_1 i_2 j_2}^n)^{\boldsymbol{\alpha}_3}
$$

$$
\times (\Delta \tau_{j_1 i_2 j_2}^n)^{\alpha_4} (\Delta \nu_{j_1 i_2 j_2}^n)^{\alpha_5},
$$

for any $(\boldsymbol{\alpha}_1, \boldsymbol{\alpha}_2, \boldsymbol{\alpha}_3, \rho) \neq (\boldsymbol{0}_d, \boldsymbol{0}_d, \boldsymbol{0}_d, 0)$, $j_1 \in [k_1^*]$ and $j_2 \in [k_2^*]$.

For each $(j_1, j_2) \in [k_1^*] \times [k_2^*]$, by invoking the Taylor expansion of order $r^{LL}(|\mathcal{V}_{j_2|j_1}|) := r_{j_2|j_1}^{LL}$, the term $A_{n,j_1,1,2}$ can be represented as

$$A_{n,j_1,1,2} = \sum_{j_2:|\mathcal{V}_{j_2|j_1}|>1} \sum_{|\boldsymbol{\alpha}_1|+|\boldsymbol{\alpha}_2|+|\boldsymbol{\alpha}_3|=0}^{r_{j_2|j_1}^{LL}} \sum_{\rho=0\vee 1-|\boldsymbol{\alpha}_1|-|\boldsymbol{\alpha}_2|-|\boldsymbol{\alpha}_3|}^{2(r_{j_2|j_1}^{LL}-|\boldsymbol{\alpha}_1|-|\boldsymbol{\alpha}_2|-|\boldsymbol{\alpha}_3|)} S_{n,j_2|j_1,\boldsymbol{\alpha}_1,\boldsymbol{\alpha}_2,\boldsymbol{\alpha}_3,\rho} \cdot \boldsymbol{x}^{\boldsymbol{\alpha}_3}$$

$$\times \frac{\partial^{|\boldsymbol{\alpha}_1|}F}{\partial \boldsymbol{\omega}^{\boldsymbol{\alpha}_1}}(\boldsymbol{x}; \boldsymbol{\omega}_{j_2|j_1}^*) \frac{\partial^{|\boldsymbol{\alpha}_2|}F}{\partial \boldsymbol{a}^{\boldsymbol{\alpha}_2}}(\boldsymbol{x}; \boldsymbol{a}_{j_1}^*) \frac{\partial^{|\boldsymbol{\alpha}_3|+\rho}\pi}{\partial \xi^{|\boldsymbol{\alpha}_3|+\rho}}(y|(\boldsymbol{\eta}_{j_1 j_2}^*)^\top \boldsymbol{x} + \tau_{j_1 j_2}^*, \nu_{j_1 j_2}^*) + R_{n,1,2}(\boldsymbol{x}, y),$$

where $R_{n,1,2}(\boldsymbol{x}, y)$ is a Taylor remainder such that $R_{n,1,2}(\boldsymbol{x}, y)/\mathcal{L}_{3n} \to 0$ as $n \to \infty$.

Secondly, we rewrite the term $A_{n,j_1,2}$ as follows:

$$\sum_{j_2:|\mathcal{V}_{j_2|j_1}|=1} \sum_{i_2 \in \mathcal{V}_{j_2|j_1}} \exp(\beta_{i_2|j_1}^n) \Big[ \exp(-\|\boldsymbol{\omega}_{i_2|j_1}^n - \boldsymbol{x}\|) - \exp(-\|\boldsymbol{\omega}_{j_2|j_1}^* - \boldsymbol{x}\|) \Big] \exp(-\|\boldsymbol{a}_{j_1}^n - \boldsymbol{x}\|) p_{j_1}^{LL,n}(y|\boldsymbol{x})$$

$$+ \sum_{j_2:|\mathcal{V}_{j_2|j_1}|>1} \sum_{i_2 \in \mathcal{V}_{j_2|j_1}} \exp(\beta_{i_2|j_1}^n) \Big[ \exp(-\|\boldsymbol{\omega}_{i_2|j_1}^n - \boldsymbol{x}\|) - \exp(-\|\boldsymbol{\omega}_{j_2|j_1}^* - \boldsymbol{x}\|) \Big] \exp(-\|\boldsymbol{a}_{j_1}^n - \boldsymbol{x}\|) p_{j_1}^{LL,n}(y|\boldsymbol{x})$$

$$:= A_{n,j_1,2,1} + A_{n,j_1,2,2}.$$

According to the first-order Taylor expansion, we have

$$A_{n,j_1,2,1} = \sum_{j_2:|\mathcal{V}_{j_2|j_1}|=1} \sum_{i_2 \in \mathcal{V}_{j_2|j_1}} \sum_{|\boldsymbol{\psi}|=1} \frac{\exp(\beta_{i_2|j_1}^n)}{\boldsymbol{\psi}!} (\Delta \boldsymbol{\omega}_{i_2 j_2|j_1}^n)^{\boldsymbol{\psi}}$$

$$\times \frac{\partial^{|\boldsymbol{\psi}|}F}{\partial \boldsymbol{\omega}^{\boldsymbol{\psi}}}(\boldsymbol{x}; \boldsymbol{\omega}_{j_2|j_1}^*) \exp(-\|\boldsymbol{a}_{j_1}^n - \boldsymbol{x}\|) p_{j_1}^{LL,n}(y|\boldsymbol{x}) + R_{n,2,1}(\boldsymbol{x}, y),$$

$$= \sum_{j_2:|\mathcal{V}_{j_2|j_1}|=1} \sum_{|\boldsymbol{\psi}|=1} T_{n,j_2|j_1,\boldsymbol{\psi}} \cdot \frac{\partial^{|\boldsymbol{\psi}|}F}{\partial \boldsymbol{\omega}^{\boldsymbol{\psi}}}(\boldsymbol{x}; \boldsymbol{\omega}_{j_2|j_1}^*) \exp(-\|\boldsymbol{a}_{j_1}^n - \boldsymbol{x}\|) p_{j_1}^{LL,n}(y|\boldsymbol{x}) + R_{n,2,1}(\boldsymbol{x}, y),$$

where $R_{n,2,1}(\boldsymbol{x}, y)$ is a Taylor remainder such that $R_{n,2,1}(\boldsymbol{x}, y)/\mathcal{L}_{3n} \to 0$ as $n \to \infty$, and

$$T_{n,j_2|j_1,\boldsymbol{\psi}} := \sum_{i_2 \in \mathcal{V}_{j_2|j_1}} \frac{\exp(\beta_{i_2|j_1}^n)}{\boldsymbol{\psi}!} (\Delta \boldsymbol{\omega}_{i_2 j_2|j_1}^n)^{\boldsymbol{\psi}},$$

for any $j_2 \in [k_2^*]$ and $\boldsymbol{\psi} \neq \boldsymbol{0}_d$.

Meanwhile, we apply the second-order Taylor expansion to $A_{n,j_1,2,2}$:

$$A_{n,j_1,2,2} = \sum_{j_2:|\mathcal{V}_{j_2|j_1}|>1} \sum_{|\boldsymbol{\psi}|=1}^{2} T_{n,j_2|j_1,\boldsymbol{\psi}} \cdot \frac{\partial^{|\boldsymbol{\psi}|}F}{\partial \boldsymbol{\omega}^{\boldsymbol{\psi}}}(\boldsymbol{x}; \boldsymbol{\omega}_{j_2|j_1}^*) \exp(-\|\boldsymbol{a}_{j_1}^n - \boldsymbol{x}\|) p_{j_1}^{LL,n}(y|\boldsymbol{x}) + R_{n,2,2}(\boldsymbol{x}, y),$$

where $R_{n,2,2}(\boldsymbol{x}, y)$ is a Taylor remainder such that $R_{n,2,2}(\boldsymbol{x}, y)/\mathcal{L}_{3n} \to 0$ as $n \to \infty$.

Combine the above results together, we can illustrate the term $A_n$ as

$$A_n = \sum_{j_1=1}^{k_1^*} \sum_{j_2=1}^{k_2^*} \frac{\exp(b_{j_1}^n)}{\sum_{j_2'=1}^{k_2^*} \exp(-\|\boldsymbol{\omega}_{j_2'|j_1}^* - \boldsymbol{x}\| + \beta_{j_2'|j_1}^*)} \Bigg[ \sum_{|\boldsymbol{\alpha}_1|+|\boldsymbol{\alpha}_2|+|\boldsymbol{\alpha}_3|=0}^{r_{j_2|j_1}^{LL}} \sum_{\rho=0\vee 1-|\boldsymbol{\alpha}_1|-|\boldsymbol{\alpha}_2|-|\boldsymbol{\alpha}_3|}^{2(r_{j_2|j_1}^{LL}-|\boldsymbol{\alpha}_1|-|\boldsymbol{\alpha}_2|-|\boldsymbol{\alpha}_3|)} S_{n,j_2|j_1,\boldsymbol{\alpha}_1,\boldsymbol{\alpha}_2,\boldsymbol{\alpha}_3,\rho}$$

$$\times \boldsymbol{x}^{\boldsymbol{\alpha}_3} \frac{\partial^{|\boldsymbol{\alpha}_1|}F}{\partial \boldsymbol{\omega}^{\boldsymbol{\alpha}_1}}(\boldsymbol{x}; \boldsymbol{\omega}_{j_2|j_1}^*) \frac{\partial^{|\boldsymbol{\alpha}_2|}F}{\partial \boldsymbol{a}^{\boldsymbol{\alpha}_2}}(\boldsymbol{x}; \boldsymbol{a}_{j_1}^*) \frac{\partial^{|\boldsymbol{\alpha}_3|+\rho}\pi}{\partial \xi^{|\boldsymbol{\alpha}_3|+\rho}}(y|(\boldsymbol{\eta}_{j_1 j_2}^*)^\top \boldsymbol{x} + \tau_{j_1 j_2}^*, \nu_{j_1 j_2}^*) + R_{n,1,1}(\boldsymbol{x}, y) + R_{n,1,2}(\boldsymbol{x}, y)$$

$$- \sum_{|\boldsymbol{\psi}|=0}^{2} T_{n,j_2|j_1,\boldsymbol{\psi}} \cdot \frac{\partial^{|\boldsymbol{\psi}|}F}{\partial \boldsymbol{\omega}^{\boldsymbol{\psi}}}(\boldsymbol{x}; \boldsymbol{\omega}_{j_2|j_1}^*) \exp(-\|\boldsymbol{a}_{j_1}^n - \boldsymbol{x}\|) p_{j_1}^{LL,n}(y|\boldsymbol{x}) - R_{n,2,1}(\boldsymbol{x}, y) - R_{n,2,2}(\boldsymbol{x}, y) \Bigg],$$

$$(48)$$

where $S_{n,j_2|j_1,\boldsymbol{\alpha}_1,\boldsymbol{\alpha}_2,\boldsymbol{\alpha}_3,\rho} = T_{n,j_2|j_1,\boldsymbol{\psi}} = \sum_{i_2 \in \mathcal{V}_{j_2|j_1}} \exp(\beta_{i_2|j_1}^n) - \exp(\beta_{j_2|j_1}^*)$ for any $j_1 \in [k_1^*]$, $j_2 \in [k_2^*]$, $(\boldsymbol{\alpha}_1, \boldsymbol{\alpha}_2, \boldsymbol{\alpha}_3, \rho) = (\boldsymbol{0}_d, \boldsymbol{0}_d, \boldsymbol{0}_d, 0)$ and $\boldsymbol{\psi} = \boldsymbol{0}_d$.

**Step 1B - Decompose $B_n$:** By invoking the first-order Taylor expansion, we decompose the term $B_n$ defined in equation (47) as

$$B_n = \sum_{j_1=1}^{k_1^*} \exp(b_{j_1}^n) \sum_{|\boldsymbol{\gamma}|=1} (\Delta \boldsymbol{a}_{j_1}^n)^{\boldsymbol{\gamma}} \cdot \frac{\partial^{|\boldsymbol{\gamma}|} F}{\partial \boldsymbol{a}^{\boldsymbol{\gamma}}}(\boldsymbol{x}; \boldsymbol{a}_{j_1}^*) p_{G_n}^{LL}(y|\boldsymbol{x}) + R_{n,3}(\boldsymbol{x}, y) \qquad (49)$$

where $R_{n,3}(\boldsymbol{x}, y)$ is a Taylor remainder such that $R_{n,3}(\boldsymbol{x}, y)/\mathcal{L}_{3n} \to 0$ as $n \to \infty$.

Putting the decomposition in equations (47), (48) and (49) together, we realize that $A_n$, $B_n$ and $C_n$ can be treated as a linear combination of elements from the following set union:

$$\left\{ \frac{\boldsymbol{x}^{\boldsymbol{\alpha}_3} \frac{\partial^{|\boldsymbol{\alpha}_1|} F}{\partial \boldsymbol{\omega}^{\boldsymbol{\alpha}_1}}(\boldsymbol{x}; \boldsymbol{\omega}_{j_2|j_1}^*) \frac{\partial^{|\boldsymbol{\alpha}_2|} F}{\partial \boldsymbol{a}^{\boldsymbol{\alpha}_2}}(\boldsymbol{x}; \boldsymbol{a}_{j_1}^*) \frac{\partial^{|\boldsymbol{\alpha}_3|+\rho} \pi}{\partial \xi^{|\boldsymbol{\alpha}_3|+\rho}}(y|(\boldsymbol{\eta}_{j_1 j_2}^*)^\top \boldsymbol{x} + \tau_{j_1 j_2}^*, \nu_{j_1 j_2}^*)}{\sum_{j_2'=1}^{k_2^*} \exp(-\|\boldsymbol{\omega}_{j_2'|j_1}^* - \boldsymbol{x}\| + \beta_{j_2'|j_1}^*)} : j_1 \in [k_1^*], \ j_2 \in [k_2^*], \right.$$

$$\left. 0 \le |\boldsymbol{\alpha}_1| + |\boldsymbol{\alpha}_2| + |\boldsymbol{\alpha}_3| \le 2 r_{j_2|j_1}^{LL}, 0 \le \rho \le 2(r_{j_2|j_1}^{LL} - |\boldsymbol{\alpha}_1| - |\boldsymbol{\alpha}_2| - |\boldsymbol{\alpha}_3|) \right\}$$

$$\cup \left\{ \frac{\frac{\partial^{|\boldsymbol{\psi}|} F}{\partial \boldsymbol{\omega}^{\boldsymbol{\psi}}}(\boldsymbol{x}; \boldsymbol{\omega}_{j_2|j_1}^*) \exp(-\|\boldsymbol{a}_{j_1}^n - \boldsymbol{x}\|) p_{j_1}^{LL,n}(y|\boldsymbol{x})}{\sum_{j_2'=1}^{k_2^*} \exp(-\|\boldsymbol{\omega}_{j_2'|j_1}^* - \boldsymbol{x}\| + \beta_{j_2'|j_1}^*)} : j_1 \in [k_1^*], \ j_2 \in [k_2^*], \ 0 \le |\boldsymbol{\psi}| \le 2 \right\}$$

$$\cup \left\{ \frac{\partial^{|\boldsymbol{\gamma}|} F}{\partial \boldsymbol{a}^{\boldsymbol{\gamma}}}(\boldsymbol{x}; \boldsymbol{a}_{j_1}^*) p_{j_1}^{LL,n}(y|\boldsymbol{x}), \ \frac{\partial^{|\boldsymbol{\gamma}|} F}{\partial \boldsymbol{a}^{\boldsymbol{\gamma}}}(\boldsymbol{x}; \boldsymbol{a}_{j_1}^*) p_{G_n}^{LL}(y|\boldsymbol{x}) : j_1 \in [k_1^*], \ 0 \le |\boldsymbol{\gamma}| \le 1 \right\}.$$

**Step 2 - Non-vanishing coefficients:** In this step, we demonstrate that not all the coefficients in the representation of $A_n/\mathcal{L}_{3n}$, $B_n/\mathcal{L}_{3n}$ and $C_n/\mathcal{L}_{3n}$ converge to zero as $n \to \infty$. Assume by contrary that all of them go to zero. Then, we look into the coefficients associated with the term

- $\exp(-\|\boldsymbol{a}_{j_1}^* - \boldsymbol{x}\|) p_{j_1}^{LL,n}(y|\boldsymbol{x})$ in $C_n/\mathcal{L}_{3n}$, we have

$$\frac{1}{\mathcal{L}_{3n}} \cdot \sum_{j_1=1}^{k_1^*} \left| \exp(b_{j_1}^n) - \exp(b_{j_1}^*) \right| \to 0. \qquad (50)$$

- $\dfrac{F(\boldsymbol{x}; \boldsymbol{\omega}_{j_2|j_1}^*) F(\boldsymbol{x}; \boldsymbol{a}_{j_1}^*) \pi(y|(\boldsymbol{\eta}_{j_1 j_2}^*)^\top \boldsymbol{x} + \tau_{j_1 j_2}^*, \nu_{j_1 j_2}^*)}{\sum_{j_2'=1}^{k_2^*} \exp(-\|\boldsymbol{\omega}_{j_2'|j_1}^* - \boldsymbol{x}\| + \beta_{j_2'|j_1}^*)}$ in $A_n/\mathcal{L}_{3n}$, we get that

$$\frac{1}{\mathcal{L}_{3n}} \cdot \sum_{j_1=1}^{k_1^*} \exp(b_{j_1}^n) \sum_{j_2=1}^{k_2^*} \left| \sum_{i_2 \in \mathcal{V}_{j_2|j_1}} \exp(\beta_{i_2|j_1}^n) - \exp(\beta_{j_2|j_1}^*) \right| \to 0. \qquad (51)$$

- $\dfrac{\frac{\partial^{|\boldsymbol{\alpha}_1|} F}{\partial \boldsymbol{\omega}^{\boldsymbol{\alpha}_1}}(\boldsymbol{x}; \boldsymbol{\omega}_{j_2|j_1}^*) F(\boldsymbol{x}; \boldsymbol{a}_{j_1}^*) \pi(y|(\boldsymbol{\eta}_{j_1 j_2}^*)^\top \boldsymbol{x} + \tau_{j_1 j_2}^*, \nu_{j_1 j_2}^*)}{\sum_{j_2'=1}^{k_2^*} \exp(-\|\boldsymbol{\omega}_{j_2'|j_1}^* - \boldsymbol{x}\| + \beta_{j_2'|j_1}^*)}$ in $A_n/\mathcal{L}_{3n}$ for $j_1 \in [k_1^*], j_2 \in$
  $[k_2^*] : |\mathcal{V}_{j_2|j_1}| = 1$ and $\boldsymbol{\alpha}_1 = e_{d,u}$ where $e_{d,u} := (0, \ldots, 0, \underbrace{1}_{u\text{-}th}, 0, \ldots, 0) \in \mathbb{N}^d$, we receive
  that

$$\frac{1}{\mathcal{L}_{3n}} \cdot \sum_{j_1=1}^{k_1^*} \exp(b_{j_1}^n) \sum_{j_2 \in [k_2^*]: |\mathcal{V}_{j_2|j_1}|=1} \sum_{i_2 \in \mathcal{V}_{j_2|j_1}} \exp(\beta_{i_2|j_1}^n) \|\boldsymbol{\omega}_{i_2|j_1}^n - \boldsymbol{\omega}_{j_2|j_1}^*\|_1 \to 0.$$

  Note that since the norm-1 is equivalent to the norm-2, then we can replace the norm-1 with the norm-2, that is,

$$\frac{1}{\mathcal{L}_{3n}} \cdot \sum_{j_1=1}^{k_1^*} \exp(b_{j_1}^n) \sum_{j_2 \in [k_2^*]: |\mathcal{V}_{j_2|j_1}|=1} \sum_{i_2 \in \mathcal{V}_{j_2|j_1}} \exp(\beta_{i_2|j_1}^n) \|\boldsymbol{\omega}_{i_2|j_1}^n - \boldsymbol{\omega}_{j_2|j_1}^*\| \to 0. \qquad (52)$$

- $\boldsymbol{x}^{\boldsymbol{\alpha}_3} \dfrac{F(\boldsymbol{x}; \boldsymbol{\omega}^*_{j_2|j_1}) F(\boldsymbol{x}; \boldsymbol{a}^*_{j_1}) \frac{\partial^{|\boldsymbol{\alpha}_3|} \pi}{\partial \xi^{|\boldsymbol{\alpha}_3|}} (y|(\boldsymbol{\eta}^*_{j_1 j_2})^\top \boldsymbol{x} + \tau^*_{j_1 j_2}, \nu^*_{j_1 j_2})}{\sum_{j'_2=1}^{k^*_2} \exp(-\|\boldsymbol{\omega}^*_{j'_2|j_1} - \boldsymbol{x}\| + \beta^*_{j'_2|j_1})}$ in $A_n/\mathcal{L}_{3n}$ for $j_1 \in [k^*_1], j_2 \in [k^*_2] : |\mathcal{V}_{j_2|j_1}| = 1$ and $\boldsymbol{\alpha}_3 = e_{d,u}$, we have that

$$\frac{1}{\mathcal{L}_{3n}} \cdot \sum_{j_1=1}^{k^*_1} \exp(b^n_{j_1}) \sum_{j_2 \in [k^*_2] : |\mathcal{V}_{j_2|j_1}|=1} \sum_{i_2 \in \mathcal{V}_{j_2|j_1}} \exp(\beta^n_{j_2|j_1}) \|\boldsymbol{\eta}^n_{j_1 i_2} - \boldsymbol{\eta}^*_{j_1 j_2}\| \to 0. \quad (53)$$

- $\dfrac{\partial^{|\boldsymbol{\gamma}|} F}{\partial \boldsymbol{a}^{\boldsymbol{\gamma}}}(\boldsymbol{x}; \boldsymbol{a}^*_{j_1}) p^{LL}_{G_n}(y|\boldsymbol{x})$ in $B_n/\mathcal{L}_{3n}$ for $j_1 \in [k^*_1]$ and $\boldsymbol{\gamma} = e_{d,u}$, we obtain

$$\frac{1}{\mathcal{L}_{3n}} \cdot \sum_{j_1=1}^{k^*_1} \exp(b^n_{j_1}) \|\boldsymbol{a}^n_{j_1} - \boldsymbol{a}^*_{j_1}\| \to 0. \quad (54)$$

- $\dfrac{\frac{\partial^{|\boldsymbol{\alpha}_1|} F}{\partial \boldsymbol{\omega}^{\boldsymbol{\alpha}_1}}(\boldsymbol{x}; \boldsymbol{\omega}^*_{j_2|j_1}) F(\boldsymbol{x}; \boldsymbol{a}^*_{j_1}) \pi(y|(\boldsymbol{\eta}^*_{j_1 j_2})^\top \boldsymbol{x} + \tau^*_{j_1 j_2}, \nu^*_{j_1 j_2})}{\sum_{j'_2=1}^{k^*_2} \exp(-\|\boldsymbol{\omega}^*_{j'_2|j_1} - \boldsymbol{x}\| + \beta^*_{j'_2|j_1})}$ in $A_n/\mathcal{L}_{3n}$ for $j_1 \in [k^*_1], j_2 \in [k^*_2] : |\mathcal{V}_{j_2|j_1}| > 1$ and $\boldsymbol{\alpha}_1 = 2e_{d,u}$, we receive that

$$\frac{1}{\mathcal{L}_{3n}} \cdot \sum_{j_1=1}^{k^*_1} \exp(b^n_{j_1}) \sum_{j_2 \in [k^*_2] : |\mathcal{V}_{j_2|j_1}|>1} \sum_{i_2 \in \mathcal{V}_{j_2|j_1}} \exp(\beta^n_{i_2|j_1}) \|\boldsymbol{\omega}^n_{i_2|j_1} - \boldsymbol{\omega}^*_{j_2|j_1}\|^2 \to 0. \quad (55)$$

- $\dfrac{\boldsymbol{x}^{\boldsymbol{\alpha}_3} F(\boldsymbol{x}; \boldsymbol{\omega}^*_{j_2|j_1}) F(\boldsymbol{x}; \boldsymbol{a}^*_{j_1}) \frac{\partial^{|\boldsymbol{\alpha}_3|} \pi}{\partial \xi^{|\boldsymbol{\alpha}_3|}}(y|(\boldsymbol{\eta}^*_{j_1 j_2})^\top \boldsymbol{x} + \tau^*_{j_1 j_2}, \nu^*_{j_1 j_2})}{\sum_{j'_2=1}^{k^*_2} \exp(-\|\boldsymbol{\omega}^*_{j'_2|j_1} - \boldsymbol{x}\| + \beta^*_{j'_2|j_1})}$ in $A_n/\mathcal{L}_{3n}$ for $j_1 \in [k^*_1], j_2 \in [k^*_2] : |\mathcal{V}_{j_2|j_1}| > 1$ and $\boldsymbol{\alpha}_3 = 2e_{d,u}$, we have that

$$\frac{1}{\mathcal{L}_{3n}} \cdot \sum_{j_1=1}^{k^*_1} \exp(b^n_{j_1}) \sum_{j_2 \in [k^*_2] : |\mathcal{V}_{j_2|j_1}|>1} \sum_{i_2 \in \mathcal{V}_{j_2|j_1}} \exp(\beta^n_{i_2|j_1}) \|\boldsymbol{\eta}^n_{j_1 i_2} - \boldsymbol{\eta}^*_{j_1 j_2}\|^2 \to 0. \quad (56)$$

Combine the above limits and the formulation of the loss $\mathcal{L}_{3n}$ in equation (46), we deduce that

$$\frac{1}{\mathcal{L}_{3n}} \cdot \sum_{j_1=1}^{k^*_1} \exp(b^n_{j_1}) \sum_{j_2 : |\mathcal{V}_{j_2|j_1}|>1} \sum_{i_2 \in \mathcal{V}_{j_2|j_1}} \exp(\beta^n_{i_2|j_1}) \Big( |\Delta\tau^n_{j_1 i_2 j_2}|^{r^{LL}_{j_2|j_1}} + |\Delta\nu^n_{j_1 i_2 j_2}|^{\frac{r^{LL}_{j_2|j_1}}{2}} \Big) \not\to 0.$$

This indicates that there exist indices $j^*_1 \in [k^*_1]$ and $j^*_2 \in [k^*_2] : |\mathcal{V}_{j^*_2|j^*_1}| > 1$ such that

$$\frac{1}{\mathcal{L}_{3n}} \cdot \sum_{i_2 \in \mathcal{V}_{j^*_2|j^*_1}} \exp(\beta^n_{i_2|j^*_1}) \Big( |\Delta\tau^n_{j^*_1 i_2 j^*_2}|^{r^{LL}_{j^*_2|j^*_1}} + |\Delta\nu^n_{j^*_1 i_2 j^*_2}|^{\frac{r^{LL}_{j^*_2|j^*_1}}{2}} \Big) \not\to 0. \quad (57)$$

WLOG, we may assume that $j^*_1 = j^*_2 = 1$. Then, considering the coefficients of the term $F(\boldsymbol{x}; \boldsymbol{\omega}^*_{j_2|j_1}) F(\boldsymbol{x}; \boldsymbol{a}^*_{j_1}) \frac{\partial^\rho \pi}{\partial \xi^\rho}(y|(\boldsymbol{\eta}^*_{j_1 j_2})^\top \boldsymbol{x} + \tau^*_{j_1 j_2}, \nu^*_{j_1 j_2})$ in $A_n/\mathcal{L}_{3n}$ where $j_1 = j_2 = 1$, we get $\exp(b^n_1) S_{n,1|1,\mathbf{0}_d,\mathbf{0}_d,\mathbf{0}_d,\rho}/\mathcal{L}_{3n} \to 0$, or equivalently,

$$\frac{1}{\mathcal{L}_{3n}} \cdot \sum_{i_2 \in \mathcal{V}_{1|1}} \sum_{\alpha_4 + 2\alpha_5 = \rho} \frac{\exp(\beta^n_{i_2|1})}{2^{\alpha_5} \alpha_4! \alpha_5!} \cdot (\Delta\tau^n_{1 i_2 1})^{\alpha_4} (\Delta\nu^n_{1 i_2 1})^{\alpha_5} \to 0. \quad (58)$$

Next, we divide the left hand side of equation (57) by that of equation (58), and get that

$$\frac{\sum_{i_2 \in \mathcal{V}_{1|1}} \sum_{\alpha_4 + 2\alpha_5 = \rho} \frac{\exp(\beta^n_{i_2|1})}{2^{\alpha_5} \alpha_4! \alpha_5!} \cdot (\Delta\tau^n_{1 i_2 1})^{\alpha_4} (\Delta\nu^n_{1 i_2 1})^{\alpha_5}}{\sum_{i_2 \in \mathcal{V}_{1|1}} \exp(\beta^n_{i_2|1}) \Big( |\Delta\tau^n_{1 i_2 1}|^{r^{LL}_{1|1}} + |\Delta\nu^n_{1 i_2 1}|^{\frac{r^{LL}_{1|1}}{2}} \Big)} \to 0. \quad (59)$$

Let us define $\overline{M}_n := \max\{\|\Delta\tau^n_{1i_21}\|, \|\Delta\nu^n_{1i_21}\|^{1/2} : i_2 \in \mathcal{V}_{1|1}\}$, and $\overline{\beta}_n := \max_{i_2 \in \mathcal{V}_{1|1}} \exp(\beta^n_{i_2|1})$. Since the sequence $\exp(\beta^n_{i_2|1})/\overline{\beta}_n$ is bounded, we can replace it by its subsequence which has a positive limit $p^2_{i_2} := \lim_{n\to\infty} \exp(\beta^n_{i_2|1})/\overline{\beta}_n$. Note that at least one among the limits $p^2_{i_2}$ must be equal to one. Next, let us define

$$(\Delta\tau^n_{1i_21})/\overline{M}_n \to q_{4i_2}, \quad (\Delta\nu^n_{1i_21})/2\overline{M}_n \to q_{5i_2}.$$

Note that at least one among $q_{4i_2}, q_{5i_2}$ must be equal to either 1 or $-1$.

By dividing both the numerator and the denominator of the term in equation (45) by $\overline{\beta}_n \overline{M}^\rho_n$, we obtain the system of polynomial equations:

$$\sum_{i_2 \in \mathcal{V}_{1|1}} \sum_{\alpha_4 + 2\alpha_5 = \rho} \frac{1}{\alpha_4! \alpha_5!} \cdot p^2_{i_2} q^{\alpha_4}_{4i_2} q^{\alpha_5}_{5i_2} = 0, \quad 1 \le \rho \le r^{LL}_{1|1}.$$

According to the definition of the term $r^{LL}_{1|1}$, the above system does not have any non-trivial solutions, which is a contradiction. Consequently, at least one among the coefficients in the representation of $A_n/\mathcal{L}_{3n}$, $B_n/\mathcal{L}_{3n}$ and $C_n/\mathcal{L}_{3n}$ must not approach zero as $n \to \infty$.

**Step 3 - Application of the Fatou's lemma.** In this stage, we show that all the coefficients in the formulations of $A_n/\mathcal{L}_{3n}$, $B_n/\mathcal{L}_{3n}$ and $C_n/\mathcal{L}_{3n}$ go to zero as $n \to \infty$. Denote by $m_n$ the maximum of the absolute values of those coefficients, the result from Step 2 induces that $1/m_n \not\to \infty$.

By employing the Fatou's lemma, we have

$$0 = \lim_{n\to\infty} \frac{\mathbb{E}_{\boldsymbol{X}}[V(p^{LL}_{G_n}(\cdot|\boldsymbol{X}), p^{LL}_{G_*}(\cdot|\boldsymbol{X}))]}{m_n \mathcal{L}_{3n}} \ge \int \liminf_{n\to\infty} \frac{|p^{LL}_{G_n}(y|\boldsymbol{x}) - p^{LL}_{G_*}(y|\boldsymbol{x})|}{2m_n \mathcal{L}_{3n}} \mathrm{d}(\boldsymbol{x}, y).$$

Thus, we deduce that

$$\frac{|p^{LL}_{G_n}(y|\boldsymbol{x}) - p^{LL}_{G_*}(y|\boldsymbol{x})|}{2m_n \mathcal{L}_{3n}} \to 0,$$

which results in $Q_n/[m_n \mathcal{L}_{3n}] \to 0$ as $n \to \infty$ for almost surely $(\boldsymbol{x}, y)$.

Next, we denote

$$\frac{\exp(b^n_{j_1}) S_{n,j_2|j_1,\boldsymbol{\alpha}_1,\boldsymbol{\alpha}_2,\boldsymbol{\alpha}_3,\rho}}{m_n \mathcal{L}_{3n}} \to \phi_{j_2|j_1,\boldsymbol{\alpha}_1,\boldsymbol{\alpha}_2,\boldsymbol{\alpha}_3,\rho}, \qquad \frac{\exp(b^n_{j_1}) T_{n,j_2|j_1,\boldsymbol{\psi}}}{m_n \mathcal{L}_{3n}} \to \varphi_{j_2|j_1,\boldsymbol{\psi}},$$

$$\frac{\exp(b^n_{j_1})(\Delta\boldsymbol{a}^n_{j_1})^{\boldsymbol{\gamma}}}{m_n \mathcal{L}_{3n}} \to \lambda_{j_1,\boldsymbol{\gamma}}, \qquad \frac{\exp(b^n_{j_1}) - \exp(b^*_{j_1})}{m_n \mathcal{L}_{3n}} \to \chi_{j_1}$$

with a note that at least one among them is non-zero. Then, the decomposition of $Q_n$ in equation (47) indicates that

$$\lim_{n\to\infty} \frac{Q_n}{m_n \mathcal{L}_{3n}} = \lim_{n\to\infty} \frac{A_n}{m_n \mathcal{L}_{3n}} - \lim_{n\to\infty} \frac{B_n}{m_n \mathcal{L}_{3n}} + \lim_{n\to\infty} \frac{C_n}{m_n \mathcal{L}_{3n}},$$

in which

$$\lim_{n\to\infty} \frac{A_n}{m_n \mathcal{L}_{3n}} = \sum_{j_1=1}^{k^*_1} \sum_{j_2=1}^{k^*_2} \left[ \sum_{|\boldsymbol{\alpha}|=0}^{2} \phi_{j_2|j_1,\boldsymbol{\alpha}_1,\boldsymbol{\alpha}_2,\boldsymbol{\alpha}_3,\rho} \cdot \boldsymbol{x}^{\boldsymbol{\alpha}_3} \frac{\partial^{|\boldsymbol{\alpha}_1|} F}{\partial\boldsymbol{\omega}^{\boldsymbol{\alpha}_1}}(\boldsymbol{x}; \boldsymbol{\omega}^*_{j_2|j_1}) \frac{\partial^{|\boldsymbol{\alpha}_2|} F}{\partial\boldsymbol{a}^{\boldsymbol{\alpha}_2}}(\boldsymbol{x}; \boldsymbol{a}^*_{j_1}) \right.$$

$$\times \frac{\partial^{|\boldsymbol{\alpha}_3|+\rho} \pi}{\partial\xi^{|\boldsymbol{\alpha}_3|+\rho}}(y|(\boldsymbol{\eta}^*_{j_1 j_2})^\top \boldsymbol{x} + \tau^*_{j_1 j_2}, \nu^*_{j_1 j_2})$$

$$\left. - \sum_{|\boldsymbol{\psi}|=0}^{2} \varphi_{j_2|j_1,\boldsymbol{\psi}} \cdot \frac{\partial^{|\boldsymbol{\psi}|} F}{\partial\boldsymbol{\omega}^{\boldsymbol{\psi}}}(\boldsymbol{x}; \boldsymbol{\omega}^*_{j_2|j_1}) \exp(-\|\boldsymbol{a}^*_{j_1} - \boldsymbol{x}\|) p^{LL,*}_{j_1}(y|\boldsymbol{x}) \right] \frac{1}{\sum_{j'_2=1}^{k^*_2} \exp(-\|\boldsymbol{\omega}^*_{j'_2|j_1} - \boldsymbol{x}\| + \beta^*_{j'_2|j_1})},$$

$$\lim_{n\to\infty} \frac{B_n}{m_n \mathcal{L}_{3n}} = \sum_{j_1=1}^{k^*_1} \sum_{|\boldsymbol{\gamma}|=1} \lambda_{j_1,\boldsymbol{\gamma}} \cdot \frac{\partial^{|\boldsymbol{\gamma}|} F}{\partial\boldsymbol{a}^{\boldsymbol{\gamma}}}(\boldsymbol{x}; \boldsymbol{a}^*_{j_1}) p^{LL}_{G_*}(y|\boldsymbol{x}),$$

$$\lim_{n\to\infty} \frac{C_n}{m_n \mathcal{L}_{3n}} = \sum_{j_1=1}^{k^*_1} \chi_{j_1} \exp(-\|\boldsymbol{a}^*_{j_1} - \boldsymbol{x}\|) \left[ p^{LL,*}_{j_1}(y|\boldsymbol{x}) - p^{LL}_{G_*}(y|\boldsymbol{x}) \right].$$

Since the set

$$\left\{\frac{\boldsymbol{x}^{\boldsymbol{\alpha}_3}\frac{\partial^{|\boldsymbol{\alpha}_1|}F}{\partial\boldsymbol{\omega}^{\boldsymbol{\alpha}_1}}(\boldsymbol{x};\boldsymbol{\omega}^*_{j_2|j_1})\frac{\partial^{|\boldsymbol{\alpha}_2|}F}{\partial\boldsymbol{a}^{\boldsymbol{\alpha}_2}}(\boldsymbol{x};\boldsymbol{a}^*_{j_1})\frac{\partial^{|\boldsymbol{\alpha}_3|+\rho}\pi}{\partial\xi^{|\boldsymbol{\alpha}_3|+\rho}}(y|(\boldsymbol{\eta}^*_{j_1j_2})^\top\boldsymbol{x}+\tau^*_{j_1j_2},\nu^*_{j_1j_2})}{\sum_{j_2'=1}^{k_2^*}\exp(-\|\boldsymbol{\omega}^*_{j_2'|j_1}-\boldsymbol{x}\|+\beta^*_{j_2'|j_1})}:j_1\in[k_1^*],\right.$$

$$\left. j_2\in[k_2^*], 0\le|\boldsymbol{\alpha}_1|+|\boldsymbol{\alpha}_2|+|\boldsymbol{\alpha}_3|\le r^{LL}_{j_2|j_1}, 0\le\rho\le 2(r^{LL}_{j_2|j_1}-|\boldsymbol{\alpha}_1|-|\boldsymbol{\alpha}_2|-|\boldsymbol{\alpha}_3|\right\}$$

$$\cup\left\{\frac{\frac{\partial^{|\boldsymbol{\psi}|}F}{\partial\boldsymbol{\omega}^{\boldsymbol{\psi}}}(\boldsymbol{x};\boldsymbol{\omega}^*_{j_2|j_1})\exp((\boldsymbol{a}^*_{j_1})^\top\boldsymbol{x})p^{LL,*}_{j_1}(y|\boldsymbol{x})}{\sum_{j_2'=1}^{k_2^*}\exp(-\|\boldsymbol{\omega}^*_{j_2'|j_1}-\boldsymbol{x}\|+\beta^*_{j_2'|j_1})}:j_1\in[k_1^*],j_2\in[k_2^*], 0\le|\boldsymbol{\psi}|\le 2\right\}$$

$$\cup\left\{\boldsymbol{x}^{\boldsymbol{\gamma}}\exp((\boldsymbol{a}^*_{j_1})^\top\boldsymbol{x})p^{LL}_{G_*}(y|\boldsymbol{x}), \exp((\boldsymbol{a}^*_{j_1})^\top\boldsymbol{x})p^{LL,*}_{j_1}(y|\boldsymbol{x}), \exp((\boldsymbol{a}^*_{j_1})^\top\boldsymbol{x})p^{LL}_{G_*}(y|\boldsymbol{x})\right.$$

$$\left.:j_1\in[k_1^*], 0\le|\boldsymbol{\gamma}|\le 2\right\}$$

is linearly independent, we obtain that $\phi_{j_2|j_1,\boldsymbol{\alpha}_1,\boldsymbol{\alpha}_2,\boldsymbol{\alpha}_3,\rho}=\varphi_{j_2|j_1,\boldsymbol{\psi}}=\lambda_{j_1,\boldsymbol{\gamma}}=\chi_{j_1}=0$ for all $j_1\in[k_1^*]$, $j_2\in[k_2^*]$, $0\le|\boldsymbol{\alpha}_1|+|\boldsymbol{\alpha}_2|+|\boldsymbol{\alpha}_3|\le r^{LL}_{j_2|j_1}$, $0\le\rho\le 2(r^{LL}_{j_2|j_1}-|\boldsymbol{\alpha}_1|-|\boldsymbol{\alpha}_2|-|\boldsymbol{\alpha}_3|)$, $0\le|\boldsymbol{\psi}|\le 2$ and $0\le|\boldsymbol{\gamma}|\le 1$, which is a contradiction. As a consequence, we obtain the inequality in equation (12). Hence, the proof is completed. $\qquad\square$

# F  PROOFS FOR CONVERGENCE OF DENSITY ESTIMATION

*Proof of Theorem 1.* To streamline the arguments for this proof, it is necessary to define some notations that will be used in the sequel. First of all, let $\mathcal{P}^{type}_{k_1^*,k_2}(\Theta)$ stand for the set of conditional density functions w.r.t mixing measures in $\mathcal{G}_{k_1^*,k_2}(\Theta)$ where $type\in\{SS,SL,LL\}$, that is,

$$\mathcal{P}^{type}_{k_1^*,k_2}(\Theta):=\{p^{type}_G(y|\boldsymbol{x}):G\in\mathcal{G}_{k_1^*,k_2}(\Theta)\}.$$

Additionally, we also define

$$\widetilde{\mathcal{P}}^{type}_{k_1^*,k_2}(\Theta):=\{p^{type}_{(G+G_*)/2}(y|\boldsymbol{x}):G\in\mathcal{G}_{k_1^*,k_2}(\Theta)\},$$

$$\widetilde{\mathcal{P}}^{type,1/2}_{k_1^*,k_2}(\Theta):=\{(p^{type}_{(G+G_*)/2})^{1/2}(y|\boldsymbol{x}):G\in\mathcal{G}_{k_1^*,k_2}(\Theta)\}.$$

Next, for each $\delta>0$, we define the $L^2$-ball centered around the regression function $p^{type}_{G_*}$ and intersected with the set $\widetilde{\mathcal{P}}^{type,1/2}_{k_1^*,k_2}(\Theta)$ as

$$\widetilde{\mathcal{P}}^{type,1/2}_{k_1^*,k_2}(\Theta,\delta):=\left\{p^{1/2}\in\widetilde{\mathcal{P}}^{type,1/2}_{k_1^*,k_2}(\Theta):h(p,p^{type}_{G_*})\le\delta\right\}.$$

Following the suggestion from Geer et. al. van de Geer (2000), we utilize the following integral to capture the size of the above $L^2$-ball:

$$\mathcal{J}_B(\delta,\widetilde{\mathcal{P}}^{type,1/2}_{k_1^*,k_2}(\Theta,\delta)):=\int_{\delta^2/2^{13}}^{\delta}H_B^{1/2}(t,\widetilde{\mathcal{P}}^{type,1/2}_{k_1^*,k_2}(\Theta,t),\|\cdot\|_{L^2})\,\mathrm{d}t\vee\delta, \tag{60}$$

where the term $H_B(t,\widetilde{\mathcal{P}}^{type,1/2}_{k_1^*,k_2}(\Theta,t),\|\cdot\|_{L^2})$ denotes the bracketing entropy van de Geer (2000) of $\widetilde{\mathcal{P}}^{type,1/2}_{k_1^*,k_2}(\Theta,t)$ under the $L^2$-norm, and $t\vee\delta:=\max\{t,\delta\}$.

Let us recall the statement of Theorem 7.4 in van de Geer (2000) with adapted notations to our paper as follows:

**Lemma 2** (Theorem 7.4, van de Geer (2000)). *Let $\Psi(\delta)\ge\mathcal{J}_B(\delta,\widetilde{\mathcal{P}}^{type,1/2}_{k_1^*,k_2}(\Theta,\delta))$ be such that $\Psi(\delta)/\delta^2$ is a non-increasing function of $\delta$. Then, for some universal constant $c$ and for some sequence $(\delta_n)$ such that $\sqrt{n}\delta_n^2\ge c\Psi(\delta_n)$, the following inequality holds for all $\delta\ge\delta_n$:*

$$\mathbb{P}\left(\mathbb{E}_{\boldsymbol{X}}[h(p^{type}_{\widehat{G}_n^{type}}(\cdot|\boldsymbol{X}),p^{type}_{G_*}(\cdot|\boldsymbol{X}))]>\delta\right)\le c\exp\left(-\frac{n\delta^2}{c^2}\right).$$

**Proof overview.** Given that the expert functions are Lipschitz continuous, we begin with showing that the following bound holds for any $0 < \varepsilon \leq 1/2$:

$$H_B(\varepsilon, \mathcal{P}_{k_1^*, k_2}^{type}(\Theta), h) \lesssim \log(1/\varepsilon), \tag{61}$$

which yields that

$$\mathcal{J}_B(\delta, \widetilde{\mathcal{P}}_{k_1^*, k_2}^{type, 1/2}(\Theta, \delta)) = \int_{\delta^2/2^{13}}^{\delta} H_B^{1/2}(t, \widetilde{\mathcal{P}}_{k_1^*, k_2}^{type, 1/2}(\Theta, t), \|\cdot\|_{L^2}) \, \mathrm{d}t \vee \delta$$

$$\leq \int_{\delta^2/2^{13}}^{\delta} H_B^{1/2}(t, \mathcal{P}_{k_1^*, k_2}^{type}(\Theta, t), h) \, \mathrm{d}t \vee \delta$$

$$\lesssim \int_{\delta^2/2^{13}}^{\delta} \log(1/t) dt \vee \delta. \tag{62}$$

Let $\Psi(\delta) = \delta \cdot [\log(1/\delta)]^{1/2}$, then it can be checked that $\Psi(\delta)/\delta^2$ is a non-increasing function of $\delta$. Moreover, the result in equation (62) implies that $\Psi(\delta) \geq \mathcal{J}_B(\delta, \widetilde{\mathcal{P}}_{k_1^*, k_2}^{type, 1/2}(\Theta, \delta))$. By choosing $\delta_n = \sqrt{\log(n)/n}$, we have that $\sqrt{n}\delta_n^2 \geq c\Psi(\delta_n)$ for some universal constant $c$. Then, the conclusion of this theorem is achieved according to Lemma 2. Consequently, it is sufficient to derive the bracketing entropy bound in equation (61).

**Proof for the bound** (61). To begin with, we provide an upper bound for the Gaussian density function $\pi(y|\eta^\top \boldsymbol{x} + \tau, \nu)$. In particular, since the input space $\mathcal{X}$ and the parameter space $\Theta$ are both bounded, we can find some constant $\kappa, \ell, u > 0$ such that $-\kappa \leq \eta^\top \boldsymbol{x} + \tau \leq \kappa$ and $\ell \leq \nu \leq u$. Then, it can be validated that

$$\pi(y|\eta^\top \boldsymbol{x} + \tau, \nu) = \frac{1}{\sqrt{2\pi\nu}} \exp\left(-\frac{(y - (\eta^\top \boldsymbol{x} + \tau))^2}{2\nu}\right) \leq \frac{1}{\sqrt{2\pi\ell}},$$

for any $|y| < 2\kappa$. On the other hand, for $|y| \geq 2\kappa$, since $\frac{(y-(\eta^\top \boldsymbol{x}+\tau))^2}{2\nu} \geq \frac{y^2}{8u}$, we have that

$$\pi(y|\eta^\top \boldsymbol{x} + \tau, \nu) \leq \frac{1}{\sqrt{2\pi\ell}} \exp\left(-\frac{y^2}{8u}\right).$$

Therefore, we deduce that $\pi(y|\eta^\top \boldsymbol{x} + \tau, \nu) \leq M(y|\boldsymbol{x})$, where

$$M(y|\boldsymbol{x}) = \begin{cases} \frac{1}{\sqrt{2\pi\ell}} \exp\left(-\frac{y^2}{8u}\right), & \text{for } |y| \geq 2\kappa, \\ \frac{1}{\sqrt{2\pi\ell}}, & \text{for } |y| < 2\kappa. \end{cases}$$

Next, let $0 < \tau \leq \varepsilon$ and $\{\pi_1, \ldots, \pi_N\}$ be the $\tau$-cover under the $L^\infty$-norm of the set $\mathcal{P}_{k_1^*, k_2}^{type}(\Theta)$ where $N := N(\tau, \mathcal{P}_{k_1^*, k_2}^{type}(\Theta), \|\cdot\|_{L^\infty})$ stands for the $\tau$-covering number of the norm space $(\mathcal{P}_{k_1^*, k_2}^{type}(\Theta), \|\cdot\|_{L^\infty})$. Equipped with the brackets of the form $[L_i, U_i]$ where

$$L_i(y|\boldsymbol{x}) := \max\{\pi_i(y|\boldsymbol{x}) - \tau, 0\},$$
$$U_i(y|\boldsymbol{x}) := \max\{\pi_i(y|\boldsymbol{x}) + \tau, M(y|\boldsymbol{x})\},$$

for all $i \in [N]$, we can validate that $\mathcal{P}_{k_1^*, k_2}^{type}(\Theta) \subset \cup_{i=1}^N [L_i, U_i]$, and $U_i(y|\boldsymbol{x}) - L_i(y|\boldsymbol{x}) \leq \min\{2\tau, M\}$. Those results yield that

$$\|U_i - L_i\|_{L^1} = \int (U_i(y|\boldsymbol{x}) - L_i(y|\boldsymbol{x})) \mathrm{d}(\boldsymbol{x}, y) \leq \int 2\tau \mathrm{d}(\boldsymbol{x}, y) = 2\tau,$$

From the definition of the bracketing entropy, we have

$$H_B(2\tau, \mathcal{P}_{k_1^*, k_2}^{type}(\Theta), \|\cdot\|_{L^1}) \leq \log N = \log N(\tau, \mathcal{P}_{k_1^*, k_2}^{type}(\Theta), \|\cdot\|_{L^\infty}). \tag{63}$$

Therefore, it suffices to provide an upper bound for the covering number $N$. Indeed, let us denote $\Delta := \{(b, \boldsymbol{a}) \in \mathbb{R} \times \mathbb{R}^d : (b, \boldsymbol{a}, \beta, \boldsymbol{\omega}, \tau, \boldsymbol{\eta}, \nu) \in \Theta\}$ and $\Omega := \{(\beta, \boldsymbol{\omega}, \tau, \boldsymbol{\eta}, \nu) \in \mathbb{R} \times \mathbb{R}^d \times \mathbb{R} \times \mathbb{R}^d \times \mathbb{R}_+ : (b, \boldsymbol{a}, \beta, \boldsymbol{\omega}, \tau, \boldsymbol{\eta}, \nu) \in \Theta\}$. As $\Theta$ is a compact set, so are $\Delta$ and $\Omega$. Thus, we can find $\tau$-covers $\Delta_\tau$ and $\Omega_\tau$ for $\Delta$ and $\Omega$, respectively. Furthermore, it can be validated that

$$|\Delta_\tau| \leq \mathcal{O}_P(\tau^{-(d+1)k_1^*}), \quad |\Omega_\tau| \leq \mathcal{O}_P(\tau^{-(2d+3)k_1^* k_2}).$$

For each mixing measure $G = \sum_{i_1=1}^{k_1^*} \exp(b_{i_1}) \sum_{i_2=1}^{k_2} \exp(\beta_{i_2|i_1}) \delta_{(\boldsymbol{a}_{i_1}, \boldsymbol{\omega}_{i_2|i_1}, \boldsymbol{\eta}_{i_1 i_2}, \tau_{i_1 i_2}, \nu_{i_1 i_2})} \in \mathcal{G}_{k_1^*, k_2}(\Theta)$, we consider two other mixing measures $G'$ and $\overline{G}$ defined as

$$G' := \sum_{i_1=1}^{k_1^*} \exp(b_{i_1}) \sum_{i_2=1}^{k_2} \exp(\overline{\beta}_{i_2|i_1}) \delta_{(\boldsymbol{a}_{i_1}, \overline{\boldsymbol{\omega}}_{i_2|i_1}, \overline{\boldsymbol{\eta}}_{i_1 i_2}, \overline{\tau}_{i_1 i_2}, \overline{\nu}_{i_1 i_2})},$$

$$\overline{G} := \sum_{i_1=1}^{k_1^*} \exp(\overline{b}_{i_1}) \sum_{i_2=1}^{k_2} \exp(\overline{\beta}_{i_2|i_1}) \delta_{(\overline{\boldsymbol{a}}_{i_1}, \overline{\boldsymbol{\omega}}_{i_2|i_1}, \overline{\boldsymbol{\eta}}_{i_1 i_2}, \overline{\tau}_{i_1 i_2}, \overline{\nu}_{i_1 i_2})}.$$

Above, $(\overline{\beta}_{i_2|i_1}, \overline{\boldsymbol{\omega}}_{i_2|i_1}, \overline{\boldsymbol{\eta}}_{i_1 i_2}, \overline{\tau}_{i_1 i_2}, \overline{\nu}_{i_1 i_2}) \in \Omega_\tau$ such that $(\overline{\beta}_{i_2|i_1}, \overline{\boldsymbol{\omega}}_{i_2|i_1}, \overline{\boldsymbol{\eta}}_{i_1 i_2}, \overline{\tau}_{i_1 i_2}, \overline{\nu}_{i_1 i_2})$ is the closest to $(\beta_{i_2|i_1}, \boldsymbol{\omega}_{i_2|i_1}, \boldsymbol{\eta}_{i_1 i_2}, \tau_{i_1 i_2}, \nu_{i_1 i_2})$ in that set, while $(\overline{b}_{i_1}, \overline{\boldsymbol{a}}_{i_1}) \in \Delta_\tau$ is the closest to $(b_{i_1}, \boldsymbol{\omega}_i)$ in that set.

Now, we begin bounding the term $\|p_G^{type} - p_{G'}^{type}\|_{L^\infty}$. For brevity, we will consider only the case when $type = SS$, while the other two cases when $type = SL$ and $type = LL$ can be argued in a similar fashion.

**When $type = SS$:** Let us define

$$p_{i_1}^{SS}(\boldsymbol{x}) := \sum_{i_2=1}^{k_2} \sigma((\boldsymbol{\omega}_{i_2|i_1})^\top \boldsymbol{x} + \beta_{i_2|i_1}) \pi(y|(\boldsymbol{\eta}_{i_1 i_2})^\top \boldsymbol{x} + \tau_{i_1 i_2}, \nu_{i_1 i_2}),$$

$$\overline{p}_{i_1}^{SS}(\boldsymbol{x}) := \sum_{i_2=1}^{k_2} \sigma((\overline{\boldsymbol{\omega}}_{i_2|i_1})^\top \boldsymbol{x} + \overline{\beta}_{i_2|i_1}) \pi(y|(\overline{\boldsymbol{\eta}}_{i_1 i_2})^\top \boldsymbol{x} + \overline{\tau}_{i_1 i_2}, \overline{\nu}_{i_1 i_2}).$$

Then, we have

$$\|p_G^{SS} - p_{G'}^{SS}\|_{L^\infty} = \sum_{i_1=1}^{k_1^*} \sigma\left((\boldsymbol{a}_{i_1})^\top \boldsymbol{x} + b_{i_1}\right) \cdot \|p_{i_1}^{SS} - \overline{p}_{i_1}^{SS}\|_{L^\infty} \leq \sum_{i_1=1}^{k_1^*} \|p_{i_1}^{SS} - \overline{p}_{i_1}^{SS}\|_{L^\infty}. \quad (64)$$

Next, we need to bound the terms $p_{i_1}^{SS}(\boldsymbol{x}) - \overline{p}_{i_1}^{SS}(\boldsymbol{x})$ using the triangle inequality

$$\|p_{i_1}^{SS} - \overline{p}_{i_1}^{SS}\|_{L^\infty} \leq \|p_{i_1}^{SS} - \widetilde{p}_{i_1}^{SS}\|_{L^\infty} + \|\widetilde{p}_{i_1}^{SS} - \overline{p}_{i_1}^{SS}\|_{L^\infty}, \quad (65)$$

where we define

$$\widetilde{p}_{i_1}^{SS}(\boldsymbol{x}) := \sum_{i_2=1}^{k_2} \sigma((\boldsymbol{\omega}_{i_2|i_1})^\top \boldsymbol{x} + \beta_{i_2|i_1}) \pi(y|(\overline{\boldsymbol{\eta}}_{i_1 i_2})^\top \boldsymbol{x} + \overline{\tau}_{i_1 i_2}, \overline{\nu}_{i_1 i_2}).$$

Firstly, we have

$$\|p_{i_1}^{SS} - \widetilde{p}_{i_1}^{SS}\|_{L^\infty} \leq \sum_{i_2=1}^{k_2} \sigma((\boldsymbol{\omega}_{i_2|i_1})^\top \boldsymbol{x} + \beta_{i_2|i_1})$$

$$\times \|\pi(y|(\boldsymbol{\eta}_{i_1 i_2})^\top \boldsymbol{x} + \tau_{i_1 i_2}, \nu_{i_1 i_2}) - \pi(y|(\overline{\boldsymbol{\eta}}_{i_1 i_2})^\top \boldsymbol{x} + \overline{\tau}_{i_1 i_2}, \overline{\nu}_{i_1 i_2})\|_{L^\infty}$$

$$\leq \sum_{i_2=1}^{k_2} \|\pi(y|(\boldsymbol{\eta}_{i_1 i_2})^\top \boldsymbol{x} + \tau_{i_1 i_2}, \nu_{i_1 i_2}) - \pi(y|(\overline{\boldsymbol{\eta}}_{i_1 i_2})^\top \boldsymbol{x} + \overline{\tau}_{i_1 i_2}, \overline{\nu}_{i_1 i_2})\|_{L^\infty}$$

$$\lesssim \sum_{i_2=1}^{k_2} \left( \|\boldsymbol{\eta}_{i_1 i_2} - \overline{\boldsymbol{\eta}}_{i_1 i_2}\| + |\tau_{i_1 i_2} - \overline{\tau}_{i_1 i_2}| + |\nu_{i_1 i_2} - \overline{\nu}_{i_1 i_2}| \right) \lesssim \tau. \quad (66)$$

Secondly, since $\mathcal{X}$ is a bounded set, we may assume that $\|\boldsymbol{x}\| \leq B$ for any $\boldsymbol{x} \in \mathcal{X}$. Then, it follows that

$$
\begin{aligned}
\|\widetilde{p}_{i_1}^{SS} - \overline{p}_{i_1}^{SS}\|_{L^\infty} &\leq \sum_{i_2=1}^{k_2} \left| \sigma((\boldsymbol{\omega}_{i_2|i_1})^\top \boldsymbol{x} + \beta_{i_2|i_1}) - \sigma((\overline{\boldsymbol{\omega}}_{i_2|i_1})^\top \boldsymbol{x} + \overline{\beta}_{i_2|i_1}) \right| \\
&\qquad\qquad\qquad\qquad \times \|\pi(y|(\overline{\boldsymbol{\eta}}_{i_1 i_2})^\top \boldsymbol{x} + \overline{\tau}_{i_1 i_2}, \overline{\nu}_{i_1 i_2})\|_{L^\infty} \\
&\lesssim \sum_{i_2=1}^{k_2} \left[ \|\boldsymbol{\omega}_{i_2|i_1} - \overline{\boldsymbol{\omega}}_{i_2|i_1}\| \cdot |\boldsymbol{x}| + |\beta_{i_2|i_1} - \overline{\beta}_{i_2|i_1}| \right] \\
&\leq \sum_{i_2=1}^{k_2} \left( \tau B + \tau \right) \lesssim \tau.
\end{aligned}
\tag{67}
$$

From the results in equations (64), (65), (66) and (67), we deduce that

$$
\|p_G^{SS} - p_{G'}^{SS}\|_{L^\infty} \lesssim \tau.
\tag{68}
$$

Furthermore, we have

$$
\begin{aligned}
\|p_{G'}^{SS} - p_{\overline{G}}^{SS}\|_{L^\infty} &= \sum_{i_1=1}^{k_1^*} |\sigma((\boldsymbol{a}_{i_1})^\top \boldsymbol{x} + b_{i_1}) - \sigma((\overline{\boldsymbol{a}}_{i_1})^\top \boldsymbol{x} + \overline{b}_{i_1})| \cdot \|\pi(y|(\overline{\boldsymbol{\eta}}_{i_1 i_2})^\top \boldsymbol{x} + \overline{\tau}_{i_1 i_2}, \overline{\nu}_{i_1 i_2})\|_{L^\infty} \\
&\lesssim \sum_{i_1=1}^{k_1^*} \left( \|\boldsymbol{a}_{i_1} - \overline{\boldsymbol{a}}_{i_1}\| \cdot \|\boldsymbol{x}\| + |b_{i_1} - \overline{b}_{i_1}| \right) \\
&\leq \sum_{i_1=1}^{k_1^*} (\tau B + \tau) \lesssim \tau.
\end{aligned}
\tag{69}
$$

According to the triangle inequality and the results in equations (68), (69), we have

$$
\|p_G^{SS} - p_{\overline{G}}^{SS}\|_{L^\infty} \leq \|p_G^{SS} - p_{G'}^{SS}\|_{L^\infty} + \|p_{G'}^{SS} - p_{\overline{G}}^{SS}\|_{L^\infty} \lesssim \tau.
$$

By definition of the covering number, we deduce that

$$
\begin{aligned}
N(\tau, \mathcal{P}_{k_1^*, k_2}^{type}(\Theta), \|\cdot\|_{L^2(\mu)}) &\leq |\Delta_\tau| \times |\Omega_\tau| \\
&\leq \mathcal{O}_P(\tau^{-(d+1)k_1^*}) \times \mathcal{O}_P(\tau^{-(2d+3)k_1^* k_2}) \\
&\leq \mathcal{O}_P(\tau^{-(d+1)k_1^* - (2d+3)k_1^* k_2}).
\end{aligned}
\tag{70}
$$

Combine the result in equation (63) with that in (70), we arrive at

$$
H_B(2\tau, \mathcal{P}_{k_1^*, k_2}^{type}(\Theta), \|\cdot\|_{L^1}) \lesssim \log(1/\tau).
$$

Let $\tau = \varepsilon/2$, then it follows that

$$
H_B(\varepsilon, \mathcal{P}_{k_1^*, k_2}^{type}(\Theta), \|.\|_{L^1}) \lesssim \log(1/\varepsilon).
$$

Finally, due to the inequality between the Hellinger distance and the $L^1$-norm $h \leq \|\cdot\|_{L^1}$, we achieve the conclusion that

$$
H_B(\varepsilon, \mathcal{P}_{k_1^*, k_2}^{type}(\Theta), h) \lesssim \log(1/\varepsilon).
$$

Hence, the proof is completed. $\qquad\square$

## G  PROOF OF LEMMA 1

Firstly, let us recall the system of polynomial equations given in equation (4):

$$
\sum_{i_2=1}^{m} \sum_{\alpha \in \mathcal{I}_{\boldsymbol{\rho}_1, \rho_2}^{SS}} \frac{p_{i_2}^2 \, \boldsymbol{q}_{1 i_2}^{\boldsymbol{\alpha}_1} \, \boldsymbol{q}_{2 i_2}^{\boldsymbol{\alpha}_2} \, \boldsymbol{q}_{3 i_2}^{\boldsymbol{\alpha}_3} \, q_{4 i_2}^{\alpha_4} \, q_{5 i_2}^{\alpha_5}}{\boldsymbol{\alpha}_1! \, \boldsymbol{\alpha}_2! \, \boldsymbol{\alpha}_3! \, \alpha_4! \alpha_5!} = 0, \quad 1 \leq |\boldsymbol{\rho}_1| + \rho_2 \leq r,
\tag{71}
$$

where $\mathcal{I}_{\boldsymbol{\rho}_1, \rho_2}^{SS} = \{\alpha = (\boldsymbol{\alpha}_1, \boldsymbol{\alpha}_2, \boldsymbol{\alpha}_3, \alpha_4, \alpha_5) \in \mathbb{N}^d \times \mathbb{N}^d \times \mathbb{N}^d \times \mathbb{N} \times \mathbb{N} : \boldsymbol{\alpha}_1 + \boldsymbol{\alpha}_2 + \boldsymbol{\alpha}_3 = \boldsymbol{\rho}_1, \ \alpha_4 + 2\alpha_5 = \rho_2 - |\boldsymbol{\alpha}_3|\}$.

**When $m = 2$:** By observing a portion of the above system when $\boldsymbol{\rho}_1 = \mathbf{0}_d$, which is given by

$$\sum_{i_2=1}^{m} \sum_{\alpha_4 + 2\alpha_5 = \rho_2} \frac{p_{i_2}^2 \, q_{4i_2}^{\alpha_4} \, q_{5i_2}^{\alpha_5}}{\alpha_4! \, \alpha_5!} = 0, \quad \rho_2 = 1, 2, \dots, r. \tag{72}$$

Proposition 2.1 in Ho & Nguyen (2016) shows that the smallest $r \in \mathbb{N}$ such that the system (72) does not admit any non-trivial solutions when $m = 2$ is $r = 4$. Note that a solution of the system 72 is called non-trivial in Ho & Nguyen (2016) if all the values of $p_{i_2}$ are different from zero, whereas at least one among $q_{4i_2}$ is non-zero. This definition of non-trivial solutions totally aligns with ours for the system (71). Therefore, we have $\bar{r}(m) \le 4$, and it suffices to prove that $\bar{r}(m) > 3$.

Indeed, when $r = 3$, we demonstrate that the system (71) admits a non-trivial solution: $p_{i_2} = 1$, $\boldsymbol{q}_{1i_2} = \boldsymbol{q}_{2i_2} = \boldsymbol{q}_{3i_2} = \mathbf{0}_d$ for all $i_2 \in [m]$, $q_{41} = 1$, $q_{42} = -1$, $q_{51} = q_{52} = -\frac{1}{2}$. Since $\boldsymbol{q}_{1i_2} = \boldsymbol{q}_{2i_2} = \boldsymbol{q}_{3i_2} = \mathbf{0}_d$, this solution clearly satisfies the equations associated with $\boldsymbol{\rho}_1 \ne \mathbf{0}_d$. Thus, we only need to verify those with $\boldsymbol{\rho}_1 = \mathbf{0}_d$, which are given by

$$\sum_{j=1}^{m} p_{i_2}^2 q_{4i_2} = 0,$$

$$\sum_{i_2=1}^{m} p_{i_2}^2 \left( \frac{1}{2} q_{4i_2}^2 + q_{5i_2} \right) = 0,$$

$$\sum_{i_2=1}^{m} p_{i_2}^2 \left( \frac{1}{3!} q_{4i_2}^3 + q_{4i_2} q_{5i_2} \right) = 0.$$

By simple calculations, we can check that $p_{i_2} = 1$, $q_{41} = 1$, $q_{42} = -1$, $q_{51} = q_{52} = -\frac{1}{2}$ satisfies the above equations. Hence, we obtain that $\bar{r}(m) > 3$, leading to $\bar{r}(m) = 4$.

**When $m = 3$:** Note that $\bar{r}(m)$ is a monotonically increasing function of $m$. Therefore, it follows from the previous result that $\bar{r}(m) > \bar{r}(2) = 4$, or equivalently, $\bar{r}(m) \ge 5$. Additionally, according to Proposition 2.1 in Ho & Nguyen (2016), we deduce that $\bar{r}(m) \le 6$ based on the reduced system in equation (72). Thus, we only need to show that $\bar{r}(m) > 5$.

Indeed, we show that the following is a non-trivial solution of the system (71) when $r = 5$:
$$p_{i_2} = 1, \quad \boldsymbol{q}_{1i_2} = \boldsymbol{q}_{2i_2} = \boldsymbol{q}_{3i_2} = \mathbf{0}_d, \quad \forall i_2 \in [m],$$
$$q_{41} = \frac{\sqrt{3}}{3}, \quad q_{42} = -\frac{\sqrt{3}}{3}, \quad q_{43} = 0,$$
$$q_{51} = q_{52} = -\frac{1}{6}, \quad q_{53} = 0.$$

Since $\boldsymbol{q}_{1i_2} = \boldsymbol{q}_{2i_2} = \boldsymbol{q}_{3i_2} = \mathbf{0}_d$, this solution clearly satisfies the equations associated with $\boldsymbol{\rho}_1 \ne \mathbf{0}_d$. Thus, we only need to verify those with $\boldsymbol{\rho}_1 = \mathbf{0}_d$, which are given by

$$\sum_{j=1}^{m} p_{i_2}^2 q_{4i_2} = 0,$$

$$\sum_{i_2=1}^{m} p_{i_2}^2 \left( \frac{1}{2} q_{4i_2}^2 + q_{5i_2} \right) = 0,$$

$$\sum_{i_2=1}^{m} p_{i_2}^2 \left( \frac{1}{3!} q_{4i_2}^3 + q_{4i_2} q_{5i_2} \right) = 0,$$

$$\sum_{i_2=1}^{m} p_{i_2}^2 \left( \frac{1}{4!} q_{4i_2}^4 + \frac{1}{2!} q_{4i_2}^2 q_{5i_2} + \frac{1}{2!} q_{5i_2}^2 \right) = 0,$$

$$\sum_{i_2=1}^{m} p_{i_2}^2 \left( \frac{1}{5!} q_{4i_2}^5 + \frac{1}{3!} q_{4i_2}^3 q_{5i_2} + \frac{1}{2!} q_{4i_2} q_{5i_2}^2 \right) = 0.$$

By simple calculations, it can be validated that $p_{i_2} = 1$, $q_{41} = \frac{\sqrt{3}}{3}$, $q_{42} = -\frac{\sqrt{3}}{3}$, $q_{43} = 0$, $q_{51} = q_{52} = -\frac{1}{6}$, $q_{53} = 0$ satisfies the above equations. Hence, we conclude $\bar{r}(m) > 5$, meaning that $\bar{r}(m) = 6$.

## H    IDENTIFIABILITY OF THE GAUSSIAN HMOE

**Proposition 1.** *For each* $type \in \{SS, SL, LL\}$, *suppose that the equation* $p_G^{type}(y|\boldsymbol{x}) = p_{G_*}^{type}(y|\boldsymbol{x})$ *holds true for almost surely* $(\boldsymbol{x}, y)$, *then we get that* $G \equiv G_*$.

*Proof of Proposition 1.*  In this proof, we will consider only the case when $type = SS$ as other cases can be done similarly.

To start with, let us write the equation $p_G^{SS}(y|\boldsymbol{x}) = p_{G_*}^{SS}(y|\boldsymbol{x})$ explicitly as follows:

$$\sum_{i_1=1}^{k_1^*} \sigma\Big((\boldsymbol{a}_{i_1})^\top \boldsymbol{x} + b_{i_1}\Big) \sum_{i_2=1}^{k_2} \sigma\Big((\boldsymbol{\omega}_{i_2|i_1})^\top \boldsymbol{x} + \beta_{i_2|i_1}\Big) \pi(y|(\boldsymbol{\eta}_{i_1 i_2})^\top \boldsymbol{x} + \tau_{i_1 i_2}, \nu_{i_1 i_2})$$

$$= \sum_{i_1=1}^{k_1^*} \sigma\Big((\boldsymbol{a}_{i_1}^*)^\top \boldsymbol{x} + b_{i_1}^*\Big) \sum_{i_2=1}^{k_2^*} \sigma\Big((\boldsymbol{\omega}_{i_2|i_1}^*)^\top \boldsymbol{x} + \beta_{i_2|i_1}^*\Big) \pi(y|(\boldsymbol{\eta}_{i_1 i_2}^*)^\top \boldsymbol{x} + \tau_{i_1 i_2}^*, \nu_{i_1 i_2}^*). \quad (73)$$

Then, it follows from the identifiability of the location-scale Gaussian mixtures (Teicher, 1960; 1961) that the number of components and the weight set of the mixing measure $G$ equal to those of its counterpart $G_*$, i.e. $k_2 = k_2^*$ and

$$\left\{ \sigma\Big((\boldsymbol{a}_{i_1})^\top \boldsymbol{x} + b_{i_1}\Big) \cdot \sigma\Big((\boldsymbol{\omega}_{i_2|i_1})^\top \boldsymbol{x} + \beta_{i_2|i_1}\Big) : i_1 \in [k_1^*], i_2 \in [k_2^*] \right\}$$

$$= \left\{ \sigma\Big((\boldsymbol{a}_{i_1}^*)^\top \boldsymbol{x} + b_{i_1}^*\Big) \cdot \sigma\Big((\boldsymbol{\omega}_{i_2|i_1}^*)^\top \boldsymbol{x} + \beta_{i_2|i_1}^*\Big) : i_1 \in [k_1^*], i_2 \in [k_2^*] \right\},$$

for almost every $\boldsymbol{x}$. WLOG, we may assume that

$$\sigma\Big((\boldsymbol{a}_{i_1})^\top \boldsymbol{x} + b_{i_1}\Big) \cdot \sigma\Big((\boldsymbol{\omega}_{i_2|i_1})^\top \boldsymbol{x} + \beta_{i_2|i_1}\Big) = \sigma\Big((\boldsymbol{a}_{i_1}^*)^\top \boldsymbol{x} + b_{i_1}^*\Big) \cdot \sigma\Big((\boldsymbol{\omega}_{i_2|i_1}^*)^\top \boldsymbol{x} + \beta_{i_2|i_1}^*\Big),$$
$$(74)$$

for almost every $\boldsymbol{x}$, for any $i_1 \in [k_1^*], i_2 \in [k_2^*]$. Due to the assumptions that $\boldsymbol{\omega}_{k_2^*|i_1} = \boldsymbol{\omega}_{k_2^*|i_1}^* = \boldsymbol{0}_d$ and $\beta_{k_2^*|i_1} = \beta_{k_2^*|i_1}^* = 0$, we have that

$$\sigma\Big((\boldsymbol{a}_{i_1})^\top \boldsymbol{x} + b_{i_1}\Big) = \sigma\Big((\boldsymbol{a}_{i_1}^*)^\top \boldsymbol{x} + b_{i_1}^*\Big), \quad (75)$$

for almost every $\boldsymbol{x}$, for any $i_1 \in$. Since the $\sigma$ function is invariant to translations, then it follows from the equation (75) that

$$\boldsymbol{a}_{i_1} = \boldsymbol{a}_{i_1}^* + \boldsymbol{a}$$
$$b_{i_1} = b_{i_1}^* + b,$$

for some $\boldsymbol{a} \in \mathbb{R}^d$ and $b \in \mathbb{R}$. Moreover, due to the assumption that $\boldsymbol{a}_{k_1^*} = \boldsymbol{a}_{k_1^*}^*$ and $b_{k_1^*} = b_{k_1^*}^* = 0$, we get $\boldsymbol{a} = \boldsymbol{0}_d$ and $b = 0$. This leads to $\boldsymbol{a}_{i_1} = \boldsymbol{a}_{i_1}^*$ and $b_{i_1} = b_{i_1}^*$ for any $i_1 \in [k_1^*]$. Those results together with equation (74) yield that

$$\sigma\Big((\boldsymbol{\omega}_{i_2|i_1})^\top \boldsymbol{x} + \beta_{i_2|i_1}\Big) = \sigma\Big((\boldsymbol{\omega}_{i_2|i_1}^*)^\top \boldsymbol{x} + \beta_{i_2|i_1}^*\Big),$$

for almost every $\boldsymbol{x}$, for any $i_1 \in [k_1^*], i_2 \in [k_2^*]$. By employing the previous arguments, we also obtain that

$$\boldsymbol{\omega}_{i_2|i_1} = \boldsymbol{\omega}_{i_2|i_1}^*,$$
$$\beta_{i_2|i_1} = \beta_{i_2|i_1}^*.$$

Then, the equation (73) can be rewritten as

$$
\sum_{i_1=1}^{k_1^*} \exp(b_{i_1}) \sum_{i_2=1}^{k_2^*} \exp(\beta_{i_2|i_1}) \exp\left((\boldsymbol{a}_{i_1} + \boldsymbol{\omega}_{i_2|i_1})^\top \boldsymbol{x}\right) \pi(y|(\boldsymbol{\eta}_{i_1 i_2})^\top \boldsymbol{x} + \tau_{i_1 i_2}, \nu_{i_1 i_2})
$$

$$
= \sum_{i_1=1}^{k_1^*} \exp(b_{i_1}^*) \sum_{i_2=1}^{k_2^*} \exp(c_{i_2|i_1}^*) \exp\left((\boldsymbol{a}_{i_1}^* + \boldsymbol{\omega}_{i_2|i_1}^*)^\top \boldsymbol{x}\right) \pi(y|(\boldsymbol{\eta}_{i_1 i_2}^*)^\top \boldsymbol{x} + \tau_{i_1 i_2}^*, \nu_{i_1 i_2}^*). \quad (76)
$$

for almost every $\boldsymbol{x} \in \mathcal{X}$.

Next, we denote $P_1, P_2, \ldots, P_{m_1}$ as a partition of the index set $[k_1^*]$, where $m_1 \le k_1^*$, such that $\exp(b_{i_1}) = \exp(b_{i_1'}^*)$ for any $i_1, i_1' \in P_j$ and $j_1 \in [m_1]$. On the other hand, when $i_1$ and $i_1'$ do not belong to the same set $P_{j_1}$, we let $\exp(b_{i_1}) \ne \exp(b_{i_1'}^*)$.

Similarly, for each $i_1 \in [k_1^*]$, we also define $Q_{1|i_1}, Q_{2|i_1}, \ldots, Q_{m_2|i_1}$ as a partition of the index set $[k_2^*]$, where $m_2 \le k_2^*$, such that $\exp(\beta_{i_2|i_1}) = \exp(\beta_{i_2'|i_1}^*)$ for any $i_2, i_2' \in Q_{j_2|i_1}$ and $j_2 \in [m_2]$. Conversely, when $i_2$ and $i_2'$ do not belong to the same set $Q_{j_2|i_1}$, we let $\exp(\beta_{i_2|i_1}) \ne \exp(\beta_{i_2'|i_1}^*)$.

Thus, we can represent equation (76) as

$$
\sum_{j_1=1}^{m_1} \sum_{i_1 \in P_{j_1}} \exp(b_{i_1}) \sum_{j_2=1}^{m_2} \sum_{i_1 \in Q_{j_2|i_1}} \exp(\beta_{i_2|i_1}) \exp\left((\boldsymbol{a}_{i_1} + \boldsymbol{\omega}_{i_2|i_1})^\top \boldsymbol{x}\right) \pi(y|(\boldsymbol{\eta}_{i_1 i_2})^\top \boldsymbol{x} + \tau_{i_1 i_2}, \nu_{i_1 i_2})
$$

$$
= \sum_{j_1=1}^{m_1} \sum_{i_1 \in P_{j_1}} \exp(b_{i_1}^*) \sum_{j_2=1}^{m_2} \sum_{i_1 \in Q_{j_2|i_1}} \exp(\beta_{i_2|i_1}^*) \exp\left((\boldsymbol{a}_{i_1}^* + \boldsymbol{\omega}_{i_2|i_1}^*)^\top \boldsymbol{x}\right) \pi(y|(\boldsymbol{\eta}_{i_1 i_2}^*)^\top \boldsymbol{x} + \tau_{i_1 i_2}^*, \nu_{i_1 i_2}^*),
$$

for almost every $\boldsymbol{x} \in \mathcal{X}$. Recall that we have $b_{i_1} = b_{i_1}^*$, $\boldsymbol{a}_{i_1} = \boldsymbol{a}_{i_1}^*$, $\boldsymbol{\omega}_{i_2|i_1} = \boldsymbol{\omega}_{i_2|i_1}^*$ and $\beta_{i_2|i_1} = \beta_{i_2|i_1}^*$, for any $i_1 \in [k_1^*]$ and $i_2 \in [k_2^*]$, then the above result leads to

$$
\left\{ \left((\boldsymbol{\eta}_{i_1 i_2})^\top \boldsymbol{x} + \tau_{i_1 i_2}, \nu_{i_1 i_2}\right) : i_1 \in P_{j_1}, i_2 \in Q_{j_2|i_1} \right\}
$$

$$
\equiv \left\{ \left((\boldsymbol{\eta}_{i_1 i_2}^*)^\top \boldsymbol{x} + \tau_{i_1 i_2}^*, \nu_{i_1 i_2}^*\right) : i_1 \in P_{j_1}, i_2 \in Q_{j_2|i_1} \right\},
$$

for any $j_1 \in [m_1]$ and $j_2 \in [m_2]$. Consequently, we obtain that

$$
G = \sum_{j_1=1}^{m_1} \sum_{i_1 \in P_{j_1}} \exp(b_{i_1}) \sum_{j_2=1}^{m_2} \sum_{i_1 \in Q_{j_2|i_1}} \exp(\beta_{i_2|i_1}) \delta_{(\boldsymbol{a}_{i_1}, \boldsymbol{\omega}_{i_2|i_1}, \boldsymbol{\eta}_{i_1 i_2}, \tau_{i_1 i_2}, \nu_{i_1 i_2})}
$$

$$
= \sum_{j_1=1}^{m_1} \sum_{i_1 \in P_{j_1}} \exp(b_{i_1}^*) \sum_{j_2=1}^{m_2} \sum_{i_1 \in Q_{j_2|i_1}} \exp(\beta_{i_2|i_1}^*) \delta_{\boldsymbol{a}_{i_1}^*, \boldsymbol{\omega}_{i_2|i_1}^*, \boldsymbol{\eta}_{i_1 i_2}^*, \tau_{i_1 i_2}^*, \nu_{i_1 i_2}^*}
$$

$$
\equiv G_*.
$$

Hence, the proof is totally completed. $\qquad\square$

