# OpenReview forum: "On Expert Estimation in Hierarchical Mixture of Experts: Beyond Softmax Gating Functions"
_ICLR.cc/2025/Conference — ICLR 2025 Conference Withdrawn Submission_

### Official Review · Reviewer_qiEr · 2024-10-30

**Soundness:** 1
**Presentation:** 2
**Contribution:** 1
**Rating:** 3
**Confidence:** 3

**Summary:**

This paper explores the use of different gating functions in Hierarchical Mixture of Experts. In particular, the authors derive expert convergence bounds for Gaussian HMoE with Softmax-Softmax, Softmax-Laplace, and Laplace-Laplace gating. The authors  empirically evaluate models that employ the proposed HMoE layers on multiple multimodal and computer vision tasks.

While the presented analysis is solid, the main contributions of this work are highly incremental over the recent relevant work. Moreover, evaluation does not seem to entirely reflect the claims and the results of the theoretical analysis. As such, I recommend rejection of this work as of now.

**Strengths:**

- Although I have not verified it, the theoretical analysis appears exhaustive.

**Weaknesses:**

- **Limited novelty when stacked against recent related work** The main contribution of this work - convergence analysis for three HMoE variants - is highly incremental, as the main components used in this analyses (such as Voronoi losses) were already introduced in [1], while the Laplace gating for MoEs was introduced in [2]. Application of apparatus from [1] and [2] to HMoE instead of MoE is the only novel contribution of this work.
- **Empirical evaluation is not entirely aligned with the theoretical contribution of the paper** I find most of the experiments in the empirical part of the paper confusing when placed in context of the theoretical contribution. The theoretical part demonstrates the benefits of using Laplace-Laplace gating over the other variants, and I would expect the empirical part to be focused around supporting the theoretical findings. Instead, it appears the authors focus on showcasing the performence of their final HMoE-based model. Only Figure 2 compares the HMoE SS/LS/LL variants, while the rest of the empirical evaluation compares a single HMoE-based model to different multi-modal methods.
- **Unconvincing evaluation** As such, I am not entirely convinced that Laplace-Laplace gating offers consistent advantages also in the real-world, deep model setting. In line 521 the authors write: "best-performing Softmax-Laplace gating combination of HMoE in latent domain discovery". Does this mean that in some cases Laplace-Laplace has inferior performance? The authors should report the score of every variant also in Table 2 and Table 3.
- **Unclear purpose of empirical analysis** Similarly, the analysis from Figure 3 has the same issue. How does the choice of gating functions affect the token distribution? Would token distribution of a non-hierarchical MoE be significantly different? Without such a comparison the analysis is somewhat informative, but not relevant to the theoretical contribution of the paper.
- **Limited impact** In the introduction the authors mention the rising use of MoEs in modern foundational models. However, none of these use hierarchical routing, and the last work to tackle non-Gaussian HMoE was [3]. Given the community's lack of interest in HMoEs, the impact of this work may be limited.
- **Incomplete description of empirical evaluation** Every training hyperparameter should be provided for each experiment performed in the paper. The appendix describes the datasets used and data preprocessing only. The provided code is missing some of the experiments, e.g. for ImageNet. How many seeds were run for calculating the reported standard deviations?
- **Unfair comparison to MoE models** In some of the experiments the score of HMoE-based model is compared to non-hierarchical/standard MoE-based model. However, two levels of routing means two routers are used, resulting with HMoE having an effectively double the computational and memory resources spend on routing compared to MoE. How can be anyone sure that any gains in performance are due to the hierarchical nature of routing as opposed to additional computation and memory allocated for routing? Some recent works does utilize deeper routers, hinting that increased computation allocated for routing does help performance [4, 5].

***References:***

[1] Nguyen, Huy, TrungTin Nguyen, and Nhat Ho. "Demystifying softmax gating function in Gaussian mixture of experts." Advances in Neural Information Processing Systems 36 (2024).

[2] Han, Xing, et al. "Fusemoe: Mixture-of-experts transformers for fleximodal fusion." arXiv preprint arXiv:2402.03226 (2024).

[3] Shazeer, Noam, et al. "Outrageously Large Neural Networks: The Sparsely-Gated Mixture-of-Experts Layer." International Conference on Learning Representations. 2016.

[4] Chi, Zewen, et al. "On the representation collapse of sparse mixture of experts." Advances in Neural Information Processing Systems 35 (2022): 34600-34613.

[5] Zhang, Zhengyan, et al. "MoEfication: Transformer Feed-forward Layers are Mixtures of Experts." Findings of the Association for Computational Linguistics: ACL 2022. 2022.

**Questions:**

- The theoretical contribution of the paper uses Gaussian MoEs. In contrast, the empirical evaluation seems to use one of the variants of MoE layers, which became common in deep models in recent years. In comparison to Gaussian MoEs: 1) they use Top-k to achieve sparse computation 2) use plain two-layer neural networks as experts (not constrained to a single Gaussian). How relevant are the contributed convergence proofs to this type of MoE?

---

### Official Review · Reviewer_cFwL · 2024-11-03

**Soundness:** 3
**Presentation:** 1
**Contribution:** 3
**Rating:** 8
**Confidence:** 3

**Summary:**

This paper investigates hierarchical mixture of experts (HMoE) models, focusing on enhancing gating mechanisms beyond the conventional softmax function. The authors propose that varied gating functions, especially Laplace gating, improve HMoE's performance across different hierarchical levels by optimizing expert specialization and convergence rates. They conduct a theoretical analysis and show that Laplace gating functions accelerate expert convergence in comparison to softmax and a combination of softmax and Laplace gatings. Empirical results on multimodal and high-dimensional datasets (e.g., image classification, clinical data) demonstrate that HMoE with tailored gating configurations outperforms baseline mixture-of-expert models, particularly in scenarios requiring domain specialization and complex input handling.

**Strengths:**

- This paper firstly proposes HMoE models by incorporating alternative gating mechanisms, such as Laplace gating, which enable more flexible and efficient expert routing. This technical novelty allows HMoE to handle complex, multi-domain inputs with greater adaptability than traditional MoEs.

- This paper provides a thorough theoretical analysis, showing that varied gating functions, espically, Laplace gating functions can improve expert convergence rates. They establish bounds on parameter estimation, demonstrating that specific gating choices accelerate specialization, which is crucial for managing complex data.

- The experiments span diverse, realistic tasks, including medical prediction using datasets like MIMIC-IV, confirming HMoE’s real-world effectiveness, and show consistent gains over strong baselines, validating the model’s utility in high-dimensional, multi-domain settings.

**Weaknesses:**

- While the paper focuses on enhancing HMoE’s performance with alternative gating mechanisms, it lacks a detailed discussion on the associated computational complexity. A deeper analysis of the model’s resource demands and potential methods to mitigate these challenges would add valuable context, particularly for large-scale or resource-constrained applications.

**Questions:**

Please see weakness section.

---

### Official Review · Reviewer_HUor · 2024-11-04

**Soundness:** 2
**Presentation:** 2
**Contribution:** 3
**Rating:** 5
**Confidence:** 3

**Summary:**

This paper studies Hierarchical Mixture of Experts (HMoE).  First, theoretical convergence rates for HMoE using both softmax and lapalce gating functions are found in the case of hierarchical MOG-distributed data, relating number of samples drawn to the discrepancy between MLE parameters and the original true distribution parameters, finding Laplace-Laplace gating to converge faster than Softmax when the number of experts exceeds the true number of mixture components.  HMoE is then applied in practice to multimodal health data estimation problems using MIMIC-IV and eICU datasets, achieving performance similar to current SOA that uses explicit patient domain grouping, but without explicit partitions.  Some of the behavior of the model is illustrated by showing the breakdown of expert assignment according to input modality.

**Strengths:**

The theoretical analysis is very detailed, with some interesting results differing from single-gating MoE.

The use of HMoE appears effective at automatically routing multimodal data automatically among experts, which has the potential to make best use of computational resources compared to fixed subsystems for each mode.

The model obtains good results compared to other systems on the multimodal clinical datasets, and shows interesting behavior in the hierarchical assignments related to the modes in this setting.

**Weaknesses:**

While the theoretical work is very detailed, and has an interesting end result, I wasn't able to follow the arguments in detail, though I am not an expert in this sort of convergence theory.  To me, the main text of the theory section is a little too verbose and overly technical, especially since there is already a large appendix with details.  I've listed a number of questions below, but in particular:
  * I didn't see a clear summary sentence of what the results mean, along the lines of "This shows that n samples are required in order for the best MLE params to be within $O(n^{-1/2})$ of the true distribution's parameters, up to log factors".
  * Some of the equations seem unnecessary for a high-level understanding, e.g. I don't see a need to list the entire vornonoi loss function L in the main text, and also don't see a need for the "mixing measure" $G_*$.  The notation is also pretty busy, with lots of subscripts and hats/tildes/etc.  This makes it more difficult to read, and would be good to simplify as well.

For the experimental application, I didn't see an estimate of model size or computation ops in any of the comparisons, so it's hard to know if the model performed better or is simply larger than the baselines.  In particular, comparing against non-hierarchical MoE, all levels of gating should be included in op and param counts.

I also didn't see a section that corroborates the theoretical convergence rates experimentally on toy gaussian data used in the theorems.  Seeing measurements that converge at the described rates would make the theory more convincing and relatable to a wider audience.

**Questions:**

* Is soft or hard (sparse) assignment used at each level?  Since there are only 4 experts in each group and two high-level experts, top-k would need to be top-1 or 2 if there is sparse assignment.  Was this the case or did this use dense mixing?  And if sparse, how many experts were selected at each level?

* Multimodal Routing Distributions section:  It would be interesting to see the distributions for all blocks, not just the last one.  In particular, is there more separation between domains assigned to each expert in earlier blocks?  Layering can also have a domain factorization effect (eg in https://arxiv.org/abs/1312.4314).

* It also would be interesting to see the Softmax-Softmax models here, as well --- is there a difference in the qualitative behavior in these assignments using Laplace vs Softmax?

* In addition to a breakdown of expert assignment domain, it would be interesting to see the assignments according to the different patient latent domains described in comparison with SLDG at l.419.  Were any of these learned to be separated in the model automatically?

* l.528 "increasing the number of experts has a positive impact" in Fig 4(b)(c):  This looks to be the case for the number of inner experts, but I don't see this in the plot for the number of outer experts, which is flat.  But maybe setting this down to 1 would show a difference at least between 1 and 2?

* I don't see a need for the "mixing measure" $G_*$.  Why not just use best-fit parameter set $\hat \theta$, and say $\hat \theta = \arg\max_{\theta\in\Theta} \frac1n \sum_{i=1}^n \log(p_\theta(Y_i|X_i))$ ?

* I don't understand how the shapes in Eq 3 align.  In the second equation du/da = du/dw, a and w are two different shapes, a is [d k1] with k1 the number of first-level experts, w is [k1 d k2] with k2 the number of second-level experts.

* Eq 5 is very long, and could use an explanation of what it is (what it measures) beyond the fact that it is loss based on the cells.  For example, something like "loss function that measures the total distance from the closest true parameter, of each parameter in the parameter set" (if indeed that is what it is, I'm basing that description on just a reading of the first term, so I'm not sure).  Also, what if the cell for a true param index is empty?

---

### Official Review · Reviewer_e9fm · 2024-11-04

**Soundness:** 2
**Presentation:** 2
**Contribution:** 2
**Rating:** 3
**Confidence:** 2

**Summary:**

The paper investigates hierarchical mixture of experts (HMoE) and in particular the type of routing mechanism used in 2-layers HMoE (Laplace vs Softmax). They first derive an analysis to show that using Laplace gating (in both layers) instead of softmax leads to faster expert specialisation. They then verify this insights by comparing HMoE with different routing strategy on experiments involving complex-structured clinical data and the CIFAR-10/ImageNet classification benchmarks.

**Strengths:**

* Thorough analysis of the two-layers Gaussian Mixture HMoE
* Considers two different types of data (clinical data and images)

**Weaknesses:**

* The main weakness of the paper is the writing being very unclear. I'll list some examples below:
    * Jumbled transitions between sections. Example in line 126, where we suddenly talk about the "convergence of expert specialisation"(which is only again mentioned line 260) before transitioning to an entirely different topic. Another example is line 272 "moving from this section" while it is not clear at all what we (the reader) should take away from this section.

    * The mathematical notations are cumbersome and only briefly introduced
      * some notations could be omitted for the sake of readability (e.g. the $\ast$ notations in Equation 1)
      * Some notations could be simplified. For instance introducing random variables as $X$ and $Y$ in lines 108 but then only referring to their density $x$ and $y$
      * Overall, I find that the methods section could be shortened by (i) moving some descriptions in appendix, (ii) only underlying the key take-away and (iii) making notations more lightweight.


    * Some statements/motivation are not substantiated:
       * line 71: "there is a growing demand for models that can deliver accurate and individualized predictions for each subgroup. Therefore, it is worthwhile to study. HMoE, which can leverage the intrinsic information within complex input structures and achieve superior performance on corresponding tasks." -> I do not see.a good reason why HMoE are a better solution to that problem than e.g. increasing the number of experts in MoE, or a deeper structure, or any other techniques to make the decision space "more complex"

  * Limitations in experimental setting:
    * Some details are missing: such as how many experts are used, or even better, considering different number of experts. For the image classification benchmarks, it's very unclear how a "one-layer MoE module" are placed in the sparse MoE backbone: e.g. what is being fine-tuned ?
    * Since the propose HMoE has a deeper routing mechanism, the paper should also discuss efficiency of the method vs MoE when integrating the new routers.
    * I find the description of the experimental section in the abstract/intro quite misleading. First most of the experiments are on clinical data, which is different from *diverse real-world scenarios* (line 19 + 535). As for image classification, CIFAR or tiny-ImageNet are interesting benchmarks, but a bit far from real world tasks.

**Questions:**

* Line 130: I'm not sure what the citation refers to or if it is even needed ? Maximum Likelihood Estimation has been used before 2000.

---

### Note · Authors · 2024-11-16

I have read and agree with the venue's withdrawal policy on behalf of myself and my co-authors.